# Mutant huntingtin impairs neurodevelopment in human brain organoids through CHCHD2-mediated neurometabolic failure

Expansion of the glutamine tract (poly-Q) in the protein huntingtin (HTT) causes the neurodegenerative disorder Huntington's disease (HD). Emerging evidence suggests that mutant HTT (mHTT) disrupts brain development. To gain mechanistic insights into the neurodevelopmental impact of human mHTT, we engineered male induced pluripotent stem cells to introduce a biallelic or monoallelic mutant 70Q expansion or to remove the poly-Q tract of HTT. The introduction of a 70Q mutation caused aberrant development of cerebral organoids with loss of neural progenitor organization. The early neurodevelopmental signature of mHTT highlighted the dysregulation of the protein coiled-coil-helix-coiled-coil-helix domain containing 2 (CHCHD2), a transcription factor involved in mitochondrial integrated stress response. CHCHD2 repression was associated with abnormal mitochondrial morphodynamics that was reverted upon overexpression of CHCHD2. Removing the poly-Q tract from HTT normalized CHCHD2 levels and corrected key mitochondrial defects. Hence, mHTT-mediated disruption of human neurodevelopment is paralleled by aberrant neurometabolic programming mediated by dysregulation of CHCHD2, which could then serve as an early interventional target for HD.

Huntington's disease (HD) is a rare neurodegenerative disorder caused by inherited defects in the gene Huntingtin (*HTT*) encoding for the protein HTT. The mutant *HTT* (*mHTT*) gene exhibits an abnormal (>35) CAG expansion resulting in production of elongated repeats of glutamine (Q) in the poly-Q tract of the protein that could in turn become more prone to misfolding and intracellular aggregation[1,2]. HD is currently incurable and the mechanisms underlying the neurodegenerative process are not fully understood.

An increasing amount of evidence points towards impaired brain development in HD. In fact, although wild-type (WT) HTT is present in several tissues, it is expressed at the highest level in the brain, even before the completion of neuronal maturation[3]. In mice, targeted HTT disruption is embryonic lethal[4,5], brain-specific HTT inactivation leads to progressive neuronal defects[6], and extensive HTT reduction fails to support normal brain development[7]. WT HTT may thus exert a physiological role in neurodevelopment, including axonal transport[8], synapse development[9], neural rosette formation[10], neuronal migration[11], as well as regulation of neural progenitor cells (NPCs) and neurogenesis[12].

In accordance with a developmental component in HD, mHTT with a poly-Q tract longer than 60 causes severe juvenile forms of HD, which recapitulate features associated with neurodevelopmental disorders[13]. Even individuals with non-juvenile forms of HD can develop signs of impaired brain growth during childhood before the occurrence of clinical manifestation[14-16]. In mice, mHTT disrupts the division of cortical progenitors[17], and temporally-limited expression of

e-mail: jakob.metzger@mdc-berlin.de; alessandro.prigione@hhu.de

mHTT in early life is sufficient to recapitulate the disease phenotypes[18]. Importantly, the analysis of human fetuses carrying mHTT demonstrated similar neurodevelopmental defects, with diminished numbers of proliferating NPCs and higher number of NPCs that prematurely enter neuronal lineage specification[19].

An effective model system for dissecting the neurodevelopmental aspects of HD is represented by human induced pluripotent stem cells (iPSCs). The transcriptional signature of neurons differentiated from iPSC models of HD pointed toward a dysregulation of neurodevelopmental genes[20–22]. Cerebral organoids carrying mHTT exhibited aberrant organization and specification of NPCs, with immature ventricular zones[23]. The CAG repeat length could thus regulate the balance between NPC expansion and differentiation in cerebral organoids[24]. Micro-patterned neuruloids consisting of NPCs, neural crest, sensory placode and epidermis showed disrupted self-organization in the presence of mHTT, indicating faulty neurodevelopment[25].

To gain mechanistic insights into how mHTT impacts human brain development, we generated brain organoids from engineered isogenic iPSC lines in which we introduced mHTT (70Q) in one or both *HTT* alleles, or we eliminated the poly-Q stretch completely from the HTT gene. We found that biallelic 70Q introduction disrupts the development of brain organoids (unguided cerebral organoids and region-specific cortical organoids and midbrain organoids) causing defective NPC organization despite seemingly unaffected neuronal presence. To identify mechanisms regulating early developmental defects caused by mHTT, we focused on shared signatures across undifferentiated iPSCs and neural committed cells. We identified the mitochondrial protein coiled-coil-helix-coiled-coil-helix domain containing 2 (CHCHD2) as a top dysregulated factor. In accordance to the known function of CHCHD2 in mitochondrial integrated stress response (mISR)[26], neural cells carrying mHTT developed a mISR signature with disruption of mitochondrial dynamics, cristae morphology, and bioenergetics, with specific complex IV alteration and increased energy expenditures at rest, a condition known as hypermetabolism. In-frame elimination of the poly-Q tract in HTT reverted key defects in CHCHD2 expression and mitochondrial morpho-dynamics. In HD patient-derived neuronal cultures, CHCHD2 belonged to a network of factors associated with axon guidance, Hippo signaling, and mISR, and its down-regulation impaired neurite outgrowth capacity. Additionally, AAV-mediated overexpression of CHCHD2 in mutant NPCs rescued the mitochondrial phenotype. Hence, mHTT is linked to mitochondrial dysfunctions in early neural cells through the dysregulation of CHCHD2, which could potentially contribute to altering the neurodevelopment process in humans.

## Results

### mHTT impairs early neural progenitor organization in human cerebral organoids

We engineered iPSCs with a healthy genomic background (WT/WT) using CRISPR/Cas9[27,28]. The edited region within the *HTT* gene entailed both the CAG repeat stretch (poly-Q in the protein) and the CCG repeat stretch (poly-P in the protein) (Fig. 1a, Supplementary Fig. 1a). As template, we used an expanded tract composed of 70 CAG/CAA repeats mimicking mutations observed in patients. The resulting engineered iPSC lines were: i) WT/70Q carrying the expanded CAG tract only on one allele, ii) 70Q/70Q carrying the expanded CAG tract on both alleles, iii) 0Q/0Q with an in-frame deletion that eliminated the CAG and CCG stretches (Fig. 1a, b). We confirmed the successful editing by sequencing, PCR, and immunoblotting (Fig. 1c, Supplementary Fig. 1b, c, Supplementary Fig. 1e–h). The set of isogenic iPSC lines exhibited a normal karyotype (Supplementary Fig. 1i) and displayed no modification of top computationally predicted off-target sites (Supplementary Fig. 1d).

To assess the impact of mHTT on human brain development, we focused on the line 70Q/70Q, in which WT HTT was absent. In fact,

while heterozygous cells with expanded poly-Q tract have previously been used for the investigation of mHTT pathogenesis, to our knowledge a homozygous model of mHTT has not yet been studied using iPSCs. Hence, exploring the consequences of homozygous 70Q/70Q may allow us to shed light on the impact of mHTT on human neurodevelopment in the absence of potential compensatory effects exerted by WT HTT.

Using this homozygous mutant model (70Q/70Q) and its isogenic control (WT/WT), we applied an unguided cerebral organoid differentiation protocol (Fig. 1d). Cerebral organoids carrying mHTT (70Q/70Q) displayed significant disruption in cellular organization, including lack of ventricular zone-like neurogenic zones (Fig. 1e). NPCs appeared particularly impaired, as demonstrated by the reduced presence and spatial disorganization of cells positive for NPC markers SOX2 and FOXG1, and evident disruption of the tight junction marker ZO1 (Fig. 1e, Supplementary Fig. 2a). In accordance to features of neurodevelopmental impairment, cerebral organoids carrying mHTT displayed an overall reduced growth rate compared to WT organoids (Fig. 1f). Transcriptional analysis of cerebral organoids confirmed lower expression of progenitor markers (Fig. 1g). Additionally, the great majority of genes associated with the GO term "nervous system development" were downregulated in mutant organoids (Fig. 1h). Despite these alterations in the progenitor population, we could still detect neuronal cells in 70Q/70Q cerebral organoids, as indicated by the presence of markers TUJ1, MAP2, and SMI32 (Fig. 1e, Supplementary Fig. 2a, b). These findings are in line with previous data of mHTT-carrying human fetuses reporting the loss of proliferating NPCs, specific defects in ZO1 expression, and premature neuronal differentiation[19].

We next aimed to dissect the impact of mHTT on development of different brain regions and generated guided region-specific brain organoids (Fig. 2a). Neuroanatomical studies suggest that neurodegeneration in HD individuals is not limited to striatum but also affect brainstem and neocortex regions[29]. Therefore, we focused on cortical organoids and midbrain organoids. The presence of mHTT diminished the size development of both cortical and midbrain organoids (Fig. 2b, c). In both region-specific brain organoids, the overall size appeared altered already at relatively early stages, even before 10 days of differentiation (Fig. 2b, c, Supplementary Fig. 3a-d). We next checked for presence of toxic HTT aggregates during brain organoid development. We could not find significant presence of large-size HTT protein conformations after 70 days of in vitro culture of cortical organoids or midbrain organoids (Supplementary Fig. 2c, d).

To delve deeper into the impact of mHTT on neural progenitors and maturing neurons within developing brain organoids, we focused on midbrain organoids. In the protocol we used[30], midbrain organoids were smaller than cortical organoids and exhibited a simpler organization, with concomitant presence of neuronal and progenitor markers (Fig. 2d). We performed single-cell RNA-sequencing of 70Q/70Q and WT/WT midbrain organoids at 35 days. We identified four main cell populations: i) proliferative progenitors, ii) progenitors, iii) maturing neurons, iv) mature neurons (Fig. 2e, Supplementary Data 1). These four populations expressed specific gene markers, such as TOP2A for proliferative progenitors, PTPRZ1 for progenitors, ROBO3 for maturing neurons, and MAPT for mature neurons (Fig. 2g, Supplementary Data 1). In agreement with the findings seen in unguided cerebral organoids (Fig. 1e), 70Q/70Q midbrain organoids exhibited a marked reduction of "progenitors" and "proliferative progenitors" populations, despite the continued presence of mature neurons (Fig. 2e, f, Supplementary Fig. 3e, f).

Collectively, these data suggest that mHTT causes neurodevelopmental defects that occur early on during development and may specifically impact neural progenitors.

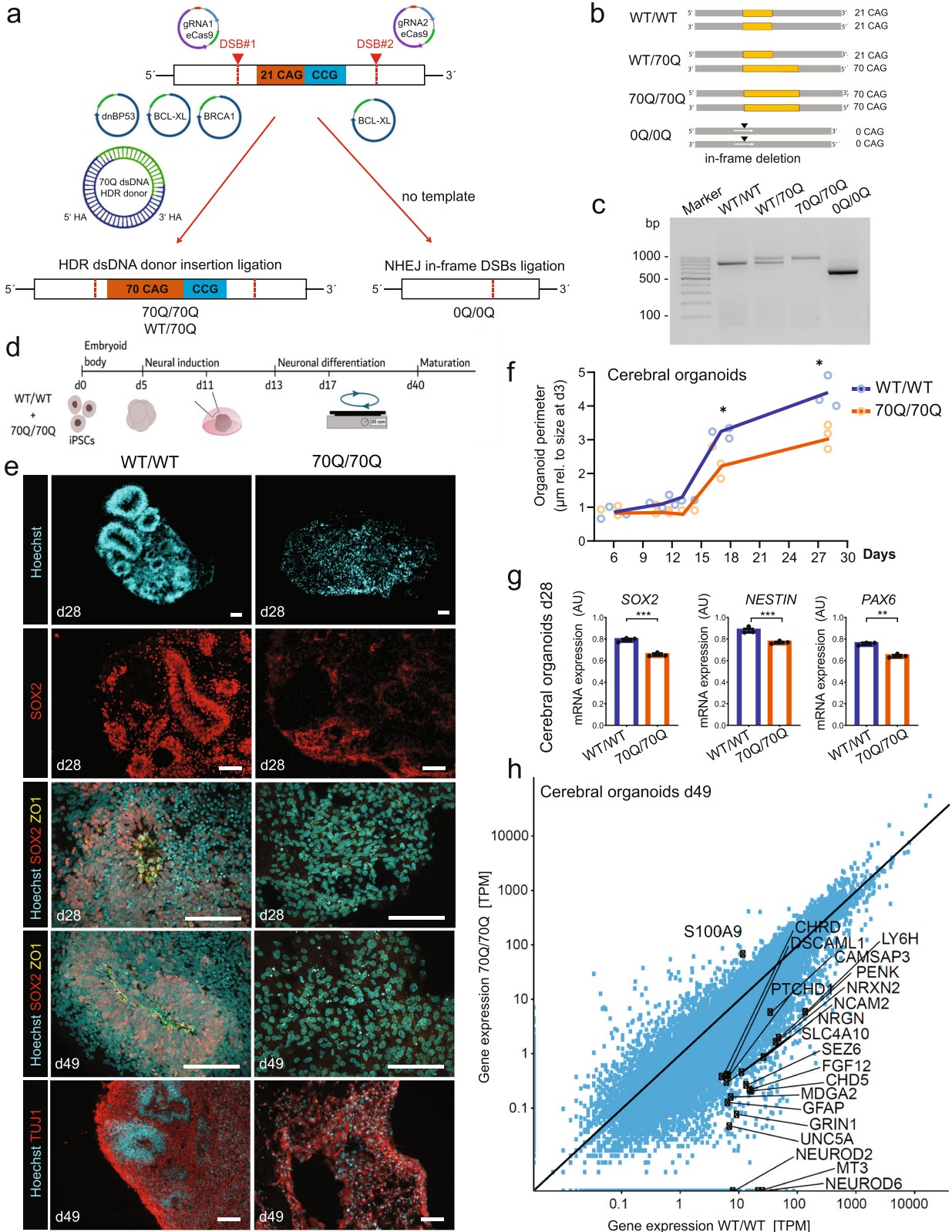

## CHCHD2 is a top dysregulated gene within the early neurodevelopmental signature of mHTT

We next aimed to transcriptionally dissect the mechanisms underlying the early neurodevelopment defects caused by mHTT. PolyQ tract motifs found in transcription factors can act as transcriptional regulating domains by facilitating the binding between transcription factors and transcriptional regulators[31]. To thoroughly analyze mHTT's

transcriptional impact we conducted total RNA sequencing of 70Q/70Q and WT/WT at different time points of neurodevelopment: i) iPSCs, ii) NPCs, iii) cerebral organoids at day 28 (for early neurogenesis), iv) cerebral organoids at day 49 (for late neurogenesis and astrogenesis) (Fig. 3a, Supplementary Data 2).

We focused on genes that were dysregulated early and persistently throughout the differentiation trajectory. We identified a

**Fig. 1 | Engineered iPSCs carrying mHTT give rise to neurodevelopmentally impaired cerebral organoids. a** Genome editing approach in control iPSCs (WT/ WT) using two double-strand breaks (DSB) sites to modify the HTT genomic region encompassing the CAG/CAA and CCG repeats stretches. To generate iPSCs carrying elongated CAG in one allele (70Q/WT) or in both alleles (70Q/70Q), we promoted homology direct repair (HDR) (with plasmids BCL-XL and BRCA1), inhibited non-homologous end joining (NHEJ) (with plasmid dnBP53), and provided a HDR donor dsDNA plasmid carrying 70Q repeats and homology arms. We harnessed NHEJ to obtain iPSCs with in-frame deletion of the CAG/CCG region (0Q/0Q). **b** Overview of the engineered isogenic iPSC lines. **c** PCR analysis of HTT in the isogenic iPSC lines. Data were repeated in three independent experiments. **d** Schematics of the protocol to generate unguided cerebral organoids from isogenic iPSC lines WT/WT and 70Q/70Q. **e** Immunostaining in cerebral organoids at day 28 and 49 showing

defective cytoarchitecture and neural progenitor cell (NPC) organization in 70/ 70Q. Data were repeated in three independent experiments. Scale bar: 100 μm. **f** Growth rate of cerebral organoids with respect to the initial perimeter size measured at day 3. *n* = 3 independent organoid differentiations (dots) per line. Each dot represents the average size of all cerebral organoids measured in one biological replicate. *$p < 0.05$ WT/WT vs. 70Q/70Q; unpaired two-tailed Welch *t* test. **g** qPCR analysis of NPC markers in cerebral organoids at day 28. Mean ± s.e.m.; *n* = 3 independent biological replicates (dots) per line; ***$p < 0.001$ WT/WT vs. 70Q/70Q; unpaired two-tailed *t* test (AU=arbitrary units). Four organoids were pooled for each individual RNA isolation. **h** Gene expression analysis of cerebral organoids at day 49 highlighting the genes belonging to the GO term "nervous system development" (GO:0007399).

consistent expression signature comprising 47 genes that were either down- or up-regulated in 70Q/70Q compared to WT/WT across all developmental stages (Fig. 3a, Supplementary Fig. 4a, b, Supplementary Data 3). Within this signature, the most downregulated gene in mHTT-expressing cells was coiled-coil-helix-coiled-coil-helix domain containing 2 (CHCHD2) (Fig. 3b). Other downregulated factors included genes involved in development (GBX2, PAX7, PAX8, LHX1, LMX1B, BARX1), neuronal development (POUF33, POUF3F2, TAF9B, EN2), and glutamate metabolism (SLC1A2, GAD1, GAD2) (Fig. 3b, Supplementary Fig. 4b). Upregulated factors in mHTT condition were genes associated with detoxification (GSTM1, IAH1), protein translation (EIF1AY, KDM5D), and protein degradation (USP9Y). In addition to this common signature, genes involved in nervous system development were downregulated in NPCs and cerebral organoids carrying mHTT (Supplementary Fig. 4d).

We performed untargeted proteomics of iPSCs and NPCs to determine whether any of the observed transcriptional changes induced by mHTT were recapitulated at the protein level. We identified metabolism-related proteins dysregulated in both iPSCs (46 metabolic proteins out 321) and NPCs (13 out of 484) (Fig. 3c, d, Supplementary Data 4, 5). Among these proteins, CHCHD2 was significantly downregulated in 70Q/70Q NPCs compared to WT/WT NPCs (Fig. 3d). Given that transcriptomics and proteomics identified CHCHD2 as dysregulated upon 70Q introduction, we inspected the transcription level of the whole family of coiled-coil-helix-coiled-coil-helix (CHCH) domain-containing proteins. These are proteins imported into the mitochondrion that have been suggested to play a role in neurodegenerative diseases[32,33]. Although other CHCH family members were dysregulated by mHTT, CHCHD2 appeared most significantly affected in both iPSCs and NPCs carrying the elongated CAG repeat in both alleles (70Q/70Q) or in only one allele (WT/70Q) (Fig. 3e). We also found a similar dysregulation of CHCHD2 in previously published datasets of neuruloids carrying mHTT[25] (Fig. 3e).

To validate the decreased expression of CHCHD2 in cells carrying mHTT, we performed immunostaining for CHCHD2 in cerebral organoids and NPCs and applied a semi-automated pipeline to quantify the expression of CHCHD2. Using this approach, we confirmed a decreased level of CHCHD2 protein in cerebral organoids and NPCs carrying mHTT (70Q/70Q) compared to WT/WT (Fig. 3f, g). NPCs carrying mHTT in only one allele (WT/70Q) also showed an overall reduced abundance of CHCHD2, although this difference was not statistically significant (Fig. 3g). This may be attributed to the insignificant reduction of HTT amount that we observed in the WT/70Q line (Supplementary Fig. 1c). The elimination of both poly-Q and poly-P tracts (0Q/0Q) significantly increased CHCHD2 protein expression in NPCs (Fig. 3g). We further confirmed these results using immunoblot analysis of CHCHD2, which showed increased expression in 0Q/0Q NPCs and reduced expression in 70Q/70Q NPCs when compared to WT/WT NPCs (Supplementary Fig. 4e, f).

In summary, mHTT leads to downregulation of CHCHD2 during early neurodevelopment and the presence of CHCHD2 can be restored

by eliminating the CAG/CCG region, suggesting a potential role for CHCHD2 in HD neuropathology.

## CHCHD2 dysregulation by mHTT is associated with a mISR signature and defective mitochondrial morpho-dynamics

We next investigated the downstream consequences associated with the dysregulation of CHCHD2. CHCHD2 is known be imported into the mitochondria and may also act as a transcription factor by translocating to the nucleus[34]. It was indeed listed among nucleic acid binding genes in our transcriptomics dataset (Supplementary Fig. 4c). Using semi-automated quantification of immunostained images, we confirmed that both expression and nuclear localization of CHCHD2, as indicated by co-localization with the DNA stain Hoechst, were diminished in NPCs and cerebral organoids (COs) carrying mHTT 70Q/70Q (Fig. 4a, b). Midbrain organoids carrying 70Q/70Q also displayed lower CHCHD2 presence (Supplementary Fig. 5a).

CHCHD2 plays a role in mitochondrial integrated stress response (mISR) and mitochondrial dynamics[26]. Accordingly, genes associated with mISR were dysregulated upon 70Q introduction (Fig. 4c). This was particularly evident for NPCs (70Q/70Q and WT/70Q), where mISR-related genes were significantly upregulated compared to WT/WT (e.g., ATF3, ATF4, DDIT3, CHAC1, PCK2). mISR dysregulation was present also in cerebral organoids at day 49 and HD neuruloids, although to a lower extent. In addition, mHTT altered the expression of genes regulating mitochondrial dynamics in mutant NPCs and cerebral organoids, with upregulation of genes related to mitochondrial fusion (e.g., FIS1, MNF1, MNF2) and changes in the expression of genes associated with mitochondrial fission (OMA1, MFF, DNM1L) (Fig. 4c). In conditions with impaired mitochondrial oxidative phosphorylation (OXPHOS), mISR is associated with dysregulation of mitochondrial quality control and mitochondrial dynamics, and cellular senescence[35]. Indeed, genes related to mitochondrial quality control, mitochondrial biogenesis, and senescence were affected by mHTT, with upregulation of quality control genes in NPCs (70Q/70Q and WT/70Q) and cerebral organoids (70Q/70Q) (e.g., YMEIL1, LONP1, LONP2, HSPA1) (Supplementary Fig. 5e).

To gain additional insights into the impact of mHTT in neural progenitors and the potential relationship with CHCHD2 dysregulation, we carried out long-read transcriptomics in 70Q/70Q NPCs and WT/WT NPCs. Since alternative splicing plays a central role during neurogenesis[36] and is particularly disrupted in HD[37], we aimed to discover potential differences in NPC isoform usage caused by mHTT (Supplementary Fig. 5b). We identified unique isoforms (128712 in WT/ WT and 152214 in 70Q/70Q) but observed no differences in the overall number of isoforms per gene in mutant NPCs compared to WT (Supplementary Fig. 5c, d). Nonetheless, we observed specific different isoform patterns for genes involved in mitochondrial dynamics, including NRF1, YME1L1, DNM1L, and MFN2 (Fig. 4d). Changes in DNM1L, encoding for a protein also known as dynamin-related protein 1 (DRP1) are of particular interest, since this protein has been found to

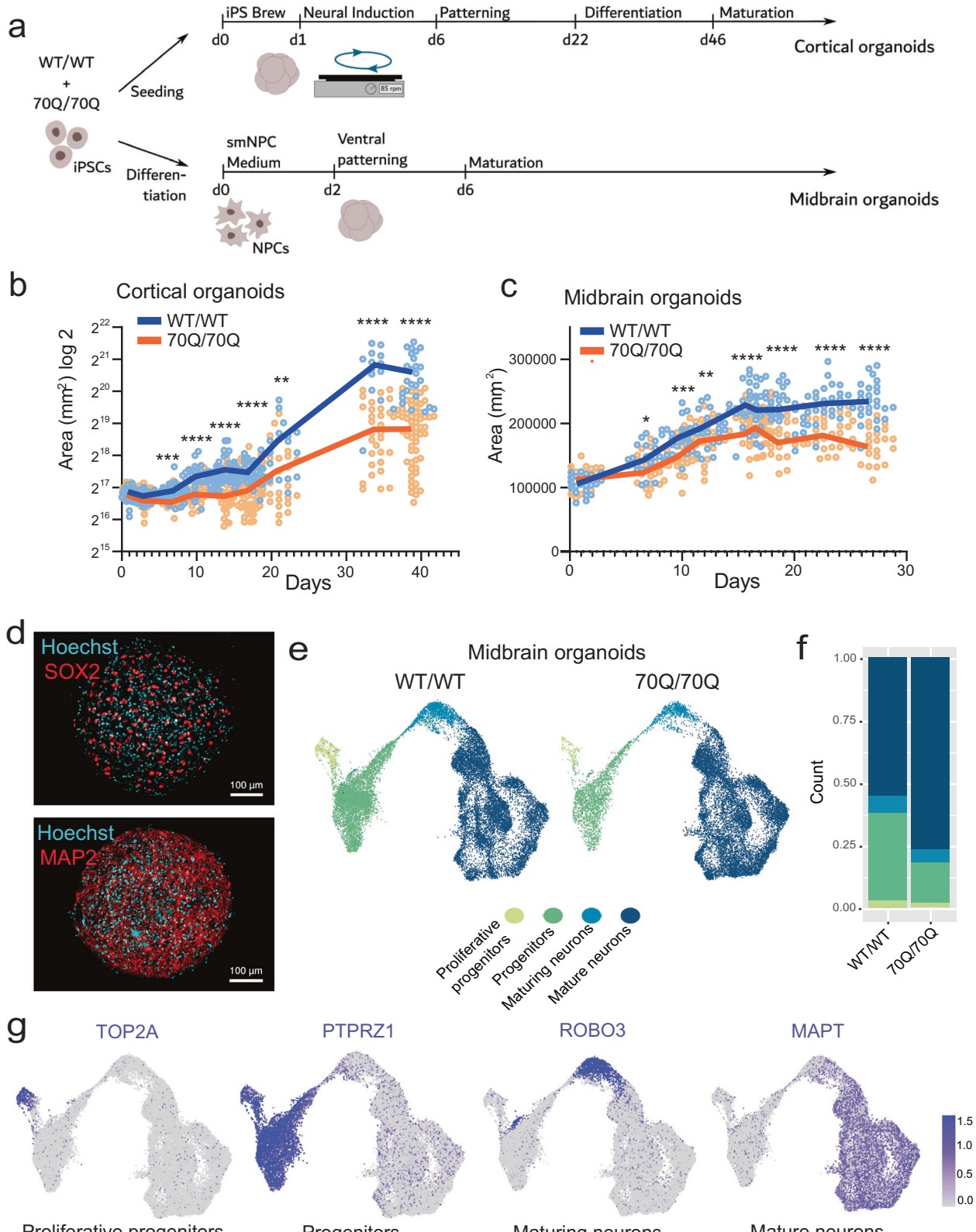

interact with HTT[38,39], and its transcript variants can influence the state of mitochondrial dynamics[40].

Altered expression and variant patterns of genes involved in mitochondrial dynamics suggested possible alterations in mitochondrial morpho-dynamics. We thus evaluated mitochondrial dynamics and mitochondrial network morphology in cerebral organoids based on the expression pattern of the mitochondrial outer membrane

marker TOM20 (Fig. 4e)[41]. TOM20 is a protein located in outer mitochondrial protein that is typically employed for visualizing mitochondrial structures. We found that mutant cerebral organoids exhibited elevated mitochondrial footprint, which we defined as the concomitant existence of fragmented mitochondria and abnormally large mitochondrial network structures (Fig. 4e, white arrows), leading to an overall increase in the intensity of TOM20-positive signals

**Fig. 2 | mHTT compromises development and neural progenitor population in region-specific brain organoids. a** Schematics of the protocol to generate guided region-specific brain organoids from iPSC lines WT/WT and 70Q/70Q. For cortical organoids, we started the differentiation from iPSCs, for midbrain organoids we started from NPCs. **b** Growth curve of cortical organoids. Dots represent individual organoids over at least two independent experiments. We compared organoid size at defined time points (day 1, 2, 7, 10, 14, 17, 21, 34, 39): ***$p < 0.001$, ****$p < 0.0001$ WT/WT vs. 70Q/70Q; two-tailed Mann-Whitney U test. **c** Growth curve of midbrain organoids. Dots represent individual organoids over at least two independent experiments. We compared organoid size at defined time points (day 1, 7, 10, 12, 16, 17, 19, 23, 27): ***$p < 0.001$, ****$p < 0.0001$ WT/WT vs. 70Q/70Q; two-tailed Mann-Whitney $U$ test. **d** Midbrain organoids stained for neural progenitor marker SOX2

and neuronal marker MAP2 to show their uniform distribution within individual organoids. Data were repeated in two independent experiments. Scale bar: 100 μm. **e** Single-cell analysis of midbrain organoids. Uniform manifold approximation and projection (UMAP) showing the overall cell composition of WT/WT and 70Q/70Q midbrain organoids. Each sample was sequenced in three biological replicates, with each replicate containing around 48 individual midbrain organoids. Shown here are merged UMAP images for all three replicates for WT/WT and three replicates for 70Q/70Q. **f** Quantification of the four annotated cell populations in WT/WT and 70Q/70Q midbrain organoids. **g** Distribution of exemplary markers for each of the four cell populations composing the midbrain organoids (see Supplementary Data 1).

(Fig. 4f). Both small mitochondrial structures and large mitochondrial structures appeared increased in mutant cerebral organoids (Supplementary Fig. 5f). The overall increase in TOM20 levels was more evident in mutant cerebral organoids at day 49 compared to day 28, suggesting that defects in mitochondrial dynamics might become more severe over time (Fig. 4f). Using a semi-automated quantification of immunostaining images, we confirmed higher relative levels of TOM20 protein cerebral organoids carrying 70Q/70Q compared to WT/WT and in 70Q/70Q NPCs compared to WT/WT NPCs. The TOM20 increase did not reach significance in NPCs with WT/70Q (Fig. 4g). These latter findings are consistent with the observation that WT/70Q did not show downregulation of CHCHD2 (Fig. 3g) or reduction of HTT levels (Supplementary Fig. 1c). At the same time, elimination of the poly-Q/poly-P region was sufficient to repress the abnormal TOM20 increase in NPCs (Fig. 4g).

Taken together, neural cells carrying mHTT exhibited CHCHD2 dysregulation and developed a mISR signature and aberrant mitochondrial morpho-dynamics, and these features could be reverted upon elimination of the CAG/CCG repeat region.

### mHTT in neural progenitors causes aberrant metabolic programming and mitochondrial complex IV defects

Following the identification of defects in mitochondrial morpho-dynamics caused by mHTT, we sought to investigate mitochondrial morphology in NPCs by electron microscopy (Fig. 5a). We did not observe significant changes in the size of mitochondria in 70Q/70Q NPCs compared to WT/WT NPCs (Supplementary Fig. 5g). However, elimination of the CAG/CCG region led to an increase in mitochondrial roundness and mitochondrial area in 0Q/0Q NPCs compared to WT/WT NPCs (Supplementary Fig. 5g). Whether these changes could have detrimental consequences remains to be determined. A specific defect that we observed in 70Q/70Q NPCs was the presence of mitochondrial cristae oriented longitudinally with respect to the outer membrane (Fig. 5a, black arrowhead; Fig. 5b) instead of transversally (Fig. 5a, black arrows; Fig. 5b). Such mitochondrial cristae alteration has been suggested to develop as a consequence of mHTT[42,43]. The mitochondrial cristae defect was reverted in 0Q/0Q NPCs, in which cristae direction appeared to be transverse as in the case of WT (Fig. 5a, b).

To elucidate the functional metabolic consequences associated with the observed changes in mitochondrial morpho-dynamics caused by mHTT, we carried out functional metabolic pathway analyses based on our proteomics dataset. Such an approach was previously carried out to pinpoint a disruption in mitochondrial metabolism in cerebral organoids from autism spectrum disorders[44]. The kinetic model used comprises the major cellular metabolic pathways of energy metabolism in neural cells, including ion membrane transport and mitochondrial membrane potential (Supplementary Data 7). Introduction of 70Q mutation in NPCs led to increased lactate production and diminished mitochondrial oxidation (Fig. 5c). Accordingly, gene set enrichment analysis of proteomics indicated that 70Q/70Q NPCs exhibited lower OXPHOS and higher glycolysis/gluconeogenesis compared to WT/WT NPCs (Fig. 5d). NPCs carrying mHTT also showed

higher glucose utilization at rest (Fig. 5e). This feature of elevated resting energy expenditure is also known as hypermetabolism, a condition developing upon impaired OXPHOS function in primary mitochondrial diseases that is associated with mISR activation[35]. We next investigated the ratio of NAD/NADH and mitochondrial NAD/NADH, crucial indicators of metabolic and redox homeostasis[45]. Both ratios appeared reduced in mutant NPCs compared to wild-type, indicating the presence of mitochondrial impairment and reductive stress (Fig. 5f).

Some of the metabolic changes detected in mutant NPCs were already present in mutant iPSCs, including increased lactate production coupled with reduced mitochondrial oxidation, and increased substrate utilization at rest (Supplementary Fig. 6a–c). Mutant iPSCs also exhibited a reduction in ATP/ADP ratio and maximal ATP production (Supplementary Fig. 6f, g). In contrast to NPCs however, the ratios of NAD/NADH and mitochondrial NAD/NADH were less significantly altered in mutant iPSCs compared to WT/WT iPSCs (Supplementary Fig. 6d, e). We speculate that this could be due to the fact that iPSCs predominantly rely on glycolytic metabolism, while NPCs already start to depend on OXPHOS as a consequence of initiation of neural commitment[46,47]. Hence, the functional consequences of defective metabolic programming may become more evident when cells acquire higher mitochondrial capacity during neurogenesis.

We next assessed whether these metabolic defects induced by mHTT could impact the protein organization of the mitochondrial respiratory chain proteins. With SDS-PAGE, we identified a specific decrease in the levels of complex IV proteins in 70Q/70Q NPCs compared to WT/WT NPCs that was restored in 0Q/0Q NPCs (Fig. 5g, h). This was in agreement with the proteomics findings in NPCs, where the expression of the complex IV subunit COX7A2 was among the most downregulated proteins in mutant NPCs compared to wild-type NPCs (Fig. 3d). Of note, COX7A2 is a key determinant for the structural and functional organization of the mitochondrial respiratory chain[48]. Blue native PAGE analysis further highlighted a reduction of complex IV assembly in 70Q/70Q NPCs compared to WT/WT NPCs that was partially reverted in 0Q/0Q NPCs (Fig. 5i, j, Supplementary Fig. 6h). Accordingly, transcriptional analysis confirmed the dysregulation of mitochondrial complex IV genes (e.g., COX7A2) and mitochondrial complex IV assembly factors (e.g., SCO2) in NPCs, cerebral organoids and neuruloids carrying mHTT (Supplementary Fig. 6i, j).

Collectively, these findings suggest that mHTT caused hypermetabolism and defects in mitochondrial cristae and complex IV assembly that could be reversed by eliminating the CAG/CCG region. This bioenergetic impairment may start early during development upon initiation of neural commitment.

### Identification of a CHCHD2-linked gene/protein/metabolite network in HD patient-derived neurons

We next sought to determine whether functional mitochondrial defects caused by mHTT could occur in human neurons in a cell-autonomous manner, independently of potential disruption of other cell types presented within cerebral organoids. Several publications

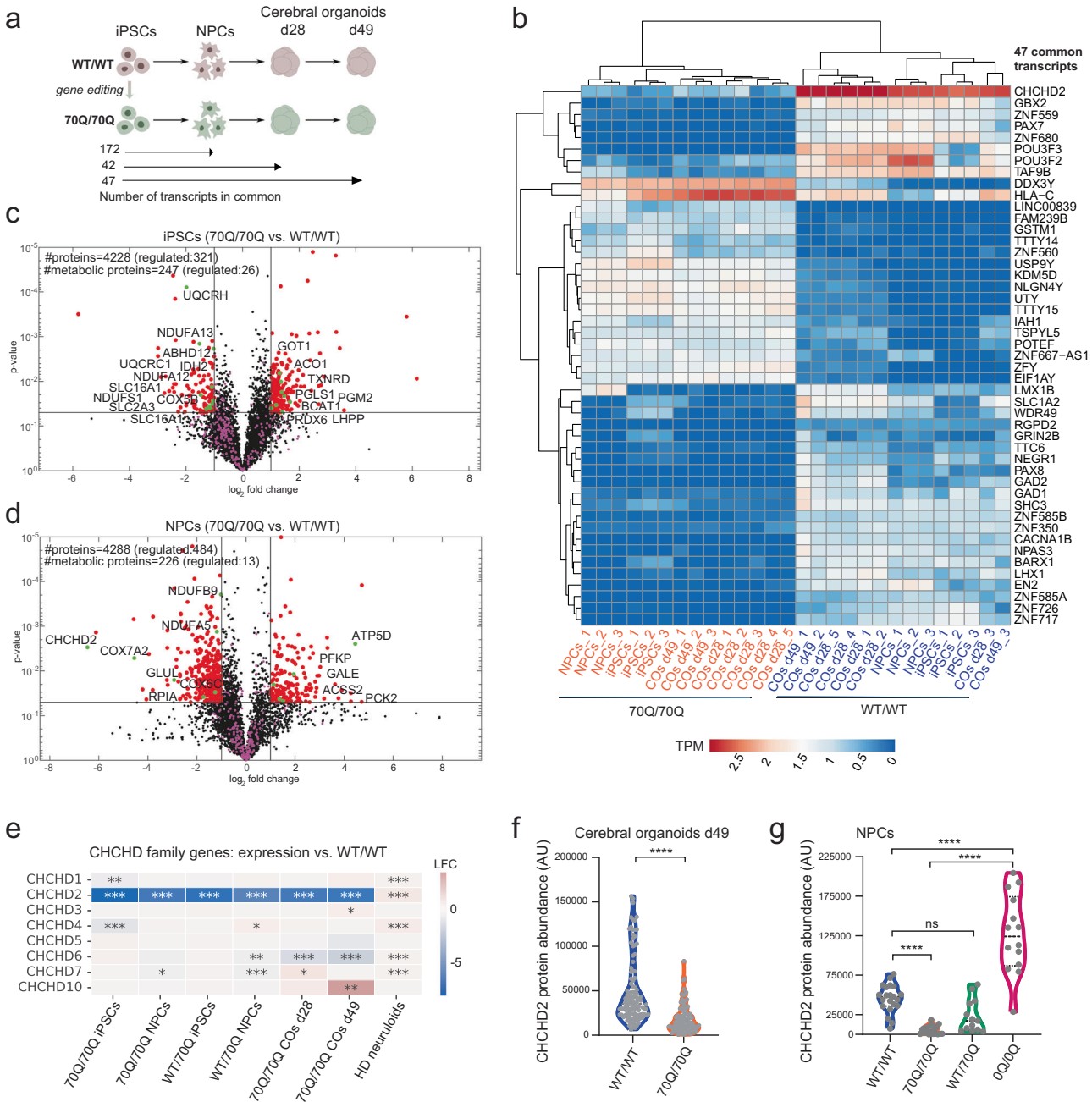

**Fig. 3 | Omics analyses of mHTT-expressing cells across neurodevelopmental stages highlight the dysregulation of CHCHD2. a** Schematics of RNA sequencing experimental set up for isogenic lines WT/WT and 70Q/70Q. Below are reported the number of transcripts uniquely in common across the neurodevelopmental stages (iPSCs, NPCs, cerebral organoids (COs) at day 23 and COs at day 49) (see Supplementary Data 2, 3). **b** Heatmap showing the 47 transcripts uniquely in common across the neurodevelopmental stages. TPM: transcripts per million. **c, d** Volcano plot of proteomic datasets in iPSCs and NPCs showing statistical significance (p value) versus magnitude of change (log₂ fold change) based on two-sample two-tailed t-test with Benjamini-Hochberg (BH, FDR of 0.05) correction for multiple testing (see Supplementary Data 4–6). Red dots indicate the significantly differently regulated proteins (right quadrant: upregulated in 70Q/70Q; left

quadrant: downregulated in 70Q/70Q). Metabolic proteins were later used for the proteomic-driven functional metabolic analysis (see Supplementary Data 6). Green dots highlight regulated proteins that are explicitly named. **e** Heatmap of log-fold change (LFC) comparisons of differential gene expression of genes belonging to the CHCHD family. Data obtained from RNA sequencing of iPSCs, NPCs, and COs; *p < 0.05, **p < 0.01, ***p < 0.001; 70Q/70Q and WT/70Q vs. WT/WT, HD neuruloids vs. WT neuruloids[25]; two-sided likelihood ratio test, without multiple comparison adjustments. **f, g** Quantifications of protein abundance of CHCHD2 based on immunostaining performed in cerebral organoids and NPCs. The amount of positive CHCHD2 signal per image was normalized to the Hoechst signal. 4****p < 0.0001, ns: not significant; two-tailed Mann-Whitney U test.

investigated iPSC-derived neurons carrying mHTT, but the various differentiation protocols employed typically resulted into a heterogeneous mixture of neuronal and glial population[21,41]. Moreover, we wanted to make sure that our findings based on engineered cells reflected phenotypes actually occurring in patient-derived cells.

To address these aspects, we used three iPSC lines that we recently derived from individuals affected by HD: HD1 with WT/180Q[49] and HD2 and HD3 with WT/58Q and WT/44Q, respectively[50] (Fig. 6a). We compared the three HD iPSC lines to three iPSC lines derived from healthy control individuals (C1, C2, C3). To obtain a homogenous

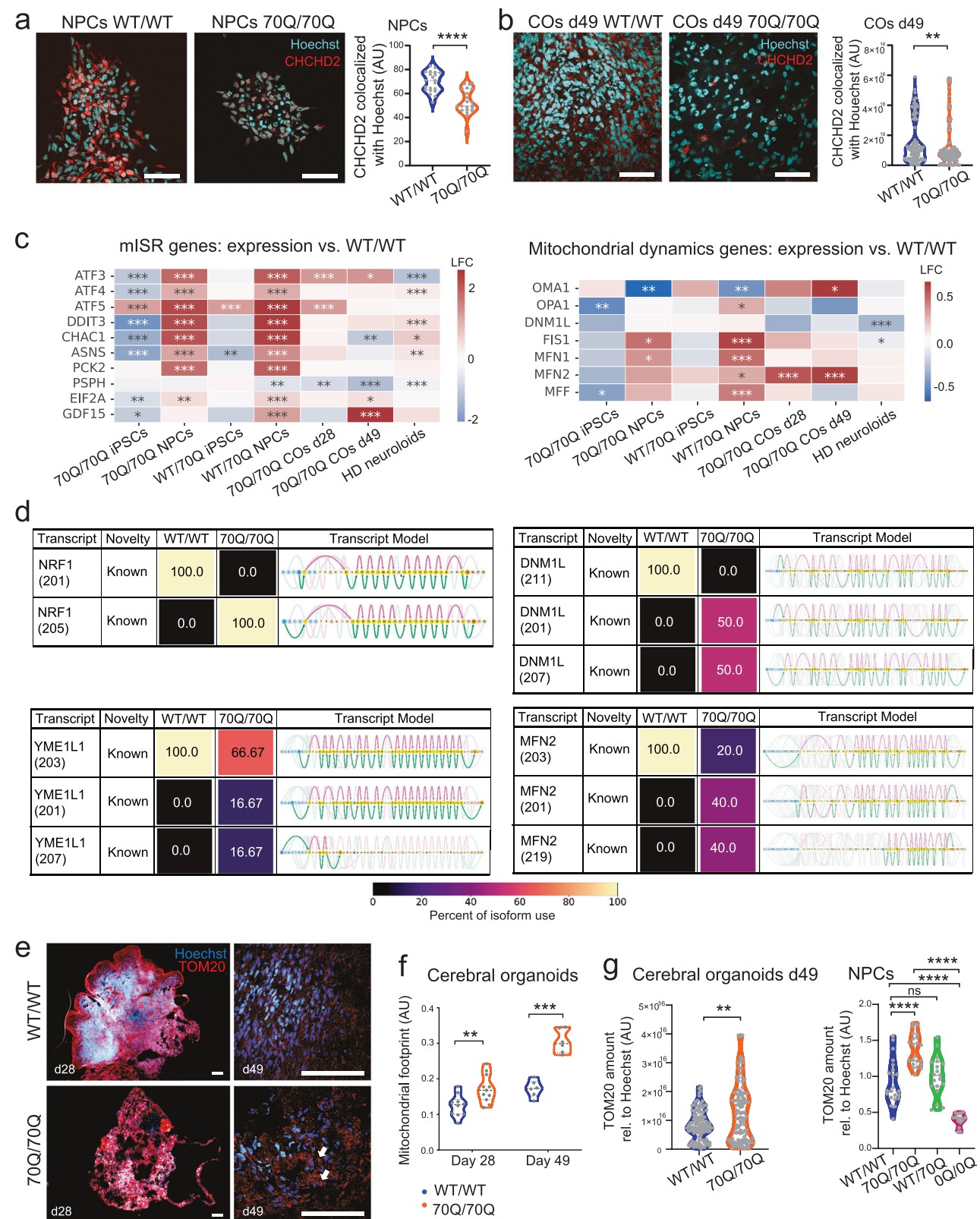

population of neurons, we overexpressed the transcription factor Neurogenin 2 (NGN2)[51]. The conventional approach for NGN2 neuron derivation is based on direct reprogramming of iPSCs into neurons[51]. However, this approach is rather artificial, as it bypasses the NPC stage, and therefore the neuronal generation is not affected by processes influencing NPC physiology. Given that NPCs carrying mHTT exhibited functional defects, we opted for an approach that was instead more

physiological: we first generated NPCs and then applied NGN2 over-expression on the obtained NPCs (Fig. 6b). In this way, we reasoned that we may be able to assess the ability of cells carrying mHTT to generate neurons even in the presence of potential defects at the NPC stage. The constructs we used contained a GFP reporter and allowed derivation of homogeneous neuronal cultures expressing neuronal and synaptic markers (Supplementary Fig. 7a). Using this approach, we

**Fig. 4 | mISR and altered mitochondrial morpho-dynamics are associated with CHCHD2 dysregulation in mHTT-carrying neural cells. a, b** Representative immunostaining and related quantifications of CHCHD2 protein colocalization with nuclear staining Hoechst in NPCs and in cerebral organoids (COs) at day 49. Dots represent individual images collected from at least two biological replicates for NPCs (n = at least 35 individual images per sample) and COs (n = at least 77 individual images per sample). **p < 0.01, ****p < 0.0001; two-tailed Mann-Whitney U test. Scale bar: 100 μm. **c** Heatmap of log-fold change (LFC) comparisons of genes involved in mitochondrial integrated stress response (mISR) and mitochondrial dynamics. Data obtained from RNA sequencing of iPSCs, NPCs, and COs; *p < 0.05, **p < 0.01, ***p < 0.001; 70Q/70Q and WT/70Q vs. WT/WT, HD neuruloids vs. WT neuruloids[25]; two-sided likelihood ratio test, without multiple comparison adjustments. **d** Isoform usage in NPCs determined with long-read transcriptomics.

Percentage of use in WT/WT NPCs and 70Q/70Q NPCs and related transcript model are shown. **e, f** Representative immunostaining and related quantification of mitochondrial footprint (comprising both small and large mitochondrial structures) based on TOM20 signal in cerebral organoids from 70Q/70Q and WT/WT. Dots represent individual images collected over three independent experiments. **p < 0.01, ***p < 0.001, unpaired two-tailed t test (AU = arbitrary units). Scale bar: 100 μm. **g** Quantifications of protein abundance of TOM20 based on immunostaining performed in cerebral organoids and NPCs. The amounts of positive TOM20 signal per image was normalized to Hoechst signal. Dots represent individual images collected from at least two biological replicates for NPCs (n = at least 35 individual images per sample) and COs (n = at least 77 individual images per sample) (AU = arbitrary units). **p < 0.01, ****p < 0.0001, ns: not significant; two-tailed Mann-Whitney U test.

found that all patient-derived NPCs generated NGN2 neurons without visible differences in terms of efficiency and neuronal appearance compared to NGN2 neurons derived from control NPCs (Supplementary Fig. 7b, c).

We applied a multi-omics approach to NGN2 neurons derived from the three HD individuals and the three controls (Fig. 6b). Principal component analyses of transcriptomics, proteomics, and metabolomics showed distinctions between HD neurons and control neurons, although the distinction was less clear in the case of metabolomics due to higher variability in the control samples (Supplementary Fig. 7d, Supplementary Data 7–10). Nonetheless, it appeared that all HD neurons clustered closely in all omics irrespective of different polyQ lengths. These results suggest that the presence of mHTT alters the global expression pattern of human neurons in a cell-autonomous manner and regardless of the number of CAG repeats.

Upregulated genes in HD neurons were related to pathways involved in synaptic signaling and axon guidance, and to diseases associated with neurodevelopmental defects (Fig. 6c). Similarly, upregulated proteins in HD neurons were related to pathways and diseases involved in axon guidance, autophagy, mTOR, and neurodegeneration, including movement diseases and HD (Fig. 6d). DNM1L also appeared among these dysregulated proteins (Fig. 6d). Conversely, downregulated genes/proteins in HD neurons were associated with carbon metabolism, Hippo signaling, and muscular disorders (Supplementary Fig. 7e–h, Supplementary Data 7–10). Axon guidance appeared also among the downregulated pathways, suggesting that this process was particularly affected by mHTT (Supplementary Fig. 7e). Differentially altered metabolites in HD neurons grouped based on structural similarities included metabolites related to NAD and mitochondrial Acetyl-CoA and to lactate and lipid metabolism (Fig. 6e, Supplementary Fig. 7i, j, Supplementary Data 7–10). To validate the metabolomics data, we assessed the bioenergetic profiles of NGN2 neurons. HD neurons exhibited defective mitochondrial function, with diminished ATP production and maximal respiration (Fig. 6f, Supplementary Fig. 7k). At the same time, the glycolytic capacity of HD neurons was also compromised, as seen by significantly reduced basal glycolysis and lactate production (Fig. 6f, g). Altogether, the presence of mHTT in pure human neurons caused on one hand an overall defect in cellular bioenergetics, and on the other hand it led to aberrant expression of genes and proteins involved in neurodevelopment, axon guidance, and synaptic activity.

To gain additional insights into these changes and their correlation, we integrated the various omics datasets. First, we looked at protein co-expression networks. The biggest cluster of co-expressed proteins (n = 272) contained three members of the CHCH domain-containing proteins (CHCHD1, CHCHD2, and CHCHD5) (Supplementary Fig. 8a). Enrichment analysis for upstream transcription factors of these 272 co-expressed proteins highlighted components related to neurodevelopment and Hippo signaling, including SIN3A, and to human diseases such as developmental delay and mental retardation

(Supplementary Fig. 8b, c). Next, we performed multi-omics integration. The network of significantly dysregulated transcripts, proteins and metabolites along with predicted intermediate miRNAs and transcription factors arranged in a comet-shaped network (Fig. 7a, Supplementary Fig. 8d). At the core of the network, we found factors related to senescence and Hippo signaling (SP1, SIN3B, YAP1), axon guidance (NRP1, GATA3, NRF1), mISR (ATF4), and the bioenergetic-associated metabolite L-carnitine (Fig. 7a). CHCHD2 and CHCHD10 were closely connected with this core network, indicating that these two proteins have strong interactions with each other within the biological system. Another member of the family, CHCHD1, was also found in close distance (Fig. 7a). Within the network, there were also other bioenergetic metabolites (Acetyl-CoA and lactate) and complex IV factors (COX6C).

qPCR analysis of members of the core network confirmed that Hippo signaling genes SIN3A, SIN3B and YAP1 were downregulated in 70Q/70Q NPCs compared to WT/WT NPCs, showing that engineered cells could recapitulate changes seen in patient-derived cells (Fig. 7b). The same downregulation occurred in 0Q/0Q NPCs compared to WT/WT NPCs, which suggested that removal of the CAG/CCG region may not be sufficient to revert these gene expression modifications in NPCs (Fig. 7b).

Overall, the multi-omics integration analysis of HD neurons highlighted the dysregulation of pathways associated with neurodevelopment, axon guidance, Hippo signaling, and carbon metabolism (Fig. 7c, Supplementary Fig. 8e). CHCHD2 was located close to the core of this integrated network, underscoring its relevance in the neuropathophysiology of mHTT.

## CHCHD2 modulation affects neurite growth and mitochondrial morpho-dynamics

Lastly, we aimed to investigate the impact of manipulating CHCHD2 expression levels in human neural cells. We first assessed the response of homogenous neuronal cultures to the repression of CHCHD2, motivated by data from cerebral organoids indicating that CHCHD2 reduction might be involved in the pathogenesis of HD. HTT is known to play a role in axonal morphogenesis and function[8,52], and our omics analysis in neurons indicated the disruption of axon guidance and neurodevelopment-related pathways. Therefore, we focused on the neurite outgrowth capacity of NGN2 neurons using a high-content microscopy-based platform (Fig. 7d)[53]. We employed a control engineered iPSC line in which we integrated the NGN2 cassette in the safe harbor locus, allowing rapid and robust NGN2 differentiation. Upon transfection with small interfering RNA (siRNA) constructs, we measured the dendritic and axonal outgrowth (Fig. 7d). In agreement with the known role of BDNF in neuronal branching[54], knock-down of BDNF in NGN2 neurons impaired the development of axonal and dendritic structures (Fig. 7e, Supplementary Fig. 9a). Similar defects in axonal and dendritic branching occurred in NGN2 neurons in which CHCHD2 or HTT were knocked down (Fig. 7e). These defects were stronger than

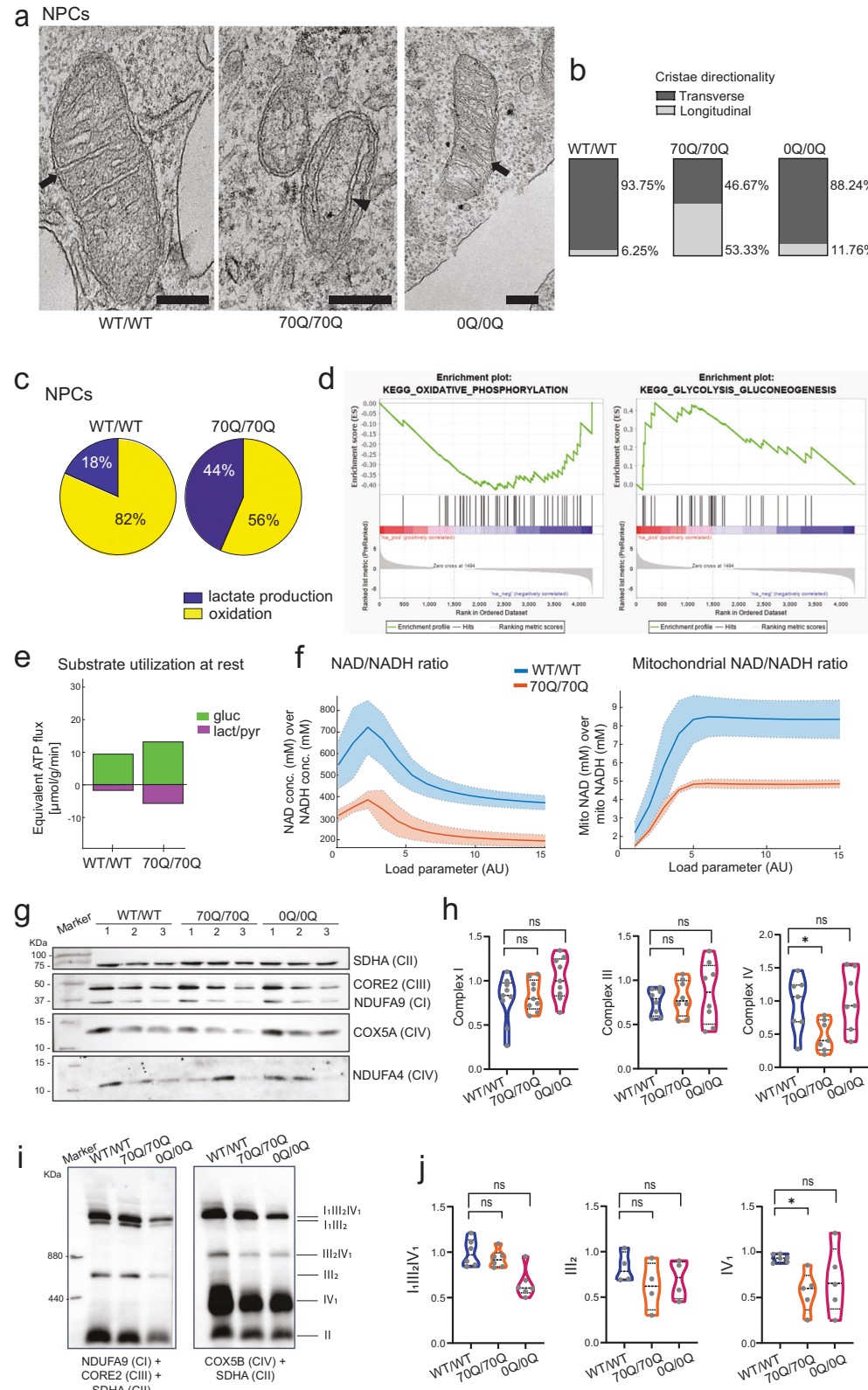

those observed in the case of VEGFA knock-down, a gene that has also been associated with neurite outgrowth (Supplementary Fig. 9a). The detrimental impact of WT HTT downregulation on neuronal morphogenesis underscores the need for cautious consideration in therapeutic approaches aiming at repressing HTT. In fact, all neural model systems carrying mHTT that we used (NPCs, cerebral organoids, neuruloids, and NGN2 neurons) showed a slight reduction in the

amount of WT HTT levels (Supplementary Fig. 9e–h, j), implying that HD phenotypes are potentially associated with reduced WT HTT.

To determine the consequences of CHCHD2 overexpression in mutant neural cells, we focused on 70Q/70Q NPCs and employed adeno-associated viruses (AAV) encoding for either CHCHD2-GFP or GFP alone. The expression signature of CHCHD2 and TOM20 in 70Q/70Q NPCs did not change upon transduction with GFP-AAV

**Fig. 5 | Neurometabolic defects induced by mHTT. a** Representative electron microscopy images of mitochondria within WT/WT NPCs, 70Q/70Q NPCs, and 0Q/0Q NPCs. Arrows indicate mitochondrial cristae with transverse direction with respect to mitochondrial outer membrane. Arrowhead indicates cristae with longitudinal direction. Data were repeated in two independent experiments. Scale bar: 200 nm. **b** Quantification of cristae direction in NPCs from WT/WT, 70Q/70Q and 0Q/0Q. For each sample, a minimum of 15 different mitochondrion were considered out of at least two biological replicates. **c** Proteomic-driven functional metabolic analysis (see Supplementary Data 7) depicting relative glucose utilization at resting energy demands in NPCs. **d** Gene Set Enrichment Analysis (GSEA) showing decreased oxidative phosphorylation (OXPHOS) and increased glycolysis/gluconeogenesis in 70Q/70Q NPCs compared to WT/WT NPCs. **e** Substrate utilization at rest in NPCs from WT/WT and 70Q/70Q. Energetic capacities were evaluated by computing the changes in metabolic state elicited by an increase of the ATP consumption rate above the resting value. mHTT-carrying cells showed higher consumption of glucose (gluc) and higher production of lactate-pyruvate (lact/pyr). **f** Metabolic state variables for NAD/NADH ratio and mitochondrial NAD/NADH ratio in NPCs in dependence of increasing energetic demands. Lines and colored areas represent mean ± s.d. **g, h** SDS-PAGE analysis and related quantification of mitochondrial complex I, III, and IV subunits in WT/WT NPCs, 70Q/70Q NPCs, and 0Q/0Q NPCs. $n = 3$ independent biological replicates per line run in three different blots; *$p < 0.05$, ns: not significant; unpaired two-tailed Welch $t$ test. **i, j** Blue native PAGE analysis and related quantification of complexes $III_2$ and IV, and supercomplexes $I_1III_2IV_{0-1}$ assembly in WT/WT NPCs, 70Q/70Q NPCs, and 0Q/0Q NPCs. $n = 3$ independent biological replicates per line run in two different blots; *$p < 0.05$, ns: not significant; unpaired two-tailed Welch $t$ test.

(Supplementary Fig. 9c), and the transduction of either CHCHD2-GFP-AAV or GFP-AAV led to similar levels of GFP signal (Supplementary Fig. 9d). As expected, CHCHD2 levels were significantly higher in 70Q/70Q NPCs transduced with CHCHD2-GFP-AAV compared to those transduced with only GFP-AAV (Fig. 7f, g). By focusing only on the GFP-positive cells, we then quantified the amount of TOM20, and found that CHCHD2 overexpression led to a significant decrease in TOM20 signal (Fig. 7f, g), thus reverting the diseased phenotype that we previously observed in cerebral organoids and NPCs carrying mHTT (Fig. 4g).

In conclusion, repression of CHCHD2 in human neurons adversely affected neurite branching capacity, while its overexpression in neural cells carrying mHTT led to amelioration of mitochondrial morpho-dynamics. These findings highlight the relevance of CHCHD2 in the neuronal and metabolic pathology of HD.

## Discussion

Increasing evidence from studies in mice and humans suggests that mutant HTT could impact the physiological development of the brain[4,5,19]. Our findings based on various human brain organoid differentiation paradigms demonstrate that mHTT disrupts the overall neurodevelopmental process and particularly the organization of neural progenitors. These results align with recent studies in mice and human fetuses showing that mHTT interferes with the cell cycle of apical progenitors, causing reduced proliferation and premature neuronal lineage specification[19].

By exploring mechanisms underlying defective human neurodevelopment caused by mHTT, we identified an early dysregulation of CHCHD2 expression. CHCH domain-containing proteins are imported into the mitochondrion and have been suggested to play a role in neurodegenerative diseases[32,33]. CHCHD2 contributes to the maintenance of mitochondrial morphology and dynamics, and its loss has been reported to disrupt mitochondrial organization in models of Parkinson's disease[55,56]. CHCHD2 expression was previously found to promote neural-ectodermal lineage differentiation of human iPSCs[57], indicating that its presence may be important during neurogenesis. Dysregulation of CHCHD2 has also been observed in neural cells carrying mHTT that were differentiated from human embryonic stem cells (hESCs) and iPSCs[58,59]. However, the role of CHCHD2 within the pathogenesis of HD remained elusive.

CHCHD2 is known to regulate mitochondrial dynamics and mitochondrial integrated stress response (mISR)[26]. mISR is a process that rises in stress conditions such as those associated with impaired mitochondrial function[35]. Accordingly, we identified a mISR signature in human neural cells carrying mHTT that was associated with aberrant mitochondrial morpho-dynamics and specific defects in mitochondrial complex IV assembly. Elimination of the poly-Q/poly-P region reverted the abnormal CHCHD2 expression and the associated mISR signature and mitochondrial defects, including those affecting cristae organization and complex IV assembly. Moreover, CHCHD2 over-expression

in mutant NPCs ameliorated mitochondrial morpho-dynamics. These findings suggest that mHTT leads to CHCHD2 dysregulation, which in turn impairs mitochondrial function and organization in early neural cells.

Dysfunction of mitochondrial activity and network morphology have been observed in multiple model systems of HD, including iPSC-derived medium spiny neurons[21,41,42,60–63]. Even Pridopidine, currently being investigated as a treatment for HD in Phase III study (NCT04556656), has been shown to repress mitochondrial stress and network defects[64]. However, again in agreement with a neurodegenerative view of the disease, mitochondrial dysfunction in HD models is usually thought to occur mainly within mature neuronal cells. Our data instead indicate that aberrant CHCHD2 expression and related mitochondrial morpho-function impairment can occur early during neurodevelopment.

Such early mitochondrial dysfunctions is expected to affect the overall process of neurodevelopment[65]. Impairment of the cellular NAD+/NADH redox balance caused by metabolic perturbation can in fact lower the global rate of protein synthesis, thereby slowing down the segmentation clock and overall developmental rate[66]. In particular, mitochondrial morphology and metabolism are emerging as important regulators of early neurogenesis[67–69]. Mitochondrial dynamics within NPCs regulate the pace of development[70], and help orchestrate the balance between proliferation and neuronal specification[71]. A correct metabolic programming towards OXPHOS starts to develop at the level of NPCs[46,47,72,73]. In our models, we found that both undifferentiated iPSCs and NPCs were able to respond to OXPHOS impairment by upregulating glycolytic metabolism. However, impaired NAD/NADH ratio became evident only in NPCs. Differentiated neurons could not compensate these defects and developed cell autonomous impairment of both OXPHOS and glycolysis. The establishment of neural fate might then pose greater pressure on cells that start relying more on OXPHOS, ultimately leading to metabolic stress and functional impairment. Hence, mHTT may interfere with the physiological metabolic programming that is required for enabling neural fate commitment.

Despite a disruption in neural progenitors, the generation of neurons (either appearing physiologically within brain organoids or forced through NGN2 overexpression) still occurred in the presence of mHTT. Indeed, we observed increased upregulation of genes related to neurodevelopment in mutant neural progenitor cultures forced to be converted into neurons. This is in agreement with premature neuronal generation observed in human fetuses carrying mHTT[19]. However, even if neurons can be generated, their functionality and resilience may be compromised. Cells carrying mHTT that prematurely commit to neural differentiation could thus produce neurons that are more susceptible to degeneration upon stress. Therefore, the loss of mechanisms protecting from stress and decompensation may play a role in the development of HD. Here, we found that CHCHD2 reduction in HD patient-derived neurons impaired neuronal branching capacity

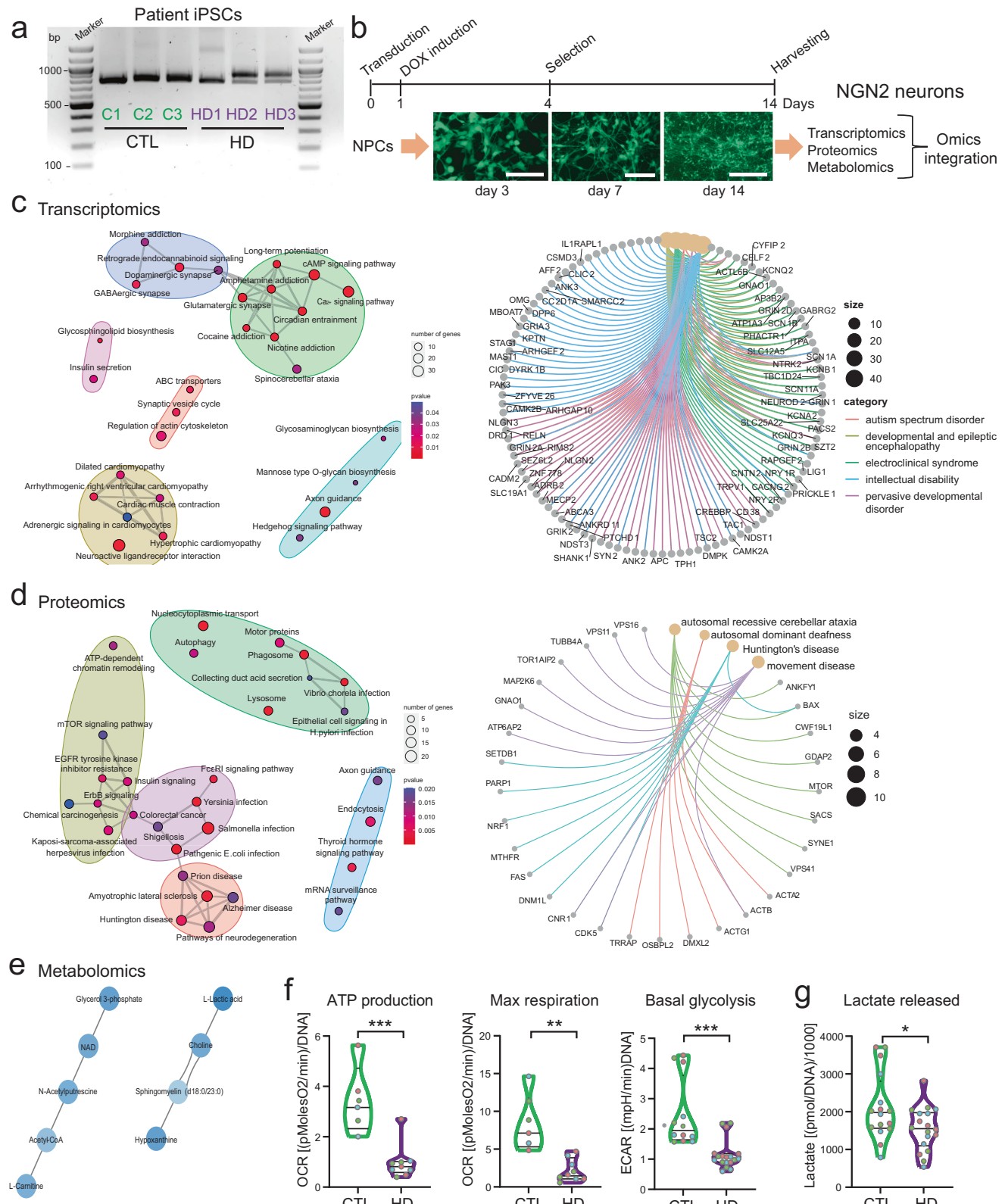

and its overexpression in mutant NPCs promoted a healthier mitochondrial state. In accordance, recent findings point towards a protective role for CHCHD2 up-regulation in HD as a protection response against oxidative stress[74].

These results are important for our understanding of the pathomechanisms of HD, as the prevalent view of the disease is that it develops as a degeneration of mature neurons[75]. If brain defects

caused by mHTT instead develop early in life, therapeutic strategies should be administered during development. It is thus possible that therapeutics based on lowering mHTT, as performed in the phase III clinical trial (NCT03761849) developed by Roche[76], might not be as effective as hoped for. If such administration is given during adult life, the brain circuity has already been altered and may not be effectively repaired anymore. Accordingly, we found that defects in neural

**Fig. 6 | Multi-omics signature of NGN2 neurons from individuals with HD underscores developmental and metabolic defects. a** PCR analysis of HTT in NPCs derived from three healthy controls (C1, C2, C3) and three individuals with HD (HD1, HD2, HD3, carrying WT/180Q, WT/58Q, and WT/44Q, respectively). Data were repeated in three independent experiments. **b** Schematics of Neurogenin 2 (NGN2)-based neuronal induction in NPCs from healthy controls (C1, C2, C3) and HD individuals (HD1, HD2, HD3). Scale bar: 100 μm. **c** Enrichment analysis of differentially expressed genes (DEGs) in NGN2 neurons from HD individuals compared to controls, hypergeometric test one-sided with FDR adjustment for multiple comparisons (filter for $p$ value < =0.05). Left: over-representation analysis (ORA) of upregulated DEGs for KEGG pathways; right: ORA of upregulated DEGs for human diseases. Yellow nodes represent the enriched diseases and gray nodes their associated genes, with colored edges indicating their connection (see Supplementary Data 7). **d** Enrichment analysis of differentially expressed proteins (DEPs)

in NGN2 neurons from HD individuals compared to controls, hypergeometric test one-sided with FDR adjustment for multiple comparisons (filter for $p$-value < =0.05). Left: ORA of upregulated DEPs for KEGG pathways; right: ORA of upregulated DEPs for human diseases (see Supplementary Data 8, 9). **e** Enrichment analysis of differentially expressed metabolites (DEMs) in NGN2 neurons from HD individuals compared to controls (see Supplementary Data 10). Interactions between the significantly dysregulated metabolites. **f** Mitochondrial bioenergetics based on oxygen consumption rate (OCR) and glycolysis based on extracellular acidification rate (ECAR) measured by Seahorse profiling in NGN2 neurons from controls (C1, C2, C3) and HD individuals (HD1, HD2, HD3). $n = 3$ independent biological replicates per line (different colors of dots refer to the three replicates). $*p < 0.05$, $**p < 0.01$, $***p < 0.001$; unpaired two-tailed $t$ test. **g** Quantification of lactate released in the supernatant by NGN2 neurons at the end of Seahorse experiments. $*p < 0.05$; unpaired two-tailed $t$ test.

progenitors and brain organoids occurred at a time point where toxic mHTT aggregates have not yet developed. Smaller intracranial volumes can be observed early in pre-manifest HD mutation carriers[14–16]. These findings raise the possibility that HD pathogenesis may also involve a loss of function mechanism, caused by the reduction of WT HTT. Indeed, WT HTT has been proposed to play a role in neurodevelopment, where it could regulate mitotic spindle orientation thereby determining cortical progenitor fate[12], and participate in axonal transport modulation[77]. In agreement with this, we found that downregulation of WT HTT was sufficient to impair neuronal branching capacity.

Individuals with HD often suffer from weight loss and cachexia irrespective of food intake[78–80]. In fact, the metabolic network impairment has been suggested as a progression biomarker of pre-manifest HD[81]. We found that mHTT caused a metabolic state indicative of elevated energy consumption at rest. This feature, also known as hypermetabolism, leads to weight loss, and is associated with greater functional decline in cancer patients or individuals with amyotrophic lateral sclerosis[82]. Hypermetabolism has recently been found to occur in the presence of OXPHOS-impairing mutations that can lead to mISR activation[35]. Previously, we and others have demonstrated that OXPHOS-impairing mutations cause aberrant neuronal morphogenesis and defective brain organoid development[83,84]. Hence, it is possible that CHCHD2 could represent a target for early intervention in HD, as its modulation would impact mitochondrial morpho-function, which in turn could be important for morphogenesis and neurodevelopment on the one hand and for overall systemic metabolic resilience on the other.

Our integrative multi-omics analysis of HD patient-derived neurons indicated that CHCHD2 belonged to a core dysregulated network composed of genes/proteins/metabolites associated with neurodevelopment, axon guidance, Hippo signaling, carbon metabolism, and mISR. Among these factors, SIN3A and YAP1 may be particularly interesting, given that SIN3A function counteracted neurodegeneration in a drosophila model of HD[85], and YAP1 was found dysregulated in HD mice and in post-mortem patient brain[79]. In fact, YAP activators ameliorated symptoms and disease phenotypes in different HD models[86,87], and modulation of YAP activity rescued key HD-related neurodevelopmental phenotypes in neuruloids carrying mHTT[88]. Hence, by targeting CHCHD2 expression, other associated factors may possibly also be positively influenced. Future studies should address whether over-expression of these factors might lead to ameliorate the neuronal defects caused by mHTT. It was previously found that polyQ inclusion sequester RNA molecules in an overexpression model of polyQ-ATXN1[89]. It is tempting to speculate that something similar might also be occurring in other polyQ diseases such as HD, where polyQ inclusions could sequester RNAs or transcription factors causing their impairment.

It is important to note that our study primarily examined cells carrying mHTT on both alleles, a condition in which potential compensatory effects of WT HTT are absent. Hence, additional research would be required to further dissect the role in CHCHD2 in different HTT genotypes, given that homozygous and heterozygous forms are not clearly different in HD patients[90,91]. At the same time, it was reassuring to see that NGN2 neurons from HD patients carrying WT/180Q, WT/58Q, and WT/44Q recapitulated similar defects regardless of the poly-Q length. Indeed, a recent study using cerebral organoids obtained from hESCs carrying WT/45Q, WT/65Q and WT/81Q described similar features indicative of neurodevelopment impairment irrespective of the polyQ length and the homozygous or heterozygous state[24]. The consequences of CAG/CCG elimination also need to be examined in more details. 0Q/0Q cells showed rescue of mitochondrial phenotypes but the expression for genes associated with Hippo signaling was not ameliorated. Furthermore, 0Q/0Q NPCs exhibited changes in mitochondrial ultrastructure that were not seen in 70Q/70Q NPCs. The biological meaning of these changes remains unclear. Further investigations are also warranted to address the impact of somatic deletion of the polyQ region directly in neural cells, in contrast to applied as in-frame deletion in iPSCs. Lastly, as CHCHD2 belongs to the high-confidence interactome map of neurodegenerative-associated human proteins[92], it will be important to be determined whether it could represent a direct binding partner of mHTT.

## Methods
### Human iPSC lines
Our research complies with all relevant ethical regulations. Ethical approval for using iPSCs from HD individuals and control subjects was obtained from the Ethic Committee of the University Clinic Düsseldorf (study number 2019-681 approved on October 11, 2019).

WT/WT iPSCs for genome editing were derived using Sendai viruses from a healthy middle-aged male individual (BIHi050-A/SCVI113). iPSCs from three male individuals with HD were described before: HD1 with WT/180Q (BIHi035-A)[49], HD2 with WT/58Q (BIHi288-A)[50], and HD3 with WT/44Q (BIHi033-A)[50]. Control male iPSCs used for comparison with HD male iPSCs were derived before: C1 (TFBJ, HHUUKDi009-A)[46], C2 (XM001, BIHi043-A)[93], and C3 (BIHi005-A). For engineering an inducible NGN2 iPSC line, we used the control healthy male iPSC line C3 (BIHi005-A). All iPSCs were cultured feeder-free on Matrigel-coated (Corning, USA) 6-well plates (Greiner Bio-One, Austria) in iPS-Brew consisting of StemMACS iPS-Brew XF basal medium supplemented with 1:50 StemMACS iPS-Brew XF supplement (both Miltenyi Biotec, Germany) and 1:500 MycoZap-Plus-CL (Lonza, Switzerland). iPSC cultures were kept in a humidified atmosphere of 5% $CO_2$ and 5% oxygen at 37 °C. We routinely monitored all cultures against mycoplasma contamination using PCR. Cells were passaged at

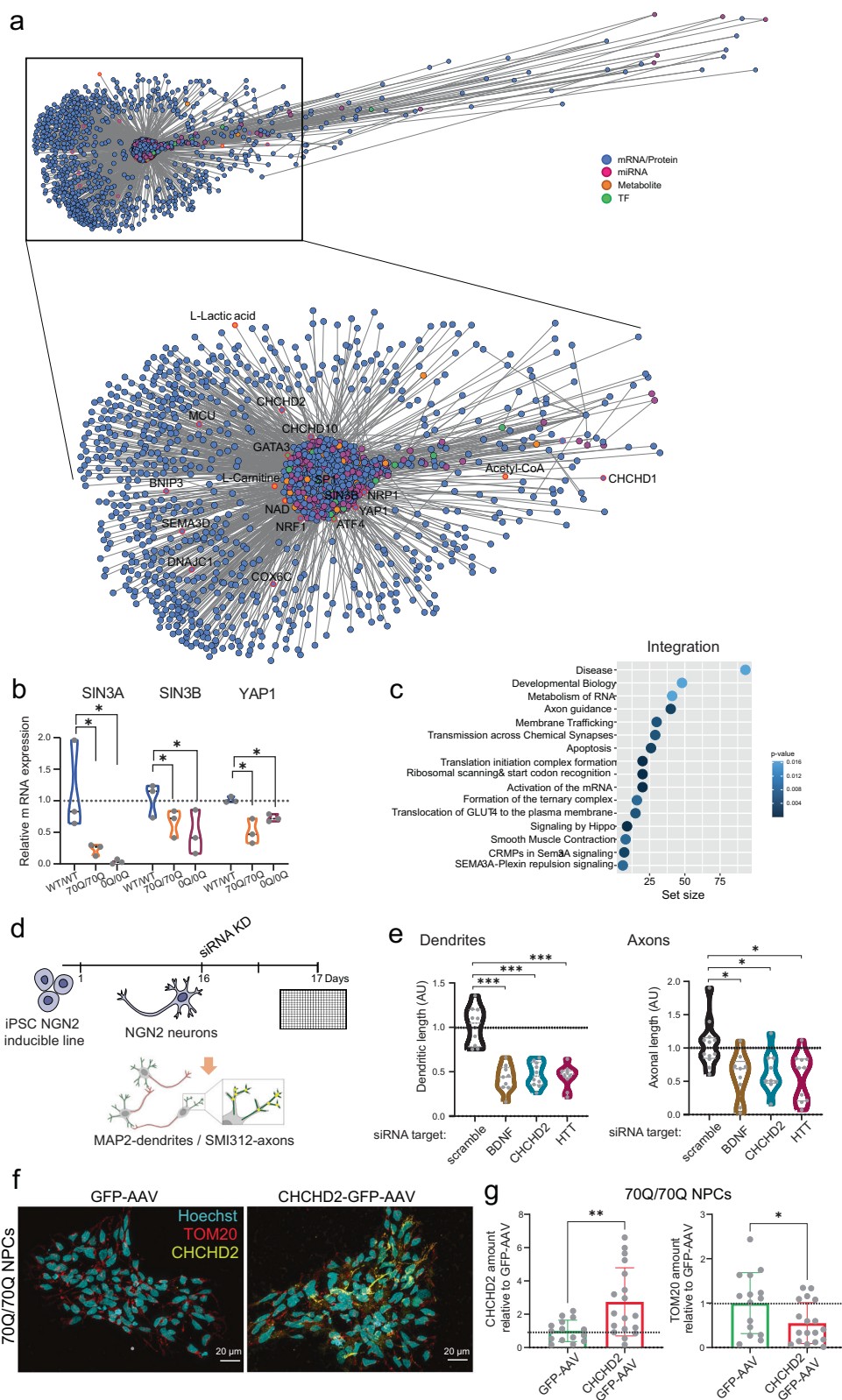

70–80% confluence with 0.5 µM EDTA (Invitrogen, USA) in 1xPBS (Gibco, USA). 10 µM Rock inhibitor (Enzo Biochem, USA) was added after splitting to promote survival.

### Engineering huntingtin in human iPSCs
Isogenic clones with a homozygous (HTT WT/70Q) and heterozygous (HTT 70Q/70Q) allelic combinations carrying 70Q insert were engineered from control (WT/WT) BIH-SCVI113 iPSC line with healthy background. We used clustered regularly interspaced short palindromic repeats (CRISPR) and high-fidelity Cas9 (eCas9), which has limited off-target effects[27]. We used CRISPR/Cas9 induced homology directed repair (HDR) with 2354 nts dsDNA donor template encoding patient reference sequence carrying 70Q consisting of both CAG and CAA triplets including 3' and 5' homology arms. Constructs were

**Fig. 7 | CHCHD2 belongs to the multi-omics network of HD neurons and its manipulation affects neurite growth and mitochondrial morpho-dynamics.**
**a** 3D multi-omics network. Significantly dysregulated mRNAs, proteins, and metabolites along with their predicted miRNAs and transcription factors (TFs) are arranged in a comet-shaped network. CHCHD2 and CHCHD10 are located closely to the core of the network. **b** qPCR analysis of components of the core network related to Hippo signaling in NPCs. Expression level of each gene was related to that of housekeeping genes ACTB and OAZ1. Mean ± s.e.m.; $n = 3$ independent biological replicates (dots) per line; *$p < 0.05$; unpaired two-tailed Welch $t$ test. **c** Enrichment analysis for Reactome biological pathways affected by the dysregulated mRNAs/proteins of the multi-omics network; hypergeometric test one-sided with FDR adjustment for multiple comparisons (filter for adj. $p$ value < =0.05). **d** Schematics

of small interfering RNA (siRNA)-mediated knockdown (KD) in NGN2 neurons from NGN2-inducible control iPSCs. Quantification of neuronal arborization assessed by high-content imaging based on antibodies labeling axons (SMI132) and dendrites (MAP2). **e** Quantification of branching outgrowth in NGN2 neurons following siRNA KD. Dot indicates different replicates out of three independent experiments. *$p < 0.05$, ***$p < 0.001$ compared to scramble siRNA KD; Kruskal-Wallis test with Dunn's multiple comparisons test. **f** Representative images of 70Q/70Q NPCs transduced with either GFP-AAV or CHCHD2-GFP-AAV. Scale bar: 20 μm. **g** Quantification of CHCHD2 and TOM20 amount in GFP-positive 70Q/70Q NPCs transduced with either GFP-AAV or CHCHD2-GFP-AAV. Data shown as mean ± s.d. Dots represent individual images collected from two independent experiments. represent **$p < 0.01$, *$p < 0.05$; unpaired two-tailed Welch $t$ test.

commercially synthetized as a synthetic HTT donor construct and assembled from synthetic oligonucleotides by GeneArt Gene Synthesis service (Thermo Fisher Scientific, USA). Silent mutations within single guide RNAs (sgRNAs) were introduced in proximity to each of the two sgRNA protospacer adjacent motif (PAM) sites (3 and 6 nt downstream) to prevent recurrent Cas9 cutting in edited cells. Synthesized template was inserted into pMK-RQ (kanR)/pMA-RQ (ampR) plasmid, purified from transformed bacteria, determined by spectroscopy and verified by sequencing. sgRNAs (sgHTT#1, sgHTT#2) targeting upstream and downstream of the HTT polyQ region were designed using CRISPOR (Supplementary Fig. 1a, Supplementary Data 11). The oligomer pairs were annealed and cloned separately into two pU6-CAG-eCas9-Venus plasmids carrying eSpCas9 variant from eSpCas9(1.1) plasmid (Addgene ID 71814) to reduce off-target effects and improve on-target cleavage[27]. To upregulate homology directed repair (HDR) of eCas9 induced double-strand breaks with dsDNA donor template, we applied ectopic expression of BRCA1 and BCL-XL. To downregulate the non-homologous end joining (NHEJ) pathway we applied dominant-negative sub-fragment of 53BP1 (dn53BP1), which counteracts endogenous 53BP1[28]. Plasmids encoding components of the DNA repair pathways (human BRCA1, BCL-XL and mouse dn53BP1) were kindly obtained from Bruna Paulsen (dn53BP1) and Xiao-Bing Zhang (BCL-XL). For generation of dn53BP1, a fragment containing the tudor domain (residues 1,221 to 1,718 of mouse 53BP1) was amplified and sub-cloned into a CAG expression plasmid and sequenced. For the generation of BCL-XL, a fragment containing the BCL-XL was amplified and sub-cloned into a pEF1-BFP expression plasmid and sequenced. pU6-sgHTT#1-CAG-eCas9-Venus and pU6-sgHTT#2-CAG-eCas9-Venus plasmids targeting upstream and downstream of the CAG/CCG stretches of the HTT gene, plasmids encoding 70Q HDR template and BRCA1, BCL-XL and mouse dn53BP1 were transformed into super competent DHA bacteria strain using heat shock and cloned. Plasmid DNA was extracted using the PureYield plasmid miniprep system (Promega) and transfected into the control iPSC line. In addition to the BIH-SCVI113 iPSCs with introduced 70Q templates, we also generated clones harboring HTT alleles using a strategy based on NHEJ mediated excision of the poly Q/P repeats region leading to in-frame reannealing of the DSB. By transfection of pU6-sgHTT#1-CAG-eCas9-Venus and pU6-sgHTT#2-CAG-eCas9-Venus with BCL-XL plasmids, we generated DSBs which upon re-ligation with the lack of HDR template resulted into an in-frame HTT coding region, lacking the N-terminal Q/P repeats. Transient transfection of plasmids was carried out in BIH-SCVI113 iPSC line grown in feeder-free conditions in StemMACS™ iPS-Brew XF culture media (Miltenyi Biotec, Germany) in a 6-well cell culture plate. One day prior to transfection, we dissociated the cells using Accutase (Sigma-Aldrich, USA) and seeded ~ $1 \times 10^5$ cells per well of a pre-coated 6-well plate as single cells or small clumps. Cells were cultivated in fresh medium containing 10 mM Rock inhibitor (Enzo Biochem, USA) overnight. Transfection was performed using Lipofectamine™ 3000 Transfection Reagent (Thermo Fisher Scientific, USA), according to the manufacturer's protocol. The plasmids were

diluted up to 2 mg DNA in 125 ml of Opti-MEM reduced serum medium and added as the DNA-lipid complex to one well of a 6-well plate in a dropwise manner with addition of 5 mM Rock inhibitor to the culture medium for 24 h. Medium was changed on the following day and the cells were kept 48 h in culture until fluorescence-activated cell sorting (FACS). Dissociated cells using Accutase for 5 min were washed and resuspended with PBS. Then, cells were filtered using Falcon polystyrene test tubes (Corning, USA) and transferred to new Falcon polypropylene test tubes. Two colors FACS enrichment was performed using BD FACSAria III at the MDC FACS Facility for cells expressing high levels of eCas9-sgRNAs, and BCL-XL (Venus, BFP). Sorted cells were suspended in recovery mTeSR™ medium (StemCell Technologies, Canada) with 1X Penicillin-Streptomycin (P/S) (Gemini Bio-products) and ROCK inhibitor and plated in low concentrations onto 6-well plates for the establishment of single cell derived colonies (5000 cells/well). Growing single cell-derived colonies were transferred from 6-well plates to one well each of 24-well plate and maintained until the colony grew big enough to be partially harvested for DNA isolation using Phire Animal Tissue Direct PCR Kit (Thermo Fisher Scientific, USA) according to manufacturer's protocol. PCR reaction was carried out using 10 ng gDNA in 25 ml with chemical hot-Start AmpliTaq Gold DNA Polymerase (Thermo Fisher Scientific, USA) with an annealing temperature of 55 °C and GC enhancer reagent (Thermo Fisher Scientific, USA). For Sanger sequencing, the PCR products were gel purified using the Wizard SV Gel and PCR Clean-Up System (Promega) and cloned into pJet cloning vectors using the CloneJET PCR Cloning Kit (Thermo Fisher Scientific, USA). Cloned PCR products were submitted to LGC for Sanger sequencing. Karyotype analysis was performed by MDC Stem Cell Core Facility. Briefly, DNA was isolated using the DNeasy blood and tissue kit (Qiagen, USA). SNP karyotyping was assessed using the Infinium OmniExpressExome-8 Kit and the iScan system from Illumina. CNV and SNP visualization was performed using KaryoStudio (v. 1.4) (Illumina, USA). Primer sequences, gRNA sequences, and HDR sequences are reported in Supplementary Data 11.

## Derivation of neural progenitor cells (NPCs)

To generate neural progenitor cells (NPCs) from iPSCs, we applied our published protocol[94]. Briefly, iPSCs were detached from Matrigel-coated plates using Accutase (Sigma-Aldrich, USA) and the collected sedimented cells were transferred into low-attachment petri dishes and kept for two days in M1 medium (1x KnockOut DMEM, 1x KnockOut serum, 0.1 mg/ml PenStrep, 2 mM Glutamine, 1x NEAA, 1 mM Pyruvate [all Gibco, USA], 1x MycoZap-Plus-CL [Lonza, Switzerland], 3 μM CHIR 99021 [Caymen Chemical, USA], 10 μM SB-431542 [Caymen Chemical, USA], 1 μM Dorsomorphin [Sigma-Aldrich, USA], 500 nM Purmorphamine [Miltenyi Biotec, Germany]). From day 2 to day 6, the media was switched to M2 medium (50% DMEM-F12, 50% Neurobasal, 0.5× N-2 supplement, 0.5x B-27 supplement without Vitamin A, 0.1 mg/ml PenStrep, 2 mM Glutamine [all Gibco, USA], 1x MycoZap-Plus-CL [Lonza, Switzerland], 3 μM CHIR 99021 [Caymen Chemical, USA], 10 μM SB-431542 [Caymen Chemical, USA],

1 µM Dorsomorphin [Sigma-Aldrich, USA], 500 nM Purmorphamine [Miltenyi Biotec, Germany]). On day 6, the suspended cells were transferred onto Matrigel-coated (Corning, USA) 6-well plates (Greiner Bio-One, Austria) using sm+ medium (50% DMEM-F12, 50% Neurobasal, 0.5x N-2 supplement, 0.5x B-27 supplement without Vitamin A, 0.1 mg/ml PenStrep, 2 mM Glutamine [all Gibco, USA], 1x MycoZap-Plus-CL [Lonza, Switzerland], 3 µM CHIR 99021 [Caymen Chemical, USA], 500 nM Purmorphamine [Miltenyi Biotec, Germany], 150 µM ascorbic acid [Sigma-Aldrich, USA]). NPCs were maintained in this media without rock inhibitor and used for experiments between passage 7 and 20.

## Generation of unguided cerebral organoids

Unguided cerebral organoids were generated from WT/WT and 70Q/70Q iPSCs using a modified version of a published protocol[95]. At day 0, iPSCs from one 80% confluent well were washed with 1x PBS (Gibco, USA) and treated with Accutase (Sigma-Aldrich, USA) for 3 min at 37 °C. Subsequently, cells were collected in 5 ml iPS-Brew and spun down for 5 min at 270 × g. After removing the supernatant, cells were resuspended in 1 ml embryoid body (EB) media (1x DMEM-F12, 1:5 KnockOut-Serum, 1x NEAA, 0.1 mg/ml PenStrep, 1x GlutaMAX [all Gibco, USA], 7 nM 2-mercaptoethanol [Merck, Germany], 4 ng/ml bFGF [PeptroTech, USA], 10 µM Rock inhibitor [Enzo Biochem, USA]). Cells were counted manually, and 9000 cells were plated in 150 µl EB media in each well of a round bottom, ultra-low attachment 96-well plate (Corning, USA). Next, the plate was spun down at 500 rpm for 2 min to assure the aggregation of cells in the center of the wells. On day 2 and day 4, 50% of the EB media was replaced with EB media without ROCK inhibitor and bFGF. At day 5, each EB was transferred to a well of a 24-well plate (Greiner bio-one, USA) in 250 µl neural induction media (IM) (1x Neurobasal, 1x N-2 supplement, 1x NEAA, 1x GlutaMax, 0.1 mg/ml PenStrep [all Gibco, USA], 1 µg/ml heparin [Sigma-Aldrich, USA]). On day 6 and 10, another 250 µl IM was added to each well. At day 11, organoids were embedded in droplets of 30 µl Matrigel (Corning, USA)[95], and 16 organoids each were transferred to 60×15 mm cell culture dishes (Greiner bio-one, USA) and cultured for 48 h in 5 ml IM at 37 °C with 5% CO$_2$. After this, IM was replaced with neuronal differentiation media 1 (DM1) (50% DMEM-F12, 50% Neurobasal, 0.5x N-2 supplement, 2x B-27 supplement without vitamin A, 0.1 mg/ml PenStrep, 1x GlutaMax, 0.5x NEAA [all Gibco, USA], 0.35 nM 2-mercaptoethanol [Merck, Germany], 2.5 µM insulin [Sigma-Aldrich, USA], 3 µM CHIR99021 [Caymen Chemical, Germany]). After 2 days, the media was replaced by fresh DM1 and at day 17, the organoids were placed on an orbital shaker (set to 85 rpm) and the media was replaced by DM2 (50% DMEM-F12, 50% Neurobasal, 0.5x N-2 supplement, 1x B-27 supplement, 0.1 mg/ml PenStrep, 1x GlutaMax, 0.5x NEAA, 1:100 chemically defined lipid concentrate [all Gibco, USA], 0.35 nM, 2-mercaptoethanol [Merck, Germany], 2.5 µM insulin [Sigma-Aldrich, USA]). After 3 days, with a full media refreshment after 2 days, the media was replaced with DM3 (50% DMEM/F12, 50% Neurobasal, 0.5x N-2 supplement, 1x B-27 supplement, 0.1 mg/ml PenStrep, 1x GlutaMax, 0.5x NEAA, 1:100 chemically defined lipid concentrate [all Gibco, USA], 0.35 nM 2-mercaptoethanol [Merck, Germany], 2.5 µM insulin [Sigma-Aldrich, USA], 0.4 µM ascorbic acid [Sigma-Aldrich, USA], 12 mM HEPES solution [Biochrom, Germany]). From this point on, full media replacements took place every 3-4 days. At day 40, the media was changed to maturation media (MM) (1x Neurobasal, 1x B-27 supplement, 0.1 mg/ml PenStrep [all Gibco, USA], 0.35 nM 2-mercaptoethanol [Merck, Germany], 0.4 µM ascorbic acid [Sigma-Aldrich, USA], 20 ng/ml BDNF [Miltenyi Biotec, Germany], 20 ng/ml GDNF [Miltenyi Biotec, Germany], 0.5 mM cAMP [Sigma-Aldrich, USA], 12 mM HEPES solution [Biochrom, Germany]). Media was checked for mycoplasma routinely. Organoid size was measured using ImageJ (v. 1.53a).

## Generation of guided region-specific brain organoids

Cortical organoids were generated following a protocol previously published[96] with minor modifications. On day 0, iPSCs at 80% confluence were incubated with Accutase (Sigma-Aldrich, USA) at 37 °C for 7 min and dissociated into single cells. For aggregation into spheroids, approximately $3 × 10^6$ single cells were seeded per AggreWell-800 well (StemCell Technologies, Canada) in StemMACS iPS Brew XF (Miltenyi Biotec, Germany) supplemented with 10 µM ROCK inhibitor (Enzo Biochem, USA), centrifuged at 100 × g for 3 min, and then incubated at 37 °C in 5 % CO$_2$. After 24 h, spheroids consisting of approximately 10,000 cells were collected from each microwell by gently pipetting the medium up and down with a cut P1000 pipet tip and transferred into a petri dish coated with Anti-adherence solution (StemCell Technologies, Canada). From day 1-6, ES Medium (KnockOut DMEM, 20% Knockout Serum Replacement, 1x NEAA, 1x Sodium Pyruvate, 1x GlutaMAX [all Gibco, USA], 1x MycoZap-Plus-CL [Lonza, Switzerland], 2.5 µM dorsomorphin [Sigma-Aldrich, USA], 10 µM SB-431542 [Caymen Chemical, USA]) was changed daily. From day 7-22, neural medium (Neurobasal-A, B-27 Supplement without Vitamin A, 1x Glutamax [all Gibco, USA], 1x MycoZap-Plus-CL [Lonza, Switzerland]) supplemented with 20 ng/ml EGF (R&D Systems, USA), and 20 ng/ml FGF2 (R&D Systems, USA). From day 22, the neural medium was supplemented with 20 ng/ml BDNF (Miltenyi Biotec, Germany), 20 ng/ml NT-3 (PeproTech, USA), 200 µM ascorbic acid (Sigma-Aldrich, USA), 50 µM db c-AMP (StemCell Technologies, Canada) and 10 µM DHA (MilliporeSigma, USA). From day 46, only neural medium containing B-27 Plus Supplement (Gibco, USA) was used for media changes every 2–3 days.

NPC-derived midbrain organoids were generated following a protocol previously published[30] with some modifications. On day 0, we detached NPCs using Accutase (Sigma-Aldrich, USA) to obtain a single-cell suspension. Using an automatic cell counter (Cytosmart, Netherlands) we gated the cell size 5-19 µm and seeded 9000 cells per well onto a low-attachment U-bottom 96-well plate (FaCellitate, Germany) in 150 µL of basal media (50% DMEM-F12, 50% Neurobasal, 0.5x N2 supplement, 0.5 x B-27 supplement without Vitamin A, 1x GlutaMAX [all Gibco, USA], 1x MycoZap-Plus-CL [Lonza, Switzerland]) supplemented with 0.5 µM PMA (Miltenyi Biotec, Germany), 3 µM CHIR 99021 (Sigma- Aldrich, USA), and 100 µM ascorbic acid (Sigma-Aldrich, USA). After 2 days, we started ventral patterning for 4 days (in two feedings) by supplementing the basal media with 100 µM ascorbic acid, 1 µM PMA, 1 ng/ml BDNF (Miltenyi Biotec, Germany) and 1 ng/ml GDNF (R&D Systems, USA). On day 6 we switched to maturation media by supplementing the basal media with 100 µM ascorbic acid, 2 ng/ml BDNF, 2 ng/ml GDNF, 1 ng/ml TGF-β3 (StemCell Technologies, Canada), 100 µM db c-AMP (StemCell Technologies, Canada) and the addition of 5 ng/ml Activin A (StemCell Technologies, Canada). From day 6 on, we refreshed the maturation media (without Activin A, which is exclusively added on day 6) 3 times a week. To measure organoid size, we first took pictures of the organoids every other day with a Nikon Eclipse Ts2 inverted routine microscope using a 4x objective. We measured the area and perimeter from at least 5 organoids per cell line for each time point. We analyzed the images with ImageJ (v. 1.53a) by manually drawing a circle around the organoid[83].

## Engineering of inducible NGN2 iPSCs

We used the healthy iPSC line C3 (BIHi005-A) to engineer a doxycycline inducible Neurogenin 2 (NGN2) expression cassette to obtain the line BIHi005-A-24. To generate this cell line, we used the strategy of TALEN based gene-editing. Each AAVS1 locus was changed differently; on one AAVS1 allele we inserted the constitutive expression cassette of the reverse trans activator domain (m2rtTA) and on the other one the mNgn2-P2A-GFP-T2A-Puromycine cDNA sequence under the tetracycline responsive element (TRE). To generate the AAVS1-TRE-mNgn2-P2A-GFP-T2A-Puromycine donor plasmid, we amplified the TRE-

mNgn2-P2A-GFP-T2A-Puromycine sequence from LV-NEP (YS-TetO-FUW-Ng2-P2A-EGFP-T2A-Puro) plasmid from Dr. Thomas Sudhof lab[51], and cloned it into the plasmid AAVS1-iCAG-copGFP (Addgene 66577). To make the lines inducible for doxycycline, the reverse TET trans activator (m2rtTA) was inserted on the other allele of the AAVS1 locus using the m2rtTA plasmid containing a Neomycin resistance gene (addgene 60843). The TALEN plasmids to target AAVS1 locus were used from addgene: hAAVS1-TAL-L (35431) and hAAVS1-TAL-R (35432). BIHi005-A iPSC cells were transfected with TALEN plasmids targeting AAVS1 locus and donor plasmids (AAVS1-TRE-mNgn2-P2A-GFP-T2A-Puro and AAVS1-NEO-M2rtTA). Then we selected the clones which were resistant for both puromycin and neomycin antibiotics. Next, we derived single-cell clones as described[97], and selected the clones that express GFP and differentiated to neurons upon exposure to 3 μg/ml doxycycline. The selected clones showed typical pluripotent stem cell morphology, expressed pluripotent markers, and showed normal karyotype.

### Derivation of NGN2 neurons

For generation of NGN2 neurons from the inducible NGN2 iPSC line BIHi005-A-24, $3 \times 10^5$ iPSCs per well were seeded in a Geltrex (Gibco, USA) coated 6-well plate (Greiner Bio-One, Austria) using StemMACS iPS-Brew medium supplemented with 10 μM Rock inhibitor (Enzo Biochem, USA). On day 0 and day 1, we added induction medium (DMEM-F12, 1x N2 supplement, 1x NEAA, 1 mg/ml Pen/Strep [all Gibco, USA], 10 ng/ml human NT-3 [PeproTech, USA], 10 ng/ml human BDNF [Miltenyi Biotec, Germany], 0.2 μg/ml murine laminin [Sigma-Aldrich, USA]) freshly supplemented with 3 μg/ml doxycycline (Sigma-Aldrich, USA). On day 2, medium was changed to neuronal medium (Neurobasal, 1x B-27 supplement, 1x GlutaMAX, 1 mg/ml Pen/Strep [all Gibco, USA], 1x MycoZapTM Plus-CL (Lonza, Switzerland), 10 ng/ml human NT-3 [PeproTech, USA], 10 ng/ml human BDNF [Miltenyi Biotec, Germany], 0.2 μg/ml murine laminin [Sigma-Aldrich, USA]) freshly supplemented with 3 μg/ml doxycycline (Sigma-Aldrich, USA). Medium was exchanged daily until day 4. From day 6 on, half of the medium was replaced every other day with neuronal medium freshly supplemented with 3 μg/ml doxycycline (Sigma-Aldrich, USA) and 2 μM AraC (Sigma-Aldrich, USA) until the cells were harvested for subsequent experiments.

For NGN2 neurons derived from healthy iPSCs and HD iPSCs, we employed NPCs derived from control iPSCs (C1, C2, C3) and from HD patient-derived iPSCs (H1, H2, H3). The protocol was modified from a published protocol[51]. For NGN2 virus production, HEK 293 cells were seeded at 70% confluency in DMEM medium in a 150 cm² dish. After cell attachment, the medium was replaced and supplemented with 25 μM Chloroquine and the cells were transfected according to the manufacturer's instructions using Lipofectamine 2000 (Thermo Fisher Scientific, USA) and the following plasmid mix: 8.1 μg pMD2.G, 12.2 μg pMDLg-pRRE, 5.4 μg pRSV-Rev, 5.4 μg FUW-M2-rtTA, 14.3 μg TetO-FUW-NGN2, 14.3 μg TetO-FUW-EGFP. After 24 h, the medium was replaced with DMEM-F12 (Gibco, USA) and 10 μM sodium butyrate (Sigma-Aldrich, USA). Virus-containing supernatant was collected 24 h and 48 h later. To concentrate the virus particles, the supernatant was first centrifuged at $500 \times g$ for 10 min to remove cells and debris and then mixed with 1 volume of cold Lenti-X concentrator (Takara Bio, Japan) to every 3 volumes of lentivirus-containing supernatant and kept overnight at 4 °C. Afterwards, the mixture was centrifuged at $1500 \times g$ for 45 min at 4 °C, and the resulting pellet was resuspended in PBS (Gibco, USA), 1:100 of the original volume. Virus aliquots were stored at −80 °C until further use. For NGN2 neuron generation and culture, $2.5 \times 10^6$ NPCs per well were seeded on a Matrigel-coated (Corning, USA) 6-well plate (Greiner Bio-One, USA) using sm+ medium (see NPC generation for media composition). After cells attached (2–24 h), the medium was replaced and supplemented with 4 μg/ml polybrene and $2.25 \times 10^6$ transducing units of both EGFP and NGN2

lentivirus. On the next day (day 0), the cells were washed 3 times with PBS and 3 volumes sm+ medium mixed with 1 volume NGN2 medium (Neurobasal, 1x B-27 supplement, 1x NEAA, 2 mM Glutamine, 0.1 mg/ml Pen/Strep [all Gibco, USA], 1x MycoZap-Plus-CL [Lonza, Switzerland], 10 ng/ml human NT-3 [PeproTech, USA], 10 ng/ml human BDNF [R&D Systems, USA]) supplemented with 2 μg/ml doxycycline (Sigma-Aldrich, USA). On day 3, medium was changed to 100% NGN2 medium supplemented with 0.8 μg/ml puromycin (Sigma-Aldrich, USA). On day 5, the selection process was stopped by replacing medium without antibiotics. Medium was changed every other day until day 14, where the cells were harvested for subsequent experiments.

### Total RNA sequencing

Bulk RNA-sequencing was carried out for the following samples (each done in biological triplicates). Dataset 1: WT/WT iPSCs, NPCs, d28 cerebral organoids, and d49 cerebral organoids, 70Q/70Q iPSCs, NPCs, d28 cerebral organoids, and d49 cerebral organoids, WT/70Q NPCs. Dataset 2: NGN2 neurons from C1, C2, C3, HD1, HD2, and HD3. Total RNA was isolated using the Qiagen isolation kit and quality-checked by Nanodrop analysis (Nanodrop Technologies). Briefly, total RNA was mixed with 1 μg of a DNA oligonucleotide pool comprising 50-nt long oligonucleotide mix covering the reverse complement of the entire length of each rRNA (28S rRNA, 18S rRNA, 16S rRNA, 5.8S rRNA, 5S rRNA, 12S rRNA), incubated with 1U of RNase H (Hybridase Thermostable RNase H, Epicentre), purified using RNA Cleanup XP beads (Agencourt), DNase treated using TURBO DNase rigorous treatment protocol (Thermo Fisher Scientific, USA) and purified again with RNA Cleanup XP beads. rRNA-depleted RNA samples were further fragmented and processed into strand-specific cDNA libraries using TruSeq Stranded Total LT Sample Prep Kit (Illumina, USA) and sequenced on NextSeq 500, High Output Kit, $1 \times 150$ cycles. Raw sequencing reads were mapped to the human genome (GRCh38 assembly) using STAR (version 2.6.0c) aligner[98]. We used the default settings, with the exception of --outFilterMismatchNoverLmax, which was set to 0.05. Reads were counted using the htseq-count tool, version 0.9.1[99], with gene annotation from GENCODE release 27[100]. Differential gene expression analysis was performed using the DESeq2 (version 1.20.00) R package[101]. All genes with the adjusted P-value lower than 0.05 were considered differentially expressed. Functional enrichment analysis was done using the gProfileR R package[102] (v. 0.6.6) with default settings. All expressed genes were used as background. For heatmap generation for pathway genes of Dataset 1, bulk-RNAseq count matrices were loaded into R (v. 4.2.2). Additional organoid data from neuruloids[25] was processed from scRNAseq data to pseudobulk data by randomly pooling single cells into 3 bulk replicates per cell line per condition (HD and WT/WT) and then also loaded into R. The raw matrices were then normalized to counts per millions (CPM) and filtered to only contain genes with more than 1 CPM using edgeR (v. 3.40.2). For the genes corresponding to a pathway of interest, all the different samples and their corresponding control samples were compared with edgeR, yielding the calculated log2fc and adjusted p values with glmLRT based on two-sided likelihood ratio test without multiple comparison adjustments. For each pathway, the obtained results were plotted in a separate heatmap with colors indicating the magnitude of the log2fc and stars indicating the statistical significance. The case of $0.05 > p\_val > 0.01$ corresponds to one star, $0.01 > p\_val > 0.001$ corresponds to two stars, and everything below corresponds to three stars.

### Long-read RNA sequencing

Long-read RNA sequencing was performed using samples from Dataset 1: WT/WT NPCs vs. 70Q/70Q NPCs. We employed full-length poly(A) and mRNA sequencing (FLAM-seq) for high-quality sequencing of entire mRNAs as described before (Legnini et al.). Briefly, total RNA was processed for poly(A) selection with the Truseq mRNA

preparation kit (Illumina, USA) starting from 10 μg of RNA. For full-length mRNA library preparation, Poly(A)-selected RNA was tailed using the USB poly(A) length assay kit (Thermo Fisher Scientific, USA) in a 20 μl reaction with 14 μl RNA eluted from the poly(A) selection, 4 μl tail buffer mix, 2 μl tail enzyme mix, for 1 h at 37 °C. Reaction was stopped with 1.5 μl of tail stop solution and 1 μl was checked on a picochip to exclude RNA degradation. Tailed RNA was then cleaned up with 1.8× RNAClean XP Beads (Beckmann Coulter, Germany) and eluted in 18 μl water. After reverse transcription, the resulting cDNA was purified with 0.6× XP DNA beads (Beckmann Coulter, Germany) and resuspended in 42 μl of $H_2O$. The full-length mRNA library was then amplified by PCR with the Advantage 2 DNA polymerase mix (Takara Bio, Japan) according to the following mix: 10 μl 10× Advantage 2SA PCR buffer, 40 μl cDNA, 2 μl 10 mM dNTPs mix, 2 μl 5′ PCR primer 10 μM, IIA, 2 μl 50× Advantage 2 Polymerase mix, 42 μl H2O and 2 μl of the 10 μM reverse transcription primer-matched universal reverse primer. The PCR reaction was then performed in a thermocycler as follows: 98 °C for 1 min, 23 × (98 °C for 10 s, 63 °C for 15 s, 68 °C for 3 min), 68 °C for 10 min. The amplified library was then purified with 0.6× Ampure XP DNA Beads and resuspended in 42 μl $H_2O$. One microliter of reaction was checked on a fragment analyzer using High Sensitivity NGS Fragment Analysis Kit (Advanced Analytical Technologies, Germany). For PacBio sequencing the purified PCR libraries were submitted to the Genomics Core Facility of the MDC. Sequencing libraries were prepared using the PacBio Amplicon Template Preparation and Sequencing Protocol (PN 100-081-600) and the SMRTbell Template Prep Kit 1.0-SPv3 according to the manufacturer's guidelines. Sequencing on the SEQUEL IIe SYSTEM (PacBio, Pacific Biosciences of California, USA) was performed in Diffusion mode using the Sequel Binding and Internal Ctrl Kit 2.0. Every library was sequenced on two or three SMRT Cells 1 M v.2 with a 1 × 600 min movie. Circular Consensus Sequence (CCS) reads were generated within the SMRT Link browser 5.0 (minimum full pass of three and minimum predicted accuracy of 90). All primer sequences are reported in Supplementary Data 11.

The bioinformatic analysis of Single Molecule Real Time (SMRT) used a modified IsoSeq3-pipeline followed by an own pipeline of open-source tools and scripts, as described before[103]. For primary analysis, the Isoseq3-Pipeline from Pacific Biosciences has been used; followed by a secondary Analysis Pipeline with open-source tools and modified scripts. This included mapping to genome with Minimap2[104], collapsing transcripts with cDNA Cupcake and Annotation with SQANTI2[105]. Semiquantitative Analysis of the Long-Read-Transcriptomic Data together with graphical illustrations of the underlying alternative splicing ("exon-intron-skippings") was performed using a modified TALON-SWAN pipeline. The scripts for the analysis of SMRT-Data are available upon request and were executed on the High-Performance Cluster using BioConda-Environment. As coding editor, "Sublime Text" for Windows (Version Build 4113) was used. The Scripts here contain the essential information on settings, please be aware that not every step (e.g., copying or moving of files) is mentioned. Transcriptomic Long-Read-Data was preprocessed for further usage through a modified pipeline using the TALON tool to enable technology-agnostic long-read analysis pipeline for transcriptome discovery and quantification. Then, using the python library "SWAN", a library for the analysis and visualization of long-read transcriptomes[106], a reference transcriptome was added and abundance information from the initial SMRT analysis was used to create graphs showing the differential expression between novel and known isoforms within the dataset. Therefore, we defined groups according to the experimental setup and used the transcript information to process the corresponding transcript path. This analysis pipeline is available upon request. Running environment of all Scripts was CentOS 7.7.1908-based at the High-Performance Cluster (HILBERT, Centre for Information and Media Technology (CIM/ZIM) at Heinrich Heine University, Düsseldorf).

## Single-cell RNA sequencing

Regionalized midbrain organoids at day 35 were prepared for single-cell RNA sequencing (scRNAseq) from the following samples of Dataset 1: WT/WT and 70Q/70Q. The data were analyzed in three biological replicates, where each replicate was composed of 48 individual midbrain organoids. Sequencing was carried out in two separate batches. In the first batch, we processed replicate 1 and 2 for WT/WT with replicates 1 and 2 for 70Q/70Q. In the second batch, we processed replicate 3 WT/WT with replicate 3 for 70Q/70Q. In this manner, we could assess the reproducibility of the sequencing results across different independent runs. Midbrain organoids were dissociated using the Papain Dissociation System (Worthington Biochemical, USA). 5 ml of EBSS was added to one papain vial, which was subsequently incubated in a water bath at 37 °C for 10 min. 500 μl of EBSS was then added to one DNase vial and gently mixed. 250 μl of that DNAse mix was added to the papain vial resulting in a final concentration of ~20 units/ml papain and 0.005% DNase. About 50-65 organoids were transferred into one well of a 24-well-plate and washed two times with PBS. 2 ml of the pre-warmed papain/DNase solution was added per well, and the plate was placed onto an orbital shaker with the speed set to 65 rpm for 5–10 min at 37 °C. Using a 1 ml pipette, the organoids were gently triturated by mixing up and down and placed back into the incubator for another 10 min for further enzymatic digestion. The digested organoids were then collected in a 15 ml tube and 5 ml of the organoid culture medium was added. By pipetting up and down with a 225 mm polished glass pipette (Brand GMBH, Germany), the mixture was triturated until mostly dissociated. Any piece of not dissociated tissue was allowed to settle to the bottom of the tube. The cell suspension was then transferred to a new 15 ml tube and filtered through a 30 μm cell strainer (Miltenyi biotech, Germany). The cell concentration was determined, and the cells were pelleted by centrifuging at $300 \times g$ for 5 min at room temperature (RT). After aspirating the supernatant, the pellet was re-suspended in 200 μL ice-cold PBS. 800 μL of freezing-cold 100% methanol was added to the cell suspension and stored at −80 °C until further analysis. Samples after rehydration were subjected to scRNAseq using the 10× Genomics Chromium Single Cell 3′ Gene Expression system with feature barcoding technology for cell multiplexing. Libraries were generated using 10x Chromium and sequenced with Illumina NovaSeq 6000 platform (Illumina, USA). We processed demultiplex, raw paired-end scRNAseq data using Cell Ranger (v. 7.1.0) from 10x Genomics (https://www.10xgenomics.com/support/software/cell-ranger/latest). We used the "cell ranger count" function to align reads to the human genome (GRCh38) and generate digital gene expression (DGE) matrices for each sample. DGEs were further processed in R (v. 4.2.2) using Seurat (v. 5.0.2). Raw DGEs were imported using the function "Read10x" and only genes detected in at least 5 cells were kept for downstream analyses. Similarly, cells will less than 500 genes and 1500 unique transcripts were discarded. After merging all samples in a single Seurat object, we performed gene expression normalization, scaling and computed principle components for each sample independently, followed by data integration using the function "IntegrateLaters" and the "CCAIntegration" method. Then, we performed unbiased clustering and removed clusters 11 and 17 due to abnormally elevated ribosomal and mitochondrial gene expression, respectively. Finally, we performed dimensionality reduction using the "RunUMAP" function and repeated the unbiased clustering step. Clusters were manually annotated in 4 main cell types analyzing their marker genes identified using the "FindAllMarkers" function. Plots were generated with ggplot2 (v. 3.5.0) and piping using dplyr (v. 1.1.4).

## Proteomic analysis

We carried out label-free quantification (LFQ) proteomics with sample preparation according to a published protocol with minor modifications[107]. We used biological triplicates of Dataset 1: i) WT/WT iPSCs, ii) WT/WT NPCs, iii) 70Q/70Q iPSCs, iv) 70Q/70Q NPCs. We also

used biological triplicates of Dataset 2: i) NGN2 neurons C1, ii) NGN2 neurons C2, iii) NGN2 neurons C3, iv) NGN2 neurons HD1, v) NGN2 neurons HD2, vi) NGN2 neurons HD3. We lysed all samples under denaturing conditions in buffer with 3 M guanidinium chloride (GdmCl), 5 mM tris-2-carboxyethyl-phosphine, 20 mM chloro-acetamide, and 50 mM Tris-HCl at pH 8.5. We denatured lysates at 95 °C for 10 min in a thermal shaker at 1000 rpm, and sonicated them in a water bath for 10 min. A small aliquot of cell lysate was used for the BCA assay to quantify protein concentration. We diluted the lysates (100 µg proteins) using a dilution buffer containing 10% acetonitrile and 25 mM Tris-HCl, pH 8.0, to reach a 1 M GdmCl concentration. We digested proteins with 1 µg LysC (MS-grade, Roche, Switzerland) while shaking at 700 rpm at 37 °C for 2 h. The digestion mixture was diluted with the same dilution buffer to reach 0.5 M GdmCl. We added 1 µg trypsin (MS-grade, Roche, Switzerland) and incubated the digestion mixture in a thermal shaker at 700 rpm at 37 °C overnight. We used solid phase extraction (SPE) disc cartridges (C18-SD, Waters, USA) for peptide desalting, according to the manufacturer's instructions. We reconstituted desalted peptides in 0.1% formic acid in water and further separated them into four fractions by strong cation exchange chromatography (SCX, 3 M Purification, Meriden, USA). We next dried the eluates in a SpeedVac, dissolved them in 20 µl 5% acetonitrile and 2% formic acid in water, briefly vortexed them, and sonicated them in a water bath for 30 s prior to injection into nano-LC-MS. We carried out LC-MS/MS by nanoflow reverse phase liquid chromatography (Dionex Ultimate 3,000, Thermo Fisher Scientific, USA) coupled online to a Q-Exactive HF Orbitrap mass spectrometer (Thermo Fisher Scientific, USA). LC separation was performed with a PicoFrit analytical column (75 µm ID × 55 cm long, 15 µm Tip ID; New Objectives, Woburn, MA, USA) in-house packed with 3 µm C18 resin (Reprosil-AQ Pur, Dr. Maisch, Germany). We eluted peptides using a gradient from 3.8 to 40% solvent B in solvent A over 120 min at 266 nl per minute flow rate. Solvent A was: 0.1% formic acid, while solvent B was: 79.9% acetonitrile, 20% H2O, 0.1% formic acid. Nanoelectrospray was generated by applying 3.5 kV. A cycle of one full Fourier transformation scan mass spectrum (300−1750 m/z, resolution of 60,000 at m/z 200, AGC target 1e6) was followed by 12 data-dependent MS/MS scans (resolution of 30,000, AGC target 5e5) with a normalized collision energy of 25 eV. We used a dynamic exclusion window of 30 s to avoid repeated sequencing of the same peptides. We sequenced only peptide charge states between two and eight. Raw MS data were processed with MaxQuant software (v1.6.10.43) and searched against the human proteome database UniProtKB with 20,600 entries, released in 05/2020. Parameters of MaxQuant database searching were a false discovery rate (FDR) of 0.01 for proteins and peptides, a minimum peptide length of 7 amino acids, a mass tolerance of 4.5 ppm for precursor, and 20 ppm for fragment ions. We used the function "match between runs". A maximum of two missed cleavages was allowed for the tryptic digest. We set cysteine carbamidomethylation as fixed modification, while N-terminal acetylation and methionine oxidation were set as variable modifications. We excluded any contaminants from further analysis, as well as any proteins identified by site modification or derived from the reversed part of the decoy database. We report the MaxQuant processed output files, peptide and protein identification, accession numbers, percentage sequence coverage of the protein, *q*-values, and LFQ intensities in Supplementary Data 4, Data 5, Data 8. We performed the correlation analysis of biological replicates and the calculation of significantly different metabolites and proteins using Perseus (v1.6.14.0). We transformed by log2 the LFQ intensities originating from at least two different peptides per protein group. We employed only protein groups with valid values within compared experiments. We carried out statistical analysis by a two-sample two-tailed t-test with Benjamini-Hochberg (BH, FDR of 0.05) correction for multiple testing. For comprehensive proteome data analyses, we applied gene set enrichment analysis (GSEA)[108]. This was carried out to determine if a priori defined sets of proteins show statistically significant and concordant differences between mutations and controls. For GSEA analysis, we used all proteins with ratios calculated by Perseus. We applied GSEA standard settings, except that the minimum size exclusion was set to 5 and Reactome (v. 7.2) and KEGG (v. 7.2) were used as gene set databases. The cutoff for significantly regulated pathways was set to be ≤0.05 p-value and ≤0.25 FDR. We report the results of the GSEA analysis in Supplementary Data 9.

## Metabolomics analysis

Metabolites were analyzed by a targeted LC-MS approach. Metabolite extraction was performed as reported previously[109]. We harvested biological triplicates of Dataset 2: i) NGN2 neurons C1, ii) NGN2 neurons C2, iii) NGN2 neurons C3, iv) NGN2 neurons HD1, v) NGN2 neurons HD2, vi) NGN2 neurons HD3. We aspirated the culture medium, quickly rinsed the cells twice with ice-chilled 1x PBS, pelleted the cells, and shock-froze them in liquid nitrogen. A portion of the protein pellets were used in BCA protein assay for normalization among samples. We extracted the metabolites with methyl tert-butyl-ether (MTBE), methanol, and water. The protein pellets were reconstituted in water, acetonitrile, and 50% methanol in acetonitrile. To each sample we added an internal standard mixture containing chloramphenicol, C13-labeled L-glutamine, L-arginine, L-proline, L-valine, and uracil (10 µM final concentration). We used SpeedVac to dry the aliquots. We dissolved dry residuals in three different solvents: i) 100 µl 50% acetonitrile in MeOH with 0.1% formic acid, ii) 100 µl MeOH with 0.1% formic acid for analysis by HILIC column, or iii) 100 µl water, 0.1% formic acid for C18 column mode. We transferred the supernatants to micro-volume inserts. We injected 20 µl per run for subsequent LC-MS analysis. Each sample was finally analyzed in 15 individual LC-MS runs with specific settings for ion polarity, buffers, flow rates, and columns and their temperatures. Over 400 metabolites were selected to cover most of the important metabolic pathways in mammals, based on previous studies[110]. Since metabolites are very diverse in their chemical properties, we used two different LC columns for metabolite separation: Reprosil-PUR C18-AQ (1.9 µm, 120 Å, 150 ×2 mm ID; Dr. Maisch, Germany) and zicHILIC (3.5 µm, 100 Å, 150 ×2.1 mm ID; Merck, Germany). We used the settings of the LC-MS instrument,1290 series UHPLC (Agilent Technologies, USA) online coupled to a QTrap 6500 (Sciex, USA)[110]. The buffer conditions were A1, 10 mM ammonium acetate, pH 3.5 (adjusted with acetic acid); B1, 99.9% acetonitrile with 0.1% formic acid; A2, 10 mM ammonium acetate, pH 7.5 (adjusted with ammonia solution); B2, 99.9% methanol with 0.1% formic acid. We prepared all buffers in LC-MS grade water and organic solvents. We performed peak integration with MultiQuantTM software v.2.1.1 (Sciex, USA) and reviewed it manually. We normalized peak intensities, first against the internal standards, and subsequently against protein abundances obtained from the BCA assay. We used the first transition of each metabolite for relative quantification between samples and controls. Statistical analysis was carried out with Perseus[111]. We employed a two-sample two-tailed t-test with Benjamini-Hochberg (BH, FDR of 0.05) correction for multiple testing. We provide the list of all metabolites including MRM ion ratios, KEGG and HMDB metabolite identifiers, and statistical values in Supplementary Data 10. These data were obtained using a previously reported LC-MS method containing the list of metabolites, transitions, and retention times[110].

## Proteomic-driven functional metabolic analysis

For the functional metabolic analysis of the proteomic Dataset 1 (including iPSCs and NPCs from WT/WT and 70Q/70Q), we used the Quantitative System Metabolism (QSM) pipeline developed by Doppelganger Biosystem GmbH. We used the approach as described before[44]. QSM data analysis used quantitative information on the expression levels of metabolic proteins (enzymes) to determine metabolic profiles, metabolic states and capacities, and metabolic

fluxes. To ensure data quality, we used the quality control (QC) score, which counts the number of proteins of interest found, and the QSM score, which evaluates the number of metabolic processes associated with the enzymes found. For our dataset, the QC score was about 80% and the QSM score was around 100%, indicating excellent data quality that ensures reliable interpretability of the results. The kinetic model includes major cellular metabolic pathways of energy metabolism in neuronal cells, as well as key electrophysiological processes at the inner mitochondrial membrane, the mitochondrial membrane potential membrane, the transport of various ions, and the utilization of the proton motive force. Maximal enzyme activities (Vmax values) were estimated based on functional characteristics and metabolite concentrations of healthy neuronal tissues[112]. Individual metabolic models were inferred using protein intensity profiles. The maximal activities (vmx) mean control for the normal state were previously calculated[112]. QSM was used to calculate maximal energetic capacity. Energetic capacity was assessed under saturating glucose and oxygen concentrations, corresponding to healthy physiological conditions. Energetic capacities were evaluated by computing the changes of metabolic state elicited by an increase of the ATP consumption rate above the resting value. Maximal neuronal glucose uptake rate was assessed by increasing plasma glucose concentration in a systematic manner as the only energy delivering substrate, assuming high energy demands, and providing saturating oxygen concentrations.

## Integration of transcriptomics, proteomics, and metabolomics from NGN2 neurons

Integration analysis was performed for Dataset 2: NGN2 neurons from control iPSCs (C1, C2, C3) vs. NGN2 neurons from HD patient iPSCs (HD1, HD2, HD3). For enrichment analysis, over-representation analysis (ORA) of KEGG pathways of either transcriptomics or proteomics data was performed with Bioconducter clusterProfiler (v.3.0.4), using the significantly up- or down-regulated molecules (adj. $p$ value ≤ 0.05 and logFC>0 or logFC<0, respectively). Dysregulated pathways with a set size of 2 to 500 genes and a $p$ value ≤ 0.05 were considered as significantly dysregulated. Networks of dysregulated pathways were constructed via Bioconductor Enrichplot v1.10.2 over-representation analysis (ORA) for human disease ontologies was performed using the Bioconductor DOSE package, and the top 5 enriched terms with $p$ value ≤ 0.05 were plotted in circular networks along with their associated genes. For Gene Set Enrichment Analysis (GSEA), all transcripts or proteins were loaded without prior filtering in clusterProfiler v.3.0.4 and analyzed for associated biological processes (GO BPs). Terms with an adj. p-value ≤ 0.05 and more than 30 associated genes were considered significantly dysregulated. Term redundancy was removed setting the cut-off at 0.6. The top 20 dysregulated BPs were plotted according to their normalized enrichment score (NES). For co-expression analysis, protein co-expression analysis was performed using the ProtExA tool[113]. Briefly, normalized protein expression data was uploaded and analyzed according to the steps indicated in the platform. No further normalization method was applied, while proteins were filtered for a p-value ≤ 0.05. Top-scored proteins obtained from pre-processing steps were annotated for the construction of co-expression networks using the ARACNEM algorithm[114]. Co-expressed proteins were clustered with the Walktrap algorithm and edge weight calculation. The clustered network containing the proteins of interest was imported into Gephi (v. 0.10.1)[115], and analyzed further for community detection with a 2.5 resolution. Enrichment analysis for potential transcription factors of co-expressed proteins was performed through TFEA.ChIP tool kit[116] using the default parameters, while gene-disease associations were retrieved from GeDiPNet database[117] through the EnrichR platform[118]. For metabolomics analysis, metabolites detected by LC-MS/MS were analyzed for their chemical similarity and clustered based on their class using the ChemRICH tool[119]. Significantly dysregulated metabolites with a $p$ value ≤ 0.05

were annotated in the ConsensusPathDB[120] for enrichment analysis of affected biological pathways against KEGG, Reactome, SMPDB, EHMN, INOH and HumanCyc databases[121–126]. Interactions of Significantly dysregulated metabolites were obtained using the MetaboAnalyst platform (v. 5.0)[127]. OmicsNet platform (v. 2.0) was used for the integration of transcriptomics, proteomics, and metabolomics data[128]. Significantly dysregulated transcripts (adj. $p$-value ≤ 0.05), proteins (adj. $p$-value ≤ 0.05), and metabolites ($p$-value ≤ 0.05) were uploaded as input lists. Interactions for single- and multi-modalities were predicted using multiple databases. Specifically, for proteins, protein-protein interactions were retrieved from STRING[129] with a 0.9 confidence score and protein-metabolite interactions were retrieved from KEGG. Edges were generated only for the input lists without adding new nodes. For transcripts, miRTarBase[130] and ENCODE[131] provided information for mRNA-miRNA and TF-mRNA interactions, respectively, while Recon3D[132] was used for metabolite interactions. The Prize-Collecting Steiner Forest (PCSF) algorithm[133] was applied to minimize the overall cost of including edges. The outcome was a subnetwork of the most informative nodes representing the backbone of the organization and functionality of the initial network. Nodes were annotated in KEGG and Reactome for single-omics or multi-omics enrichment analysis. Network visualization was made using the 3D BioLayout in Cytoscape (v. 3.9.1)[134,135].

## siRNA-based knock-down experiments and branching analysis

For short interfering RNA (siRNA)-based knock-down experiments, siRNA (Dharmacon, USA) was incubated with JetPRIME transfection reagent (Polyplus-transfection, France) and μClear black 384-well-plates (Greiner Bio-One, Austria) were coated with 1 pmol siRNA per well. The plates were dried in a SpeedVac and stored at 4 °C wrapped with parafilm until use. Before adding the neurons on the plates, they were additionally coated for 1 h at 37 °C with ready-to-use Geltrex (Gibco, USA). 13 days-old neurons differentiated from the BIHi-005-A-24 line were detached with Accutase (Sigma-Aldrich, USA) for 3 min and collected. The cells were centrifuged at $100 \times g$ for 5 min and resuspended in neuronal medium. We quantified neuronal branching using our high-content analysis (HCA) assay[53]. Briefly, 5,000 neurons were seeded per well to the 384-well plate and cultured for 4 days. At day 17 of differentiation, the neurons were fixed by adding 8% PFA (Thermo Fisher Scientific, USA) to the medium and incubated for 20 min. After washing 3 times with PBS (Gibco, USA), 0.05% sodium azide (Sigma-Aldrich, USA) in PBS solution was added to the wells and the plate was stored at 4 °C wrapped with parafilm until staining. For staining, the cells were blocked for 1 h at RT with blocking solution (PBS [Gibco, USA], 3% BSA, 0.05% sodium azide, 0.5% Triton-X-100 [all Sigma-Aldrich, USA]). Then, they were treated with primary antibodies anti-MAP2 and anti-SMI312 in blocking solution over night at 4 °C wrapped with parafilm. Afterwards, they were washed 3 times with PBS and treated with secondary antibodies against guinea pig (AF 488) and mouse (AF 568) and Hoechst 33342 (1:2500; Life Technologies, USA) in blocking solution for 1 h at RT. After 3 washes with PBS, the wells were filled with 0.05% sodium azide in PBS and stored at 4 °C wrapped with parafilm. They were imaged with the high-content microscope Operetta (PerkinElmer, USA) with a 0.4 NA 20× air objective and 25 images per well. The images were analyzed with the open-source software CellProfiler (v. 4.2.1) with respect to the axon and dendritic growth and arborization. The data was summarized by calculating the median per well and to compare different plates, the numbers were compared to the median of the control wells (non-targeting scrambled siRNA). siRNA sequences and antibodies are reported in Supplementary Data 11.

## PCR analyses

For HTT PCR, genomic DNA was harvested from iPSCs colonies with Phire Animal Tissue Direct PCR Kit (Qiagen, USA). Exon 1 of the HTT

locus was amplified by PCR using AmpliTaq Gold 360 DNA polymerase (Thermo Fisher Scientific, USA) and HTT-specific primers using following PCR conditions: 95 °C 10 min, 35 cycles: 95 °C for 30 s, 55 °C for 30 s, and 72 °C for 1 min. Products were visualized by agarose gel electrophoresis, individual DNA bands were excised and purified using The Wizard® SV Gel and PCR Clean-Up System (Promega), cloned with CloneJET PCR Cloning Kit (Thermo Fisher Scientific, USA), and submitted to LGC Genomics for Sanger sequencing. Chromatograms were analyzed using CLC Genomics Workbench (Qiagen, USA). For quantitative real-time PCR (qPCR) analysis of cerebral organoids, RNA extraction was performed from four cerebral organoids per condition with RNeasy kit (Qiagen, USA). cDNA was generated from 100 ng RNA using the first strand cDNA synthesis kit (Thermo Fisher Scientific, USA). qPCR experiments were conducted for three technical replicates and three biological replicates with the SYBR green Mastermix (Thermo Fisher Scientific). Amplification of 10 ng RNA was performed in a Viia7 Real-Time PCR system (Thermo Fisher Scientific) as follows: 2 min activation step at 50 °C, followed by 10 min at 95 °C, 40 cycles of denaturation for 15 s at 95 °C, and annealing for 1 min at 60 °C, finalized with 15 s at 95 °C, 1 min at 60 °C, and 15 s at 95 °C. After averaging the CT-values of the three technical replicates, the expression levels of the genes of interest were normalized relative to the average expression of housekeeping genes GAPDH, OAZ1 and ACTB using the delta-delta-CT method. All primer sequences are reported in Supplementary Data 11.

## Immunostaining

Cerebral organoids were fixed with 4% PFA for 20 min at RT and washed 3x with PBS for 15 min. Next, organoids were cryoprotected overnight in a 30% sucrose in PBS solution. Following this, organoids were embedded in a 10% sucrose/13 % gelatin solution in PBS and stored at −80 °C upon further use. Hereafter, sections of 20 μm were cut using a Leica CM3050 S cryostat, collected on Superfrost Plus slides and stored at −80 °C. Next, sections were washed three times with warm PBS for 10 s to dissolve and remove any remaining gelatin. Subsequently, sections were fixed again with 4% PFA for 20 min at RT, washed 3× for 10 min with PBS, and blocked for 1 h at RT in blocking solution (PBS, 10% normal donkey serum, 1% Triton X-100). Sections were incubated overnight at 4 °C with the primary antibodies diluted in blocking solution, washed 4× for 10 min with 0.1% Triton X-100 and 0.05% Tween-20 in PBS, and incubated for 1–2 h at RT with Hoechst 33342 (1:2500; Life Technologies, USA) and the secondary antibodies diluted in blocking solution. Finally, sections were washed 4x for 10 min with 0.1% Triton X-100 and 0.05% Tween-20 in PBS, mounted, air-dried and images were taken using the confocal microscope Zeiss Z1 (Zeiss, Germany), Olympus Fluoview 3000 (Olympus, Japan) or ZEISS Axio Observer with an apotome 3 (Zeiss, Germany). NPCs and midbrain organoids were stained following the same procedure. NPCs grown on Matrigel-coated (Corning, USA) coverslips and free-floating midbrain organoids were fixed with 4% PFA (Science Services, Germany) for 20 min at RT and washed three times with PBS. For permeabilization, cells or organoids were incubated with blocking solution (PBS [Gibco, USA], 10% normal donkey serum, 1% Triton-X-100, 0.05% Tween-20 [all Sigma-Aldrich, USA]) for 1 h at RT. Primary antibodies were diluted in blocking solution and incubated overnight at 4 °C on a shaker. Next, the cover slips or organoids were washed three times with PBS and incubated for 1 h at RT with Hoechst 33342 (1:2500; Life Technologies, USA) and the secondary antibodies in blocking solution. After incubation, cover slips or organoids were washed three times with PBS. Finally, cover slips were mounted, air-dried and images were acquired using the confocal microscope Olympus Fluoview 3000 (Olympus, Japan) or ZEISS Axio Observer with an apotome 3 (Zeiss, Germany). Images of free-floating midbrain organoids were acquired using the ZEISS Axio Observer with an apotome 3 as a z-stack, which were then deconvoluted using the Zeiss blue software (v. 3.1) default settings and z-projected with maximum

intensity. Cortical organoids were fixed with 4% PFA for 1 hr at RT and washed 3× with PBS for 15 min. Cortical organoids were embedded into tissue molds that were filled with 2% low melting point agarose. The agar blocks containing the organoids were sliced using a Leica VT1000 S vibratome in ice-cold PBS at 50 μm thickness. Free floating sections were gently transferred to well plates containing PBS and stored at 4 °C until time of staining. Staining and imaging were done in the same manner as the midbrain organoids. Detailed information about antibodies used can be found in Supplementary Data 11.

## CHCHD2 and TOM20 imaging quantification

Images of NPCs and cerebral organoids stained against TOM20, CHCHD2, and Hoechst were processed as follows. For NPCs, we analyzed at least two biological replicates, with 11 images being taken for each replicate. For the cerebral organoids, we analyzed at least two biological replicates, with at least three slices of each organoid and three images per slice. In total, we analyzed 77 individual images for 70Q/70Q and 35 individual images for WT/WT. We performed background subtraction using the rolling ball algorithm implementation of Scikit-image[136] with a radius of 50 pixels. We applied Gaussian blur with a radius of three pixels, to smoothen the images. To separate signal from background, we used Otsu's thresholding method on the respective color channels. We then calculated the amount of colocalized pixels above the threshold for each marker. For comparing amounts of active TOM20 or CHCHD2 signals of different data sets, the amounts of positive signal per image were normalized by the amount of Hoechst signal per image to account for the fact that different images contain different number of cells. Mann-Whitney U tests were used to test for differences between data sets (cell lines or treatments). For mitochondrial network morphology assay of cerebral organoids, pictures of 100x magnification were taken of TOM20 stained sections using a Zeiss Z1 microscope. Next, the pictures were analyzed using ImageJ, as described before[41]. Briefly, background was reduced, local contrast enhanced, pictures were made binary and using a "tubeness" plugin mitochondrial structures were tubed. After bandpass filtering, a threshold was set to minimize extremely small and large structures. Lastly, particles were analyzed, yielding information about the size and morphology of the structures. Small mitochondrial structures range were defined as <0.5 μm and large mitochondrial structures as >0.5 μm.

## Western blotting of HTT and filter retardation assay (FRA)

Samples from NPCs and guided brain organoids were resuspended in 150 μl RIPA buffer (150 mM NaCl, 50 mM Tris, 0.5% w/v sodium desoxycholate, 1% v/v Triton X-100, 0.1% SDS, before using freshly added 1 mM PMSF and Protease Inhibitor Cocktail 1:100). Cell suspensions were pipetted 5x by using syringes to pass through 27 G needles. The lysates were incubated at 4 °C for 1 h on a turning wheel followed by repeating the step of 27 G needle treatment. The lysates were centrifuged at 15,000 x g for 15 min at 4 °C. The supernatants were transferred to tubes and the protein concentrations were measured by using a BCA Protein Assay kit (Thermo Fisher Scientific, USA). For western blot, 5 μg proteins of each sample were used for detection. In these experiments, polyacylamide gels consisting of a 4% stacking gel and a 4-12% discontinuous resolving gel were applied. After SDS-PAGE, the proteins were transferred onto nitrocellulose membrane (Amersham Protan 0.45μm NC, GE Healthcare Life science) by tank blotting either at 500 mA, RT for 1.5 h or 100 mA, 4 °C for 16 h. Proteins on the membranes were detected by using anti-HTT (Anti-Huntingtin antibody [EPR5526] (ab109115) Abcam; 1:5,000), primary antibody followed by anti-rabbit-HRP (Abcam; 1:10,000) secondary antibody. For Filter Retardation Assay (FRA), 3 μg protein of each sample was loaded. Both cellulose acetate membrane (Cellulose Acetate Membrane Filter 0.2 Micron, STERLITECH Corporation) and nitrocellulose membrane were equilibrated in PBS/0.01% SDS. The dot blot apparatus was

assembled with either membrane and the pump system was connected and turned on continuously. After washing the membranes with PBS, the samples were applied followed by two washing steps. The membranes were dried and proteins on the membranes detected with the same antibodies applied in western blot. The quantified figures were analyzed, and all signals were evaluated in ImageJ (v. 1.53a). The signals were calculated by considering white as a maximum value of 65536, and black as a minimum value of 0. All experiments were repeated three times. The values in the figures indicate the average values of all repeats and replicates.

## Western blotting of CHCHD2

Cell pellets were re-suspended in 50 µl of lysis buffer (50 mM Tris, 150 mM NaCl, 1 mM EDTA, 1% TritonX-100 + 1x cOmplete™, EDTA-free protease inhibitor cocktail (Roche) and incubated on ice for 30 min. The lysate was then centrifuged at 14,000 x g for 5 min at 4 °C to remove any cell debris. The supernatant was transferred to new vials and the protein concentration was determined via Bradford assay (ROTI®Quant; Roth). For SDS-PAGE, 25 or 50 µg of protein per lane was loaded on a 12% acrylamide gel. Proteins were blotted onto a nitrocellulose membrane using semi-dry Western blotting (Trans-Blot Turbo Transfer System; BIO-RAD). The membranes were then blocked with 5% (w/v) milk powder in TBS-T on a shaker at RT for 1 h, followed by incubation with the primary antibody diluted in 3% (w/v) milk powder in TBS-T on a shaker at 4 °C overnight. The next day, the membranes were washed three times for 10 min in TBS-T before being incubated with the secondary antibody diluted in 3% (w/v) milk powder for 4 h at RT on a shaker. After incubation with the secondary antibody, the membranes were washed again with TBS-T before Pierce™ ECL Western Blotting Substrate was added to the membranes. The chemiluminescence signal was detected using the ChemoStar (Intas) or Amersham™ Imager 600 (GE). Western blots were then analyzed using ImageJ (v. 1.53a). CHCHD2 polyclonal antibody (19424-1-AP Proteintech 1:500–1:1000) and beta actin monoclonal antibody (ab8224 Abcam 1:1000 and 66009-1-lg Proteintech 1:6000) were used as primary antibodies. Goat IgG anti-rabbit IgG (H + L)-HRPO (111-035-003 DIANOVA 1:3000 – 1:5000) and goat IgG anti-mouse IgG (H + L)-HRPO (115-035-003 DIANOVA 1:10000) were used as secondary antibodies.

## Electron microscopy

Electron microscopy (EM) was performed at the Core Facility for Electron Microscopy (CFEM) at the medical faculty of the Heinrich Heine University Duesseldorf. NPCs were grown in 15 cm petri dishes coated with Matrigel. When the NPCs were confluent, they were fixed using 3% glutaraldehyde (Serva) buffered with 0.1 M sodium cacodylate (Serva) buffer (pH 7.2). Following fixation, the cells were washed with 0.1 M sodium cacodylate buffer (pH 7.2) and subsequently stained with 1% osmium tetroxide (Science services) for 50 min at RT. This was followed by additional staining with 1% uranyl acetate (Merck)/1% phosphotungstic acid (Merck) in 70% ethanol for 1 h. To facilitate the dehydration process, the samples underwent serial ethanol treatment before being embedded in spur resin and polymerize at 70 °C for at least 48 h. Ultra-thin sections were prepared using an ultra-microtome (EM UC7). Imaging was conducted using a JEM-2100 plus (JOEL) equipped with an EM-24830 flash CMOS camera system (JOEL). EM images were analyzed using ImageJ (v. 1.53a)[137]. In each image, mitochondrial structures were identified and classified. Then, it was determined if cristae are visible, if yes how many and along what axis they were angled. If they were perpendicular to the major axis of the mitochondria, they were classified as transverse, and contrary, if they were perpendicular to the minor axis of the mitochondria they were classified as longitudinal. Next, the mitochondrion was traced with the freehand selection tool in ImageJ and measured regarding size and shape.

## Structural analysis of the mitochondrial respiratory chain (MRC)

The steady-state levels of MRC subunits were analyzed by SDS-PAGE of whole cell lysates from NPCs (WT/WT, 70Q/70Q and WT/70Q) solubilized with HEPES extraction buffer (20 mM HEPES pH 7.4, 10 mM NaCl, 10% glycerol, 1% Triton X-100, protease inhibitors) for 30 min on ice. Lysates were centrifuged at 16,000 g for 10 min at 4 °C, protein concentration was measured with the BCA protein assay kit (Pearson), and 15 µg protein in 1:1 Laemmli 2x (Biorad) were loaded onto 12.5% polyacrylamide SDS-page gels. Gels were blotted onto nitrocellulose membranes and protein immunoblotting was performed with antibodies targeting representative MRC subunits (detailed in Supplementary Data 11). The relative abundance of MRC complexes and supercomplexes was determined by Blue Native (BN)-PAGE from NPCs (WT/WT, 70Q/70Q and WT/70Q)[48]. Briefly, mitochondrial fractions were isolated, protein concentrations were determined, and samples were solubilized with digitonin at a detergent-to-protein ratio of 4:1. Pre-cast NativePAGE™ 3-12% Bis-Tris gels (Invitrogen) were loaded with 25 µg of mitochondrial protein and processed for blue native electrophoresis. Proteins were then transferred to PVDF membranes at 40 V overnight and probed with antibodies (Supplementary Data 11). Uncropped and unprocessed blot scans are provided in Supplementary Fig. 10.

## Bioenergetic profiling

Live-cell assessment of cellular bioenergetics in NGN2 neurons was performed using Seahorse XF96 extracellular flux analyzer (Seahorse Bioscience, USA), as we described before[83]. Briefly, 20,000 neurons were plated into each Matrigel-coated (Corning, USA) well of the XF96 well plates. Cells were maintained in the plates for 2 weeks. On the assay day, neurons were incubated at 37 °C and 5% CO₂ for 60 min to allow media temperature and pH to reach equilibrium before starting the simultaneous measurement of mitochondrial respiration (oxygen consumption rate, OCR) and anaerobic glycolysis (extracellular acidification rate, ECAR) using the sequential introduction of oligomycin, FCCP, and then rotenone plus antimycin A (all products at 1 µM; Sigma-Aldrich, USA). Normalization to DNA content in each well of the plate was performed using the CyQUANT Kit (Molecular Probes, USA). The supernatants were stored before and after the seahorse assay and used for lactate measurement using a Lactate Fluorometric Assay Kit (Biovision, USA).

## CHCHD2 overexpression

NPCs were transduced with adeno-associated viruses (AAV) carrying a construct to overexpress CHCHD2 (pAAV EF1a CHCHD2 GFP WPRE3; Viral Core Facility Charité, Germany). As a control, AAV which only express GFP were used (scAAV9/pTRs-KS-CBh-EGFP-BGH). These particles were kindly provided by Qinglan Ling and Steven Gray (University of Texas Southwestern Medical Center). NPCs were first seeded onto coverslips coated with Matrigel in a 24-well-plate at a density of $4.8 \times 10^4$ cells/well. 24 h later, they were treated with 1.75E + 11 viral particles per well. After another 24 h, the media was exchanged. 2 days later, the NPCs were fixed and subsequently stained against TOM20 and CHCHD2 as described in the section „Immunostaining". The images were semi-automatically analyzed as follows. All images got a gaussian blur with sigma = 0.5 applied, resulting in a three-pixel blurring radius, to smoothen the images before thresholding. Hoechst and TOM20 images were individually thresholded using Otsu thresholding, and EGFP and CHCHD2 images were thresholded using Triangle thresholding. To be sure that the effect of the AAV transduction is analyzed correctly, only cells with stronger GFP signal than the background fluorescence were considered. The thresholded images were loaded into QuPath[138], and the cells with a stronger marker were manually annotated. The annotated cells showing stronger GFP signal were exported as JSON files before being converted to segmentation masks. The masks were used during quantification to only analyze the

respective areas instead of the whole image. Therefore, it was not necessary to use automated background noise subtraction. The images were then quantified as described in the method section "CHCHD and TOM20 quantification". For mean intensity, the amount of signal (pixels with brightness > 0) was observed. For the area, the amount of signal was observed. The two groups (overexpression and control) were compared using Welch's *t*-test.

## Statistical analysis

Data are expressed as mean and standard deviation (mean ± SD) where normality of the distribution could be verified, or as median and quartiles (median [1st; 4th quartiles]) otherwise. Significance was assessed using parametric tests (Welch's *t*-test, ANOVA) for normally-distributed data and non-parametric tests (Mann-Whitney *U* test, Kruskal-Wallis) when normal distribution could not be verified. Unless otherwise indicated, data were analyzed using GraphPad-Prism (v. 4.0) (GraphPad Software, USA). Schematics were drawn using Inkscape (v. 1.2.1).

## Reporting summary

Further information on research design is available in the Nature Portfolio Reporting Summary linked to this article.

## Data availability

There are restrictions to the availability of the patient-derived iPSC lines due to the nature of our ethical approval that does not support sharing to third parties and does not allow to perform genomic studies to respect the European privacy protection law. The datasets generated during this study are available, in cases data protection laws did not prevent the original datasets from being published. Source Data are provided as a Source Data file. • The RNA-sequencing data generated in this study have been deposited in NCBI Gene Expression Omnibus (GEO) database. Bulk RNA sequencing can be found under accession code GSE233916: https://www.ncbi.nlm.nih.gov/geo/query/acc.cgi?acc=GSE233916. • Single-cell RNA sequencing can be found under accession code GSE271852: https://www.ncbi.nlm.nih.gov/geo/query/acc.cgi?acc=GSE271852. • Long-reads RNA sequencing data generated in this study have been deposited in NCBI Sequence Read Archive (SRA) Bioproject repository according to MIAME compliant data submissions and can be found under the accession code PRJNA1138763: https://www.ncbi.nlm.nih.gov/bioproject/PRJNA1138763/. • The mass spectrometry proteomics data generated in this study have been deposited in the ProteomeXchange Consortium via the PRIDE partner repository[139] under the accession code PXD041846: http://www.ebi.ac.uk/pride/archive/projects/PXD041846. • The mass spectrometry metabolomics data generated in this study have been deposited in Peptide Atlas repository under the accession code PASS04827: http://www.peptideatlas.org/PASS/PASS04827. Source data are provided with this paper.

## Code availability

Codes used for sRNAseq data analysis and related figure plotting are available on GitHub: https://github.com/rajewsky-lab/Huntington_midbrain_organoids[140]. Codes used for CHCHD2 and TOM20 quantification analyses are available on GitHub: https://github.com/Scaramir/HD_colocalization[141]. The pipeline for the analysis of neurite outgrowth with the open-source software CellProfiler is available here: https://zenodo.org/records/6642365[53].

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

## Acknowledgements

We are grateful to Qinglan Ling and Steven Gray (University of Texas Southwestern Medical Center) for providing the GFP-AAV particles. We thank Daria Mochly-Rosen (Stanford University) for help with the mitochondrial network morphology assay. We thank Andrea Borchardt (Institute of Biochemistry and Molecular Biology I, Heinrich Heine University Düsseldorf) for her technical assistance in electron microscopy. We acknowledge support from the Deutsche Forschungsgemeinschaft (DFG) (PR1527/5-1 to A.P., RTG 2155 ProMoAge to H.O. and L.A.M.K., SFB167 B07 to J.P., RU2795: "Synapses under stress": PR-1527/6-1 to A.P. and AN-1440/4-1 to R.A.), the Berlin Institute of Health (BIH) (to S.D., J.P., R.K., and A.P.), the Bundesministerium für Bildung und Forschung (BMBF) (AZ. 031L0211 and 01GM2002A to A.P. and 01EE2303B to J.P.), the Medical Faculty of Heinrich Heine University (FoKo grant to A.P. and S.C.), the European Commission's Horizon Europe Program (SIMPATHIC #101080249 to A.P.), the National Science Centre, Poland (NCN grant No. 20 16/22/M/NZ2/00548 and 2017/27/B/NZ1/02401 to P.L.), the UK Dementia Research Institute programme grant (to J.P.), the Instituto de Salud Carlos III (ISCIII) grant PI20-00057 (to C.U.), the Berlin School of Integrative Oncology through the GSSP program of the German Academy of Exchange Service (DAAD) and the Joachim Herz Foundation through the Add-on Fellowship program (to T.M.P.), and the Studienstiftung des deutschen Volkes (to Se.Li.). We acknowledge the Center for Advanced Imaging (CAi) at Heinrich Heine University Düsseldorf for providing access to the PerkinElmer Operetta CLS (DFG grant number INST 208/760-1 FUGG) and Olympus FV3000 microscope. Electron microscopy was performed at the Core Facility for Electron Microscopy (CFEM) at the Medical Faculty of the Heinrich Heine University Duesseldorf.

## Author contributions

Conceptualization, A.P., P.L.; Methodology, P.L., Se.Li., B.M., A.R-W., St.Le., C.M., W.D., S.L-C., Y.R., L.A.M.K., H.W., D.C.M., M.O., N.N., T.H., A.B., N.S.T.; Formal Analysis, J.J.M., R.A., T.M.P., D.O., M.O., S.N. P.R., P.G., I.L., D.M.; Resources, S.P., C.U., E.M., Am.Po., E.E.W., N.R., R.K., J.K., J.P., S.C., H.O., S.D., J.J.M., A.P.; Writing –Original Draft, A.P.; Writing –Review & Editing, A.P., Se.Li., J.J.M., J.P., S.C., A.R-W.; Supervision, A.P., J.J.M., J.K., S.C., N.R., H.O., E.E.W., J.P., C.U., S.P., R.K.; Visualization, A.P., Se.Li.; Funding Acquisition, A.P., P.L., S.C., J.J.M., H.O., J.P., and C.U.

## Funding

## Competing interests
The authors declare no competing interests.

## Additional information

Pawel Lisowski[1,2,3,4,32], Selene Lickfett [5,6,7,32], Agnieszka Rybak-Wolf [2,8], Carmen Menacho [5,6], Stephanie Le [5,6], Tancredi Massimo Pentimalli [2,9,10], Sofia Notopoulou[11], Werner Dykstra[2,29], Daniel Oehler [12], Sandra López-Calcerrada[13], Barbara Mlody[2,30], Maximilian Otto [1,2], Haijia Wu[14], Yasmin Richter[15], Philipp Roth[1,2], Ruchika Anand [16], Linda A. M. Kulka [17], David Meierhofer[18], Petar Glazar[2,9,18], Ivano Legnini[2,9,31], Narasimha Swamy Telugu[2], Tobias Hahn[2], Nancy Neuendorf[2], Duncan C. Miller[2], Annett Böddrich[2], Amin Polzin[12], Ertan Mayatepek [6], Sebastian Diecke [2,19], Heidi Olzscha[14,17], Janine Kirstein [15,20], Cristina Ugalde [13,21,22], Spyros Petrakis [11], Sidney Cambridge[7,23], Nikolaus Rajewsky [2,9,19,24,25,26], Ralf Kühn [2], Erich E. Wanker [2], Josef Priller [3,26,27,28], Jakob J. Metzger [1,2,33] ✉ & Alessandro Prigione [2,6,33] ✉

[1]Quantitative Stem Cell Biology, Berlin Institute for Medical Systems Biology (BIMSB), Berlin, Germany. [2]Max Delbrück Center for Molecular Medicine in the Helmholtz Association (MDC), Berlin, Germany. [3]Department of Psychiatry and Psychotherapy, Neuropsychiatry and Laboratory of Molecular Psychiatry, Charité – Universitätsmedizin, Berlin, Germany. [4]Department of Molecular Biology, Institute of Genetics and Animal Biotechnology, Polish Academy of Sciences, Jastrzebiec n/Warsaw, Poland. [5]Faculty of Mathematics and Natural Sciences, Heinrich Heine University, Düsseldorf, Germany. [6]Department of General Pediatrics, Neonatology and Pediatric Cardiology, Medical Faculty, University Hospital Düsseldorf, Heinrich Heine University, Düsseldorf, Germany. [7]Institute of Anatomy II, Heinrich-Heine-University, Düsseldorf, Germany. [8]Organoid Platform, Berlin Institute for Medical Systems Biology (BIMSB), Berlin, Germany. [9]Laboratory for Systems Biology of Gene Regulatory Elements, Berlin Institute for Medical Systems Biology (BIMSB), Berlin, Germany. [10]Charité – Universitätsmedizin, Berlin, Germany. [11]Institute of Applied Biosciences (INAB), Centre For Research and Technology Hellas (CERTH), Thessaloniki, Greece. [12]Division of Cardiology, Pulmonology, and Vascular Medicine, Medical Faculty and University Hospital Düsseldorf, Cardiovascular Research Institute Düsseldorf (CARID), Düsseldorf, Germany. [13]Instituto de Investigación Hospital 12 de Octubre (i + 12), Madrid, Spain. [14]Institute of Molecular Medicine, Medical School, Hamburg, Germany. [15]Cell Biology, University of Bremen, Bremen, Germany. [16]Institute of Biochemistry and Molecular Biology I, Medical Faculty and University Hospital Düsseldorf, Heinrich Heine University, Düsseldorf, Germany. [17]Institute of Physiological Chemistry, Martin-Luther-University, Halle-Wittenberg, Germany. [18]Quantitative RNA Biology, Max Planck Institute for Molecular Genetics, Berlin, Germany. [19]German Center for Cardiovascular Research (DZHK), Berlin, Germany. [20]Leibniz Institute on Aging – Fritz-Lipmann-Institute, Jena, Germany. [21]Centro de Investigaciones Biológicas Margarita Salas (CIB-CSIC), Madrid, Spain. [22]Centro de Investigación Biomédica en Red de Enfermedades Raras (CIBERER), Madrid, Spain. [23]Dr. Senckenberg Anatomy, Anatomy II, Goethe-University, Frankfurt, Germany. [24]NeuroCure Cluster of Excellence, Berlin, Germany. [25]National Center for Tumor Diseases (NCT), German Cancer Consortium (DKTK), Berlin, Germany. [26]German Center for Neurodegenerative Diseases (DZNE), Berlin, Germany. [27]Department of Psychiatry and Psychotherapy; School of Medicine and Health, Technical University of Munich and German Center for Mental Health (DZPG), Munich, Germany. [28]University of Edinburgh and UK Dementia Research Institute, Edinburgh, UK. [29]Present address: Department of Translational Neuroscience, University Medical Center Utrecht Brain Center, Utrecht, The Netherlands. [30]Present address: Centogene, Rostock, Germany. [31]Present address: Human Technopole, Milan, Italy. [32]These authors contributed equally: Pawel Lisowski, Selene Lickfett. [33]These authors jointly supervised this work: Jakob J. Metzger, Alessandro Prigione. ✉e-mail: jakob.metzger@mdc-berlin.de; alessandro.prigione@hhu.de

