## [Peer Review File · Nature Communications]

Mutant huntingtin impairs neurodevelopment in human brain organoids through CHCHD2-mediated neurometabolic failureREVIEWER COMMENTS

Reviewer #1 (Remarks to the Author):

In this paper, the authors detail an HD phenotype observed in organoids of multiple lineages and also undirected differentiated organoids. They observe that HD homozygous organoids, both cortical and midbrain do not grow as well as WT. They find that HD homozygous undirected cerebral organoids show transcriptional changes related to mitochondrial function, including the gene CHCHD2 that is downregulated at both RNA and protein level. They go on to find that this protein is mis-localised in HD homozygous NPC and Cos and find that genes involved in mitochondrial integrated stress response (mISR) and mitochondrial dynamics genes are altered in the presence of mutant HTT, and that this mitochondrial footprint is increased in HD. Next, they use proteomics to demonstrate that the HD cells have higher demand for glucose and pyruvate, increased lactate production and decreased oxidative phosphorylation and increased glycolysis in HD homozygous cells. Finally, they use an inducible NGN2 protocol to make pure neurons with patient derived and control iPSCs. In these neurons, proteomics and transcriptomics show similar signatures to the organoids with metabolic processes being overrepresented in the differential proteomic analysis. Using SeaHorse assays, they show deficits in ATP production, max respiration, spare capacity, basal glycolysis and lactate levels in HD iPSC neurons. Using the same differentiation paradigm, they observe that siRNA mediated single knockdown of BDNF, CHCHD2 and HTT all elicited a similar defect of axonal and dendritic outgrowth in new neurons. This is a very interesting study with very rigorous analyses and an important message relating to bioenergetics and the contribution to neurodevelopment. However, there are some concerns to be addressed.

Concerns:

- 1) Supp 1b – Is the level of HTT expression different between the various conditions? Could this be quantified?
- 2) Is there a cerebral organoid growth deficit with HD homozygous mutation? – It seems odd to start with these models and then move to undirected differentiation organoids without much justification
- 3) It would be good to characterize the undirected cerebral organoids, are there some cells contributing to the signature more? And are these more cortical in nature?
- 4) Fig4A looks like there is more colocalization of CHCHD2 in the HD than the WT sample, could maybe use a more representative image for the WT
- 5) In the results, please describe a bit more the meaning of an increased footprint of mitochondria, as it is difficult to understand in real terms what the metric means.
- 6) The authors seem to have left out reference or any discussion to a paper by the HD iPSC consortium from 2020 that details bioenergetic deficits in an MSN enriched population derived from patient iPSC – specifically ATP, OCAR and ECAR as well as changes to protein expression.
- 7) It would be interesting to overexpress CHCHD2 in HD cells alongside the siRNA experiments – if proposing a therapeutic target you would want to be sure this is the mechanism and if it can be fully/partially rescued with overexpression of this gene.

8) A major weakness is how the authors describe production of the 0Q lines and state that there is rescue of phenotypes using the 0Q lines, but in reality, only look at the expression of the CHCHD2 gene and TOMM20. If the authors are going to cite this line and use it as a metric for rescue, then there needs to be additional evidence for phenotypic rescue and not just expression of two gene products. Further, some discussion of why this would rescue is needed given that the CAGs/Qs appear to be necessary for protein function, are evolutionarily conserved and may act as a “hinge” region in normal function.

9) Please discuss the fact that in this human organoid system, which is replicating features of normal brain development, the homozygous state is so much more severe, given that in patients, homozygotes are not significantly different than heterozygotes in terms of disease onset or severity. In mice, it is true you see more severe phenotypes, particularly in the juvenile repeat range, when homozygous. Perhaps that is the case here using a 70 repeat. However this should be directly discussed. A couple references are below:

Clinical manifestations of homozygote allele carriers in Huntington disease. Cubo E, Martinez-Horta SI, Santalo FS, Descalls AM, Calvo S, Gil-Polo C, Muñoz I, Llano K, Mariscal N, Diaz D, Gutierrez A, Aguado L, Ramos-Arroyo MA; European HD Network. *Neurology*. 2019 Apr 30;92(18):e2101-e2108.

Homozygotes for Huntington's disease.

Wexler NS, Young AB, Tanzi RE, Travers H, Starosta-Rubinstein S, Penney JB, Snodgrass SR, Shoulson I, Gomez F, Ramos Arroyo MA, et al. *Nature*. 1987 Mar 12-18;326(6109):194-7.

10) Another point for discussion would be that the authors posit that CHCHD2 reduction, given its effect on neuronal branching, could be an early target for intervention in HD. However, even treating early would not address a neurodevelopmental impairment. Please discuss.

Reviewer #2 (Remarks to the Author):

The manuscript by Lisowski et al investigates mHtt in various models including organoids and iPSC derived NPCs and neurons. The manuscript is well written and makes some interesting and novel observations about the role of Htt in development which may be pertinent to the development of HD and in particular when therapeutics should be given. The manuscript focuses on the role of CHCHD2 as this is found to be reduced and indeed restoration of CHCHD2 restores many of the changes observed in organoids. Whilst the manuscript could make a novel addition to the field, there are some limitations for the authors to address and some further mechanistic links to be made before CHCHD2 would be a validated therapeutic target for HD.

One major issue which the authors need to address is the data reported in Figures 2, 3 and 4 from the organoids. All of this data states that each dot on the graph represents one field of view across 2 differentiations. It is incredibly important as much of the data and indeed whole premise for the CHCHD2 observations is dependent on this data including size of organoids, all the staining and quantification for each genotype is based on 2 differentiations. This is critical to be repeated across

more differentiations and data presented in a way which enables statistical tests across multiple differentiations. In addition, it is important to be able to see which data represented is from which differentiation to see the variability between differentiations. In particular in Figure 4, the data is based on 10 images for 2 differentiations which is a very limited dataset to draw conclusions from. The omics datasets are very interesting and enhance the understanding and hypothesis of the metabolic changes occurring in various cell types at different stages. It would be important for the authors to validate some of these interpretations using functional techniques.

Next the authors turn to iPSC neuron model from HD patients with various CAG repeat lengths compared to controls. The data from this model in addition to the organoid model is a good addition to the manuscript and enables further interpretation of the data and role of mHtt in metabolism. Some more detail is required in Supp Fig 5, could the authors provide images from at least 1 control and 1 HD patient here. Furthermore, the images from Supp Fig 5c suggest HD1 may have a lower yield of neurons than the controls, could the authors provide some quantification of the yield from the 3 patients and controls.

In the omics dataset from iPSC's the authors comment that the HD patients (regardless of their disparate CAG repeat lengths) cluster together, which is true and reassuring for the changes occurring, however, the controls appear variable, could the authors comment on this.

In Figure 6, the Seahorse (and dendritic outgrowth) dataset is from at least 3 repeats per line, it would be interesting to see each control and HD patient line colour coded in order to assess the variability between differentiations of the same line and across lines.

For all the knockdown experiments it is important for the authors to show knockdown efficiency, it seems regardless of target of knockdown or BDNF withdrawal, the outgrowth is retarded to the same extent, is this the floor of the assay, could the authors comment? Some images of the dendritic outgrowth effect would also be important to include.

In particular the CHCHD2 knockdown experiments are important to dwell on some more, the observations made in the organoids would suggest CHCHD2 is pivotal, however, in neurons the authors state the expression level is low therefore any changes do not reach significance, however therefore the knockdown experiments need to be put in context of what CHCHD2 role could be in neurons and dataset showing expression and knockdown reached.

The NGN2 is added at NPC stage, so it will important to address CHCHD2 levels at the point of NGN2 addition, are the NPC's from HD patients already metabolically altered, could the authors measure this functionally in addition to in the neurons which develop from these NPC's. This would go some way to making the mechanistic link between mHtt and CHCHD2.

Overall, it is this mechanistic link between mHtt and CHCHD2 which is lacking in the manuscript. The manuscript would be greatly strengthened by the additional work enabling this clear link to be elucidated.

Reviewer #3 (Remarks to the Author):

In the manuscript "Mutant Huntingtin impairs neurodevelopment in human brain organoids through CHCHD2-mediated neurometabolic failure" the authors provide evidence that introducing biallelic 70Q disrupts the development of brain organoids, regardless of whether an unguided or

regionalized differentiation protocol is used. This disruption results in defective organization of neural progenitors. The authors investigate the regulatory mechanisms underlying these early developmental defects and identify the mitochondrial protein CHCHD2 as a top dysregulated factor responsible for the disruption of mitochondrial dynamics and bioenergetic alterations. Furthermore, the authors show that the elimination of the poly-Q tract in HTT reverses the defects in CHCHD2 expression and mitochondrial dynamics.

The manuscript presents novel and intriguing findings, highlighting the involvement of CHCHD2 in Huntington's disease (HD). These findings shed light on the potential of CHCHD2 as a target for early intervention in the disease.

These aspects significantly enhance the overall significance of the study within the field. Although the manuscript shows potential for publication, the conclusions are only partially supported by the results. To strengthen the scientific impact of the study, it is crucial to provide a comprehensive and detailed characterization of the brain organoids employed. Additionally, the authors establish a correlation between impaired mitochondrial function alterations, and dysregulation of CHCHD2. However, a direct and definitive link between CHCHD2 dysregulation and the observed neurodevelopmental defects in the brain organoids is not clearly shown, leading to a weakened message and a decrease in initial enthusiasm.

Major comments:

- 1) In order to draw firm conclusions regarding the impairment of NPC cytoarchitecture, the authors should perform an analysis of the apico-basal polarity organization of neuroepithelial progenitors within the 70Q/70Q cerebral organoids (e.g. PAX6, ZO1, E-CAD staining).
- 2) It is unclear whether the 70Q/70Q brain organoids show a defect in spatial organization or if there is an overall downregulation of NPC markers. The authors should provide a comprehensive analysis of rosette formation (e.g., PAX6, SOX2, FOXG1) at different time points during cerebral organoid differentiation. This detailed analysis will contribute to understanding the developmental processes and potential disruptions in rosette structure formation within the 70Q/70Q brain organoids.
- 3) The authors are encouraged to thoroughly investigate the internal organizational structure of cortical organoids by performing a detailed analysis and comparison of the formation of distinct cortical layers between the control and mutant samples. This analysis would provide valuable insights into the impact of the 70Q/70Q on cortical layer formation.
- 4) The authors have observed a reduction in the size of the brain organoids, regardless of the differentiation protocols employed. To provide further insights, it is recommended that the authors conduct additional experiments to investigate whether this reduction in size is attributed to a decrease in the neural progenitor pool or an accelerated differentiation process. This clarification would contribute to a better understanding of the cellular dynamics and fate determination within the brain organoids, thereby strengthening the significance of the findings.
- 5) The authors claim in the Discussion "Under physiological conditions, cortical organoids typically reach a much larger volume over time than midbrain organoids, which instead remain relatively

small and are kept under static growth”. The authors may consider to refer in the manuscript the studies of Fiorenzano et al. in Nature Communications, 2021, or Jo et al. in Cell Stem Cell, 2017, where midbrain organoids were successfully cultured on an orbital shaker, resulting in the generation of functionally mature and pigmented dopaminergic neurons over a long-term period.

6) To better understand the impact of developmental alterations in different brain regions, it is recommended that the authors perform a more detailed analysis of midbrain organoids. Specifically, they should quantify the generation of floor plate rosettes (marked by e.g. FOXA2, SOX2, ZO-1, E-CAD) at early time points and assess the presence of TH+ neurons in long-term cultures within mutant midbrain organoids.

7) While the authors' efforts are commendable, they have not provided sufficient evidence to establish the mechanism linking metabolic impairment resulting from CHCHD2 reduction with early neurodevelopmental defects (i.e. spatial organization and molecular impairment) in organoids. Given the importance of the findings, it is suggested that the authors utilize the available transcriptomic datasets to provide a mechanistic explanation for the observed defects.

Reviewer #4 (Remarks to the Author):

Lisowski et al used stem cells to study the impact of mutant Huntingtin on neurodevelopment. Introducing a mutant expansion led to abnormal development of cerebral organoids and dysregulation of the CHCHD2 protein. CHCHD2 repression was associated with mitochondrial defects and increased energy expenditure. Removing the expansion normalised CHCHD2 expression and mitochondrial function. These findings suggest that targeting CHCHD2 could be an early intervention for Huntington's disease.

I have a few queries that I'd appreciate being clarified within the manuscript as a revised version:

1. There is a contradiction on line 850 where authors wrote 'profiling by targeted LC-MS'. By definition, metabolic profiling is an untargeted exercise, to maximise coverage and generate a profile of metabolites without any targets in mind.
2. Lines 856-857 are pre-mature and renders the method irreproducible. The extracts once aliquot, were subjected to addition of internal standards and then dried in SpeedVac. The pellets were reconstituted in solution described on line 857, not the aliquots themselves.
3. It is unclear what 400 metabolites were selected and how they were chosen.
4. Why were all 400 metabolites normalised to protein abundances obtained from the BCA assay? Are all 400 associated with the proteins measured? The authors should cite if there is an association between the selected 400 metabolites and measured proteins.

5. Supplementary Data 9 contains only 128 metabolites and their associated data. How did authors reduce variables from 400 to 128? Student's t-test columns are empty for these 128 variables and q-values are all greater than 0.05.

6. What quality control was in place to ensure that metabolites measured were free from analytical errors?

7. Functional proteome-metabolome data are interesting and important for this work. How did the authors ensure that measured proteome and metabolome reflects the cellular biomass that was quenched and extracted correctly?

REVIEWER COMMENTS

Reviewer #1 (Remarks to the Author):

In this paper, the authors detail an HD phenotype observed in organoids of multiple lineages and also undirected differentiated organoids. They observe that HD homozygous organoids, both cortical and midbrain do not grow as well as WT. They find that HD homozygous undirected cerebral organoids show transcriptional changes related to mitochondrial function, including the gene CHCHD2 that is downregulated at both RNA and protein level. They go on to find that this protein is mis-localised in HD homozygous NPC and Cos and find that genes involved in mitochondrial integrated stress response (mISR) and mitochondrial dynamics genes are altered in the presence of mutant HTT, and that this mitochondrial footprint is increased in HD. Next, they use proteomics to demonstrate that the HD cells have higher demand for glucose and pyruvate, increased lactate production and decreased oxidative phosphorylation and increased glycolysis in HD homozygous cells. Finally, they use an inducible NGN2 protocol to make pure neurons with patient derived and control iPSCs. In these neurons, proteomics and transcriptomics show similar signatures to the organoids with metabolic processes being overrepresented in the differential proteomic analysis. Using Seahorse assays, they show deficits in ATP production, max respiration, spare capacity, basal glycolysis and lactate levels in HD iPSC neurons. Using the same differentiation paradigm, they observe that siRNA mediated single knockdown of BDNF, CHCHD2 and HTT all elicited a similar defect of axonal and dendritic outgrowth in new neurons. This is a very interesting study with very rigorous analyses and an important message relating to bioenergetics and the contribution to neurodevelopment. However, there are some concerns to be addressed.

Concerns:

1) Supp 1b – Is the level of HTT expression different between the various conditions? Could this be quantified?

We thank the reviewer for noticing that the quantification was missing. We quantified the immunoblot images and included the results (**Supplementary Fig. 1c**).

2) Is there a cerebral organoid growth deficit with HD homozygous mutation? – It seems odd to start with these models and then move to undirected differentiation organoids without much justification

We agree that growth rate in cerebral organoids needed to be investigated. We performed the experiments using three independent replicates and detected a decreased growth rate for mutant cerebral organoids. We included these important results (**Fig. 1f**) and thank the reviewer for this suggestion.

3) It would be good to characterize the undirected cerebral organoids, are there some cells contributing to the signature more? And are these more cortical in nature?

We included additional stainings of the unguided cerebral organoids. In particular, we found that the expression of ZO1 was disrupted in mutant organoids (**Fig. 1e**). We think that these findings are quite interesting, as they recapitulate defects in progenitor organization and ZO1 expression that have been observed in human fetuses carrying mHTT (Barnat et al, Science, 2020).

However, mutant cerebral organoids were so severely disrupted that it was hard to characterize them in more details. This is the reason why we then focused on guided organoids, which are smaller and could

differentiate more robustly in our hands. We have further characterized these guided organoids and included additional stainings, and also performed single-cell transcriptomics on midbrain organoids, which confirmed specific defects in the progenitor population (**Fig. 2d-g, Supplementary Fig. 3b,d,e-f**).

4) Fig4A looks like there is more colocalization of CHCHD2 in the HD than the WT sample, could maybe use a more representative image for the WT

We have replaced the images with more representative ones (**Fig. 4a-b**). In fact, we believe that it is not only the colocalization that is altered but that the overall expression of CHCHD2 is reduced. We also included images of midbrain organoids (**Supplementary Fig. 5a**). We also performed immunoblot analysis confirming the reduction of CHCHD2 in 70Q/70Q NPCs (**Supplementary Fig. 4e-f**).

5) In the results, please describe a bit more the meaning of an increased footprint of mitochondria, as it is difficult to understand in real terms what the metric means.

We apologize for the lack of clarity. We observed an increase of both smaller and larger mitochondrial structures in the 70Q/70Q compared to WT/WT organoids. We termed this feature as mitochondrial footprint. We modify the text to better describe this feature (**lines 303-313**), and included quantification of both small mitochondrial structures and large mitochondrial structures to show that both are increased in mutant cerebral organoids (**Supplementary Fig. 5f**).

Additionally, we carried out electron microscopy of NPCs from WT/WT, 70Q/70Q and 0Q/0Q. This analysis confirmed the disruption of mitochondrial morphology with a particular aberration in cristae directionality in 70Q/70Q NPCs that was rescued in 0Q/0Q NPCs (**Fig. 5a-b**). Furthermore, by performing long-read transcriptomics, we identified specific differences in isoform usage in genes regulating mitochondrial dynamics (**Fig. 4d, Supplementary Fig. 5b-d**). These included also DNMT1L, whose variants have been linked to altered mitochondrial dynamics (Nolden et al, Life Science All, 2022). Lastly, we carried out SDS-PAGE and blue native PAGE analysis to investigate mitochondrial complex assembly and composition and identified a particular defect in mitochondrial complex IV (**Fig. 5g-j, Supplementary Fig 6h**). Gene expression analysis of respiratory chain components and assembly factors in mutant NPCs, organoids, and neuruloids further confirmed these alterations (**Supplementary Fig 6i-j**). Altogether, we believe that these data point towards a dysregulation of mitochondrial morphology and dynamics in human neural cells caused by mHTT.

6) The authors seem to have left out reference or any discussion to a paper by the HD iPSC consortium from 2020 that details bioenergetic deficits in an MSN enriched population derived from patient iPSC – specifically ATP, OCAR and ECAR as well as changes to protein expression.

We thank the reviewer for noticing this. We included this important work (**reference number 66**) and commented it in the discussion section (**lines 537-538**).

7) It would be interesting to overexpress CHCHD2 in HD cells alongside the siRNA experiments – if proposing a therapeutic target you would want to be sure this is the mechanism and if it can be fully/partially rescued with overexpression of this gene.

We performed additional experiment to address this relevant aspect. We generated AAV constructs carrying either GFP alone or CHCHD2-GFP and used them to transduce NPCs carrying mHTT (line 70Q/70Q). Mutant NPCs transduced with CHCHD2-GFP showed an increase of CHCHD2 and also a normalization of the TOM20 signal similar to WT levels (**Fig. 7f-g, Supplementary Fig. 9c-d**). These data are in agreement with a recent publication suggesting that increasing CHCHD2 may be beneficial in HD cells to protect against oxidative stress (Liu et al, Cell Death and Disease 2024).

8) A major weakness is how the authors describe production of the 0Q lines and state that there is rescue of phenotypes using the 0Q lines, but in reality, only look at the expression of the CHCHD2 gene and TOMM20. If the authors are going to cite this line and use it as a metric for rescue, then there needs to be additional evidence for phenotypic rescue and not just expression of two gene products. Further, some discussion of why this would rescue is needed given that the CAGs/Qs appear to be necessary for protein function, are evolutionarily conserved and may act as a “hinge” region in normal function.

We agree with the reviewer that it was important to perform additional analyses with 0Q/0Q cells. We have included several new experiments with these cells: 1) immunoblot quantification of CHCHD2 (**Supplementary Fig. 4e-f**), 2) electron microscopy analysis of mitochondrial features (**Fig. 5a-b, Supplementary Fig. 5g**), 3) SDS-PAGE and blue native PAGE to monitor mitochondrial respiratory chain complex assembly (**Fig. 5g-j, Supplementary Fig 6h**), 4) expression analysis of respiratory chain components and assembly factors (**Supplementary Fig 6i-j**), 5) qPCR analysis of core network components related to the Hippo signaling pathways that we identified following a new multi-omics integration of HD neurons (**Fig. 7b**). Most of the studied aspects were rescued in 0Q/0Q cells, including mitochondrial cristae defects (**Fig. 5a-b**) and respiratory chain assembly defects (**Fig. 5g-j**). At the same time, some features were not rescued, such as the expression of the core network genes (**Fig. 7b**). We also noticed some changes in mitochondrial roundness and mitochondrial area that were significantly different between WT/WT cells and 0Q/0Q, but not significantly altered in 70Q/70Q compared to WT/WT (**Supplementary Fig. 5g**). We therefore included a section in the discussion to comment on these aspects to advise that more data would be required to fully address the impact of CAG elimination at the level of iPSCs or at the level of post-mitotic neurons using somatic gene correction (**lines 625-631**).

9) Please discuss the fact that in this human organoid system, which is replicating features of normal brain development, the homozygous state is so much more severe, given that in patients, homozygotes are not significantly different than heterozygotes in terms of disease onset or severity. In mice, it is true you see more severe phenotypes, particularly in the juvenile repeat range, when homozygous. Perhaps that is the case here using a 70 repeat. However this should be directly discussed. A couple references are below:

Clinical manifestations of homozygote allele carriers in Huntington disease. Cubo E, Martinez-Horta SI, Santalo FS, Descalls AM, Calvo S, Gil-Polo C, Muñoz I, Llano K, Mariscal N, Diaz D, Gutierrez A, Aguado L, Ramos-Arroyo MA; European HD Network. *Neurology*. 2019 Apr 30;92(18):e2101-e2108.

Homozygotes for Huntington's disease. Wexler NS, Young AB, Tanzi RE, Travers H, Starosta-Rubinstein S, Penney JB, Snodgrass SR, Shoulson I, Gomez F, Ramos Arroyo MA, et al. *Nature*. 1987 Mar 12-18;326(6109):194-7.

Thank you for providing this relevant input. We modified the text. We now do not describe anymore homozygous forms as more severe forms. Instead, we introduced homozygous cells as a model in which the impact of WT HTT is not present (**lines 160-166**). We also included the suggested references (**reference ...**). Perhaps the lack of changes observed in heterozygous cells was only a feature of our specific model. Indeed, in our WT/70Q cells the HTT expression was not significantly reduced (**Supplementary Fig. 1c**), and therefore there were no changes in CHCHD2 expression (**Fig. 3g**) or in TOM20 levels (**Fig. 4g**). We comment on these aspects in discussion to conclude that further work is needed to dissect the differences between heterozygous and homozygous genotypes (**lines 616-624**).

10) Another point for discussion would be that the authors posit that CHCHD2 reduction, given its effect on neuronal branching, could be an early target for intervention in HD. However, even treating early would not address a neurodevelopmental impairment. Please discuss.

We agree that “early intervention” is not an easy approach to take into practice. As fetuses carrying mutant HTT may already show defects (Barnat et al, Science, 2020), it remains unclear how “early” such intervention should take place. Nonetheless, given the positive effect of CHCHD2 overexpression in rescuing the mitochondrial phenotype in mutant neural cells (**Fig. 7f-g**) and the recent data suggesting CHCHD2 overexpression as a way to protect against oxidative stress in HD cells (Liu et al, Cell Death and Disease 2024), we believe that CHCHD2 could indeed represent an interesting interventional target. Moreover, we think that highlighting the relevance of early intervention may also be important when considering other treatment paradigms for HD. We commented on these aspects in the discussion (**lines 561-573**).

Reviewer #2 (Remarks to the Author):

The manuscript by Lisowski et al investigates mHtt in various models including organoids and iPSC derived NPCs and neurons. The manuscript is well written and makes some interesting and novel observations about the role of Htt in development which may be pertinent to the development of HD and in particular when therapeutics should be given. The manuscript focuses on the role of CHCHD2 as this is found to be reduced and indeed restoration of CHCHD2 restores many of the changes observed in organoids. Whilst the manuscript could make a novel addition to the field, there are some limitations for the authors to address and some further mechanistic links to be made before CHCHD2 would be a validated therapeutic target for HD.

We agree that additional experiments were needed to prove the relevance of CHCHD2 in HD cells. We demonstrated that increasing CHCHD2 expression in human neural cells carrying mHTT led to a normalization of the TOM20 signal similar to WT levels (**Fig. 7f-g, Supplementary Fig. 9c-d**). These data are in agreement with a recent publication suggesting that increasing CHCHD2 may be beneficial in HD cells to protect against oxidative stress (Liu et al, Cell Death and Disease 2024). We also identified that CHCHD2 belonged to a core network of dysregulated transcripts/proteins/metabolites that was associated with neurodevelopment and Hippo signaling in HD patient-derived neurons. qPCR analysis of neural progenitor cells engineered to carry mHTT confirmed altered expression of key members of the multi-

omics core network further demonstrating its relevance (**Fig. 7b**). Altogether, we think that our additional experiments strengthen our findings and indicate CHCHD2 as a potential interventional target in HD associated with mitochondrial defects that may occur in mutant cells during neurodevelopment.

One major issue which the authors need to address is the data reported in Figures 2, 3 and 4 from the organoids. All of this data states that each dot on the graph represents one field of view across 2 differentiations. It is incredibly important as much of the data and indeed whole premise for the CHCHD2 observations is dependent on this data including size of organoids, all the staining and quantification for each genotype is based on 2 differentiations. This is critical to be repeated across more differentiations and data presented in a way which enables statistical tests across multiple differentiations. In addition, it is important to be able to see which data represented is from which differentiation to see the variability between differentiations. In particular in Figure 4, the data is based on 10 images for 2 differentiations which is a very limited dataset to draw conclusions from.

We agree that robustness and transparency are crucial. We improved the figure legends to describe the replications and number of individual images in more details.

For organoid experiments, one key variability may indeed come from intraexperimental heterogeneity, given that individual organoids may exhibit variable size and features. This is the reason why for growth rate curves we showed all individual organoids (**Fig. 2b-c**) so that the actual distribution of all individual organoids can be seen.

For single-cell transcriptomics of midbrain organoids, we used biological triplicates. But what is important to highlight is that each RNA sample was made of 48 individual midbrain organoids pooled together, thereby increasing the robustness of the results. Moreover, the scRNAseq experiments were performed in two separate batches (in one batch we used 2 replicates for WT and 2 replicates for mutants, and in other batch we used 1 replicate for WT and 1 for mutants). We described this approach in the methods section (**lines 1037-1040**). Despite this division into two batches, and the fact that each replicate contained several individual organoids, the 3 biological replicates showed very similar profiles, both in the case of WT and in the case of mutants (**Supplementary Fig. 3e-f**). This fact further underscores the robustness of the guided organoid differentiations.

For the immunostaining images in **Figure 4**, we used indeed many more images than 10 (for NPCs at least 35 individual images per sample out of at least two biological replicates, for cerebral organoids at least 77 individual images per sample out of at least two biological replicate). We included this information in the respective figure legends. What is important to point out is that this imaging quantification was not influenced by the operator, since it was based on an automated pipeline that we established and explained the methods section (**lines 1349-1358**). We believe that the use of operator-free procedures increases the reliability of the data.

Furthermore, we included orthogonal experimental approaches to validate the findings using different approaches. For CHCHD2 expression, we used transcriptomics (**Fig. 3b,e**), proteomics (**Fig. 3d**), immunostaining (**Fig. 3f-g, Fig. 4a-b**), and western blotting (**Supplementary Fig. 4e-f**). Findings in HD patient-derived neurons confirmed that CHCHD2 belongs to a core network of dysregulated factors (**Fig. 7a, Supplementary Fig. 8d**), as members of the CHCH domain-containing proteins (CHCHD1, CHCHD2, and CHCHD5) were part of the biggest cluster of co-expressed proteins (**Supplementary Fig. 8a**). Finally, the

use of siRNA knock-down (**Fig. 7d-e**) and AAV-mediated overexpression (**Fig. 7f-g**) validated the relevance of modulating CHCHD2 levels in HD neuropathology.

For mitochondrial phenotypes, we used immunostaining (**Fig. 4e-g**), electron microscopy (**Fig. 5a-b**), Seahorse profiling (**Fig. 6f-g**), and respiratory chain complex assembly analysis with SDS PAGE (**Fig. 5g-h**) and blue native PAGE (**Fig. 5i-j**). Long-read transcriptomics (**Fig. 4d**) and proteomics-based metabolic state analysis (**Fig. 5c-f**) confirmed mitochondrial defects.

In fact, we confirmed our findings using orthogonal assays. For example: 1) to claim that we see an impact of mHTT on neurodevelopment we employed three different brain organoid protocols; 2) to study the effect of mHTT on human neural cells we used both genome engineered lines and patient-derived lines; 3) to claim an effect on CHCHD2 expression we employed gene expression, proteomics, immunostaining, multi-omics integration, immunoblotting, and AAV-mediated overexpression; 4) to suggest an effect on neural progenitors we used both 3D organoids and 2D NPC cultures and assessed them using multi-omics, single-cell RNA sequencing, immunostaining and qPCR; 5) to claim an effect on mitochondrial morphology and dynamics we used confocal microscopy, electron microscopy, long-read sequencing, gene expression, and immunoblotting. Therefore, we believe that our data are solid and that the conclusions are built on several layers of complementary evidence.

Taken together, we think that our conclusions are based on findings that are the results of a wide range of complementary analyses, and are therefore sufficiently robust and cross-validated. Indeed, a recent paper also described the importance of CHCHD2 overexpression in protecting against oxidative stress in HD (Liu et al, Cell Death and Disease 2024). This fact further underscores the robustness and reproducibility of our data.

The omics datasets are very interesting and enhance the understanding and hypothesis of the metabolic changes occurring in various cell types at different stages. It would be important for the authors to validate some of these interpretations using functional techniques.

We included several additional experiments in the revised version to strengthen our claims: 1) growth rate assessment of cerebral organoids confirming the defects seen for guided organoids (**Fig. 1f**); 2) single-cell RNA sequencing of midbrain organoids confirming the defect in neural progenitor population (**Fig. 2e-g**); 3) long-read transcriptomics confirmed that identified defects in variant use for genes related to mitochondrial dynamics (**Fig. 4d**); 4) electron microscopy confirmed dysfunction of mitochondrial and cristae morphology (**Fig. 5 a-b**); 5) SDS PAGE and blue native PAGE validated the presence of mitochondrial defects and particularly identified impairment on complex IV (**Fig. 5g-j**); 6) immunoblot analysis and quantifications of CHCHD2 (**Supplementary Fig. 4e-f**) and HTT (**Supplementary Fig. 1c**) confirmed defects observed by omics and immunostainings; 7) multi-omics integration in patient-derived neurons led to the identification of a core network for mHTT with which CHCHD2 is closely associated (**Fig. 7a**); 8) qPCR analysis validated that the expression of key members of the multi-omics core network linked to Hippo signaling showing that these were defective also in engineered mutant neural cells (**Fig. 7b**); 9) AAV-based overexpression of CHCHD2 confirmed that it is capable of rescuing the mitochondrial phenotype in mutant neural cells (**Fig. 7f-g**).

Next the authors turn to iPSC neuron model from HD patients with various CAG repeat lengths compared to controls. The data from this model in addition to the organoid model is a good addition to the

manuscript and enables further interpretation of the data and role of mHtt in metabolism. Some more detail is required in Supp Fig 5, could the authors provide images from at least 1 control and 1 HD patient here.

We included images of NGN2 neurons derived from all three controls (C1, C2, C3) and all three HD patients (HD1, HD2, HD3) (**Supplementary Fig. 7c**). In addition, we performed an integrated multi-omics analysis of NGN2 datasets that allowed us to gain further insights into the role of CHCHD2 in the network of altered genes/proteins/metabolites in HD patient-derived neurons (**Figure 7a-c, Supplementary Fig. 8a-e**).

Furthermore, the images from Supp Fig 5c suggest HD1 may have a lower yield of neurons than the controls, could the authors provide some quantification of the yield from

We could not identify any significant defects in neuronal generation caused by mHTT. Even in mutant cerebral organoids there were neuronal cells visible (**Fig. 1e, Supplementary Fig. 2a-b**). In fact, single-cell RNAseq analysis showed that mature neuronal populations were even increased in mutant midbrain compared to controls (**Fig. 2e-f, Supplementary Fig. 3e-f**). Accordingly, NGN2 transcriptomics and proteomics identified pathways associated with synaptic activity and axon guidance as significantly upregulated in HD neurons compared to controls (**Fig. 6c-d**). Hence, it is possible that the defects in neural progenitor function may result in aberrant and premature neuronal differentiation. This feature is in agreement with data described in human fetuses carrying mHTT (Barnat et al, Science, 2020). We included comments on these aspects in the discussion (**lines 561-573**).

In the omics dataset from iPSC's the authors comment that the HD patients (regardless of their disparate CAG repeat lengths) cluster together, which is true and reassuring for the changes occurring, however, the controls appear variable, could the authors comment on this.

We do not have a clear explanation why control neurons appear to show increased variability. Perhaps this is due to the inherent heterogeneity of individual lines, while the presence of mHTT mutation may cause dramatic changes leading the cells to acquire a specific signature. Nonetheless, we believe that control neurons clustered still sufficiently separately from HD neurons (**Supplementary Fig. 7d**), indicating to us reliable findings. In fact, key genes belonging to the core of the dysregulated multi-omics network identified in HD-patient neurons were also altered in edited cells carrying mHTT (**Fig. 7b**). This suggested that the omics datasets are valid and could lead to results that were validated also in a different system.

In Figure 6, the Seahorse (and dendritic outgrowth) dataset is from at least 3 repeats per line, it would be interesting to see each control and HD patient line color coded in order to assess the variability between differentiations of the same line and across lines.

We colored the dots of the Seahorse graphs to highlight the three differentiation sets (**Fig. 6f-g**). We are however unsure how much clearer are the data now. We have not done the same for the branching graphs, as we were worried that it would complicate the understanding of the figure (**Fig. 7e**). Nevertheless, we specified that the data were performed as three independent experiments in the figure legends.

For all the knockdown experiments it is important for the authors to show knockdown efficiency, it seems regardless of target of knockdown or BDNF withdrawal, the outgrowth is retarded to the same extent, is

this the floor of the assay, could the authors comment? Some images of the dendritic outgrowth effect would also be important to include.

We included representative images showing the presence of multiple branching in untreated neurons and reduced neurite presence in neurons in which BDNF or CHCHD2 were knocked down (**Supplementary Fig. 9a**). We in fact believe that this could be the floor of the assay. Indeed, BDNF represented our positive control, so what we can conclude is that CHCHD2 knock-down led to similar effects as BDNF knock-down. We cannot exclude that knock-down of other genes may lead to even lower branches amount. However, in order to address the reviewer's concern, we included the knock-down of VEGFA which led only to a small reduction of dendritic length (**Supplementary Fig. 9b**). We believe that this might suggest that the assay captures different changes and should be capable of distinguishing between milder and stronger modulators of neurite length.

In particular the CHCHD2 knockdown experiments are important to dwell on some more, the observations made in the organoids would suggest CHCHD2 is pivotal, however, in neurons the authors state the expression level is low therefore any changes do not reach significance, however therefore the knockdown experiments need to be put in context of what CHCHD2 role could be in neurons and dataset showing expression and knockdown reached.

We agree that our findings in NGN2 neurons were not clear, and we thank the reviewer for pointing out this. We performed a completely new and comprehensive analysis of all datasets for HD patient-derived neurons, through multi-omics integration and co-expression analysis (**Fig. 6c-e, Fig. 7a,c, Supplementary Fig. 7e-j, Supplementary Fig. 8a-e**). These new data provide a much more detailed description of the changes occurring in HD neurons. Importantly, the integration analysis showed that CHCHD2 (together with CHCHD10) is located closely to the core of the network of genes/proteins/metabolites that is dysregulated by mHTT (**Fig. 7a, Supplementary Fig. 8d**). These findings confirmed the importance of CHCHD2 in neurons and its dysregulation in HD. Moreover, CHCHD2 belonged also to the biggest cluster of co-expressed proteins in HD neurons based on protein co-expression network analysis (**Supplementary Fig. 8a**). We believe that these new data further support our evidence pointing towards a role for CHCHD2 in the neuropathology of HD.

The NGN2 is added at NPC stage, so it will important to address CHCHD2 levels at the point of NGN2 addition, are the NPC's from HD patients already metabolically altered, could the authors measure this functionally in addition to in the neurons which develop from these NPC's. This would go some way to making the mechanistic link between mHTT and CHCHD2. Overall, it is this mechanistic link between mHTT and CHCHD2 which is lacking in the manuscript. The manuscript would be greatly strengthened by the additional work enabling this clear link to be elucidated.

We thank the reviewer for raising these important issues. Regarding the NGN differentiation, we chose in fact to start from NPCs exactly because we wanted to test whether neurons can be generated even in the presence of possible NPC defects. We believe that using NGN2 neurons directly generated from iPSCs would have been less similar to the actual situation happening during development *in vivo* and would not have been as informative to address our questions. To clarify this reasoning, we included additional explanations in the text (**lines 398-407**).

With respect to adding experiments to demonstrate NPC dysfunction caused by mHTT, we believe that we now have several lines of experimental evidence showing NPC impairment by mHTT. We performed

experiments in NPCs as 2D culture, including immunostaining, immunoblotting, SDS-PAGE and blue native PAGE analysis, electron microscopy, proteomics-based functional metabolic analysis, and long-read transcriptomics (**Figure 3 and Figure 4**). We also have evidence of progenitor defects within unguided cerebral organoids as seen by immunostaining and gene expression (**Figure 1**) and in midbrain organoids as seen by single-cell RNA sequencing (**Figure 2**).

Regarding the CHCHD2 link to neurodevelopmental and mitochondrial defects, we demonstrate: 1) CHCHD2 expression is reduced across the neurodevelopmental stages in cells expressing mHTT, as seen by transcriptomics (**Fig. 3b,e**); 2) CHCHD2 expression is reduced in mutant NPCs as seen by proteomics (**Fig. 3d**), by immunostaining (**Fig. 3g, Fig. 4b**), and by immunoblotting (**Supplementary Fig. 4e-f**); 3) CHCHD2 expression is altered in HD neuruloids (**Fig. 3e**); 4) CHCHD2 expression is reduced in mutant cerebral organoids by immunostaining (**Fig. 4a**); 5) CHCHD2 expression is reduced in midbrain organoids (**Supplementary Fig. 5a**); 6) CHCHD2 knock-down impairs neurite growth capacity in human neurons (**Fig. 7e**); 7) CHCHD2 overexpression decrease the aberrant mitochondrial signal in human NPCs carrying mHTT (**Figure 7f-g**); 8) CHCHD2 belongs to a network of dysregulated factors in HD human neurons that is associated with axon guidance, mitochondrial integrated stress response, and Hippo signaling (**Fig. 7a-c**).

Reviewer #3 (Remarks to the Author):

In the manuscript "Mutant Huntingtin impairs neurodevelopment in human brain organoids through CHCHD2-mediated neurometabolic failure" the authors provide evidence that introducing biallelic 70Q disrupts the development of brain organoids, regardless of whether an unguided or regionalized differentiation protocol is used. This disruption results in defective organization of neural progenitors. The authors investigate the regulatory mechanisms underlying these early developmental defects and identify the mitochondrial protein CHCHD2 as a top dysregulated factor responsible for the disruption of mitochondrial dynamics and bioenergetic alterations. Furthermore, the authors show that the elimination of the poly-Q tract in HTT reverses the defects in CHCHD2 expression and mitochondrial dynamics.

The manuscript presents novel and intriguing findings, highlighting the involvement of CHCHD2 in Huntington's disease (HD). These findings shed light on the potential of CHCHD2 as a target for early intervention in the disease. These aspects significantly enhances the overall significance of the study within the field. Although the manuscript shows potential for publication, the conclusions are only partially supported by the results. To strengthen the scientific impact of the study, it is crucial to provide a comprehensive and detailed characterization of the brain organoids employed.

We thank the reviewer for raising this important aspect. We performed several new experiments to strengthen the findings and support our conclusions. With respect to brain organoids, we included additional results: 1) immunostaining showing aberrant ZO1 expression in mutant unguided cerebral organoids (**Fig. 1e**); 2) growth rate analyses of unguided cerebral organoids showing defective growth caused by mHTT (**Fig. 1f**); 3) immunostaining of cortical organoids (**Supplementary Fig. 3b**) and midbrain organoids (**Fig. 2d, Supplementary Fig. 3d**) showing that they express the correct identity markers; 4) immunostaining of midbrain organoids showing defective CHCHD2 expression (**Supplementary Fig. 5a**); 5) single-cell RNA sequencing of midbrain organoids demonstrating the impact of mHTT on neural progenitor populations (**Fig. 2e-g, Supplementary Fig. 3e-f**). We also included additional data in 2D to address the functionality of mutant NPCs. We believe that altogether our data comprehensively show that mHTT

impacts human neurodevelopment and neural progenitor function and that CHCHD2 plays a contributory role to this.

Additionally, the authors establish a correlation between impaired mitochondrial function alterations, and dysregulation of CHCHD2. However, a direct and definitive link between CHCHD2 dysregulation and the observed neurodevelopmental defects in the brain organoids is not clearly showed, leading to a weakened message and a decrease in initial enthusiasm.

We agree that our previous findings were not conclusive enough. We now have performed several additional experiments. Regarding the CHCHD2 link to neurodevelopmental and mitochondrial defects, we demonstrate: 1) CHCHD2 expression is reduced across the neurodevelopmental stages in cells expressing mHTT, as seen by transcriptomics (**Fig. 3b,e**); 2) CHCHD2 expression is reduced in mutant NPCs as seen by proteomics (**Fig. 3d**), by immunostaining (**Fig. 3g, Fig. 4b**), and by immunoblotting (**Supplementary Fig. 4e-f**); 3) CHCHD2 expression is altered in HD neuruloids (**Fig. 3e**); 4) CHCHD2 expression is reduced in mutant cerebral organoids by immunostaining (**Fig. 4a**); 5) CHCHD2 expression is reduced in midbrain organoids (**Supplementary Fig. 5a**); 6) CHCHD2 knock-down impairs neurite growth capacity in human neurons (**Fig. 7e**); 7) CHCHD2 overexpression decreases the aberrant mitochondrial signal in human NPCs carrying mHTT (**Figure 7f-g**); 8) CHCHD2 belongs to a network of dysregulated factors in HD human neurons that is associated with axon guidance, mitochondrial integrated stress response, and Hippo signaling (**Fig. 7a-c**).

Major comments:

1) In order to draw firm conclusions regarding the impairment of NPC cytoarchitecture, the authors should perform an analysis of the apico-basal polarity organization of neuroepithelial progenitors within the 70Q/70Q cerebral organoids (e.g. PAX6, ZO1, E-CAD staining).

We thank the reviewer for this suggestion. We performed stainings of ZO1 and SOX2 of cerebral organoids. In particular, we found that the expression of ZO1 was disrupted in mutant organoids (**Fig. 1e**). We think that these findings are quite interesting, as they recapitulate defects in progenitor organization and ZO1 expression that have been observed in human fetuses carrying mHTT (Barnat et al, Science, 2020). However, due to the strong disruption of the 70Q/70Q organoids, these did not show rosette formation and thus it was difficult to clearly analyze the apico-basal polarity of individual cells. The images did show though that WT/WT organoids exhibited a clear organization of the neuroepithelial progenitors while in the 70Q/70Q organoids this clear organization is disrupted (**Fig. 1e, Supplementary Fig. 2a**). Furthermore, by performing single-cell RNA sequencing of midbrain organoids, we were able to confirm clear defects in the neural progenitor population (**Fig. 2e-g, Supplementary Fig. 3e-f**).

Moreover, we included additional data in 2D NPC culture to further confirm the presence of functional defects in the NPC population: 1) long-read transcriptomics that identified defects in variant use for genes related to mitochondrial dynamics (**Fig. 4d**); 2) electron microscopy confirmed dysfunction of mitochondrial and cristae morphology (**Fig. 5 a-b**); 3) SDS PAGE and blue native PAGE to validate the impact on mitochondrial respiration assembly (**Fig. 5g-j**); 4) immunoblot analysis and quantifications of CHCHD2 (**Supplementary Fig. 4e-f**).

2) It is unclear whether the 70Q/70Q brain organoids show a defect in spatial organization or if there is an overall downregulation of NPC markers. The authors should provide a comprehensive analysis of rosette formation (e.g., PAX6, SOX2, FOXG1) at different time points during cerebral organoid differentiation. This detailed analysis will contribute to understand the developmental processes and potential disruptions in rosette structure formation within the 70Q/70Q brain organoids.

As mentioned above, we included immunostaining using the makers suggested by the reviewer including ZO1, SOX2, and FOXG1 (**Fig. 1e, Supplementary Fig. 2a**). We also carried out single-cell transcriptomics to address the impact of mHTT on various cell populations within midbrain organoids (**Fig. 2e-g, Supplementary Fig. 3e-f**). Overall, we performed several new experiments in 3D and 2D NPCs to strengthen the conclusion of our study (see answer above). We believe that our findings, based on various 2D and 3D experiments, suggest that mutant HTT affect human neurodevelopment and mitochondrial morpho-dynamics and that these defects are associated with CHCHD2 dysregulation.

3) The authors are encouraged to thoroughly investigate the internal organizational structure of cortical organoids by performing a detailed analysis and comparison of the formation of distinct cortical layers between the control and mutant samples. This analysis would provide valuable insights into the impact of the 70Q/70Q on cortical layer formation.

We believe that such analysis would go beyond the scope of this current work. In fact, it was not our goal to dissect the cortical layer formation but rather to investigate the impact of mHTT on neurometabolism and identified CHCHD2 dysregulation and further investigated its connection to mitochondrial defects. For this reason, we included several new experiments to strengthen our data on mitochondrial-related implications in mHTT mutant neural cells: 1) long-read transcriptomics that identified defects in variant use for genes related to mitochondrial dynamics (**Fig. 4d**); 2) electron microscopy confirming dysfunction of mitochondrial and cristae morphology that were rescued in 0Q/0Q NPCs (**Fig. 5 a-b**); 3) SDS PAGE and blue native PAGE that revealed defects in mitochondrial respiration assembly related to complex IV that were rescued in 0Q/0Q NPCs (**Fig. 5g-j**); 4) immunoblot analysis and quantifications of CHCHD2 (**Supplementary Fig. 4e-f**); 5) identification of a core network of dysregulated factors in HD human neurons to which CHCHD2 belonged and which was associated to processes related to axon guidance, mitochondrial integrated stress response, and Hippo signaling (**Fig. 7a-c**).

4) The authors have observed a reduction in the size of the brain organoids, regardless of the differentiation protocols employed. To provide further insights, it is recommended that the authors conduct additional experiments to investigate whether this reduction in size is attributed to a decrease in the neural progenitor pool or an accelerated differentiation process. This clarification would contribute to a better understanding of the cellular dynamics and fate determination within the brain organoids, thereby strengthening the significance of the findings.

We included additional growth rate analyses of cerebral organoids that also demonstrated a defective growth due to mHTT (**Fig. 1f**). However, we feel that the analyses suggested by the reviewer would go beyond the scope of this current work. In fact, other groups are already working on these aspects (see Zhang et al BioRxiv 2019 “Expanded huntingtin CAG repeats disrupt the balance between neural progenitor expansion and differentiation in human cerebral organoids”). Hence, we prefer to maintain our focus on aspects more related to mitochondria and neurometabolism (see answer above).

5) The authors claim in the Discussion "Under physiological conditions, cortical organoids typically reach a much larger volume over time than midbrain organoids, which instead remain relatively small and are kept under static growth". The authors may consider to refer in the manuscript the studies of Fiorenzano et al. in Nature Communications, 2021, or Jo et al. in Cell Stem Cell, 2017, where midbrain organoids were successfully cultured on an orbital shaker, resulting in the generation of functionally mature and pigmented dopaminergic neurons over a long-term period.

We apologize for the lack of clarity. We removed that sentence and rewrote the text describing midbrain organoids (**lines 184-187**).

6) To better understand the impact of developmental alterations in different brain regions, it is recommended that the authors perform a more detailed analysis of midbrain organoids. Specifically, they should quantify the generation of floor plate rosettes (marked by e.g. FOXA2, SOX2, ZO-1, E-CAD) at early time points and assess the presence of TH+ neurons in long-term cultures within mutant midbrain organoids.

We thank the reviewer for these important suggestions. We performed additional stainings on midbrain organoids demonstrating the presence of progenitors, neurons, and TH-positive dopaminergic neurons (**Supplementary Fig. 3d**). However, in the midbrain organoid protocol that we used (Renner et al eLife 2020) we did not observe rosette formation, thus staining for these markers could not be included. However, to address the composition of midbrain organoids and the impact of mHTT, we carried out single-cell transcriptomics on mutant midbrain organoids, leading to the identification of key disruption in neural progenitor populations (**Fig. 2e-g, Supplementary Fig. 3e-f**).

7) While the authors' efforts are commendable, they have not provided sufficient evidence to establish the mechanism linking metabolic impairment resulting from CHCHD2 reduction with early neurodevelopmental defects (i.e. spatial organization and molecular impairment) in organoids. Given the importance of the findings, it is suggested that the authors utilize the available transcriptomic datasets to provide a mechanistic explanation for the observed defects.

As suggested by the reviewer we have now performed and included several new experiments to demonstrate the impact of CHCHD2 on neurodevelopment and mitochondrial defects in human neural cells. We also performed new and detailed analyses of the multi-omics datasets of HD patient-derive neurons. Altogether we identified that: 1) CHCHD2 (together with CHCHD10) is located closely to the core of the network of genes/proteins/metabolites that are dysregulated in human neurons by mHTT (**Fig. 7a, Supplementary Fig. 8d**), 2) CHCHD2 belonged to the biggest cluster of co-expressed proteins in NGN2 neurons based on protein co-expression network analysis (**Supplementary Fig. 8a**); 3) The core network dysregulated factors in HD human neurons to which CHCHD2 belongs including Hippo signaling genes that were also dysregulated in engineered NPCs carrying mHTT (**Fig. 7bc**).

In addition to omics data, we included experiments to prove the impact of CHCHD2 manipulation. 1) CHCHD2 knock-down impairs neurite growth capacity in human neurons (**Fig. 7e**); 2) CHCHD2 overexpression decreases the aberrant mitochondrial signal in human NPCs carrying mHTT (**Figure 7f-g**). Altogether, we believe that these new data further support our evidence pointing towards a role for CHCHD2 in the neuropathology of HD.

Reviewer #4 (Remarks to the Author):

Lisowski et al used stem cells to study the impact of mutant Huntingtin on neurodevelopment. Introducing a mutant expansion led to abnormal development of cerebral organoids and dysregulation of the CHCHD2 protein. CHCHD2 repression was associated with mitochondrial defects and increased energy expenditure. Removing the expansion normalised CHCHD2 expression and mitochondrial function. These findings suggest that targeting CHCHD2 could be an early intervention for Huntington's disease.

I have a few queries that I'd appreciate being clarified within the manuscript as a revised version:

1. There is a contradiction on line 850 where authors wrote 'profiling by targeted LC-MS'. By definition, metabolic profiling is an untargeted exercise, to maximise coverage and generate a profile of metabolites without any targets in mind.

Indeed, this phrase is misleading, we rephrased to “Metabolites were analyzed by a targeted LC-MS approach” (**line 1146**).

2. Lines 856-857 are pre-mature and renders the method irreproducible. The extracts once aliquot, were subjected to addition of internal standards and then dried in SpeedVac. The pellets were reconstituted in solution described on line 857, not the aliquots themselves.

Thank you for pointing this out, this was corrected accordingly (**line 1153**).

3. It is unclear what 400 metabolites were selected and how they were chosen.

The mass spectrometry lab created targeted methods for more than 400 metabolites over time, which are constantly increased by new metabolites of interest. This collection of metabolites is historically grown and should reflect the main metabolic pathways in a cell, but it is by far not complete. This is mainly due to the availability of pure standards, the price tag of standards, and sometimes the poor performance of metabolites in LC-MS, as well as the workload to create such a reference library. This library was already used for several publications during the last few years (e.g. Gielisch et al, J Proteome Res. 2015). We included this publication, which included the list of all metabolites with method parameters, while describing the metabolites used (**line 1161**).

4. Why were all 400 metabolites normalised to protein abundances obtained from the BCA assay? Are all 400 associated with the proteins measured? The authors should cite if there is an association between the selected 400 metabolites and measured proteins.

This normalization was done solely to correct for input variations between samples. The assumption is that e.g. 15% more protein in a specific sample should also have 15% more metabolites in the same sample.

We chose to normalize for total protein content (not individual proteins). Another possibility would be to normalize for total DNA or RNA content, cell numbers, or for the total metabolite content, or single metabolites/proteins. In the normalization method that we used, no metabolite needs to be associated with any specific individual protein. We believe that this approach is more precise compared to western blot, in which the total input is normalized based on the abundance of one single control band (e.g. ACTB or GAPDH).

5. Supplementary Data 9 contains only 128 metabolites and their associated data. How did authors reduce variables from 400 to 128? Student's t-test columns are empty for these 128 variables and q-values are all greater than 0.05.

Most of the metabolites are not detected in a given sample. This can have several reasons: too less input material, abundance of a metabolite is below the "lower limit of detection", complete absence of a metabolite (depending on the cells/tissue used, e.g. liver cells feature different metabolites than fibroblasts), short lived/unstable metabolites, or matrix interference. All these events may prevent the reliable identification of some metabolite or their reliable quantification. In this particular case, only 128 out of 400 metabolites provided good and quantitative results, which is within the expected range based on our previous experience.

6. What quality control was in place to ensure that metabolites measured were free from analytical errors?

In addition to the input normalization, we used several internal standards to compensate for instrument variations, including C13 labeled metabolites or compounds that are not metabolites present in living organisms. To reliably identify metabolites, we measured 3 transitions per metabolite, calculated the ratios between these transitions (they are stable when the same instrument parameters are used), and recorded the retention time. Lastly, we compared all these data to the pure standards of each metabolite. Only if these data matched, we then considered the metabolite as identified. As we prefer to be very stringent in this regard, only 128 metabolites were identified following the adopted strategy. We believe that such conservative approach is crucial to enable the generation of robust and reproducible data.

7. Functional proteome-metabolome data are interesting and important for this work. How did the authors ensure that measured proteome and metabolome reflects the cellular biomass that was quenched and extracted correctly?

We followed established standard protocols. There are many protocols available; each extraction method indeed can enhance or suppress the abundance of specific metabolites or proteins. This is mainly relevant when different protocols are compared, or specific classes of proteins are searched for (e.g. membrane embedded proteins are more difficult to extract due to the poor solubility). In this particular case, we extracted all samples using the same approach. In this manner, if a membrane protein might be missing, it should then be missing in all samples. This approach would thus not compromise any comparisons between the different sample cohorts.

REVIEWERS' COMMENTS

Reviewer #1 (Remarks to the Author):

The authors have done a nice job of addressing reviewer concerns

Reviewer #2 (Remarks to the Author):

The authors have addressed most, if not all, the comments and questions addressed by the reviewers. The additional experiments and detail provide strength to the data. In particular the additional data provides strength the mechanistic link to CHCHD2 which was lacking in the original submission.

Reviewer #3 (Remarks to the Author):

The authors have addressed the major points raised by the reviewer, and the manuscript, in its current form, is now suitable for publication.

Reviewer #5 (Remarks to the Author):

The authors have done excellent work to address all of the reviewer comments, the manuscript represents important and impactful work with a broad array of methods to explore the molecular pathophysiology of HD.

I have one final suggestion for the metabolomics methods write up to enhance reproducibility. Please include the flow rate and column temperature for the metabolomics method. These recommendations are based on the metabolomics Nature Methods (PMID: 34239102) which serves as a good checklist for metabolomics: [https://www.nature.com/articles/s41592-021-01197-](https://www.nature.com/articles/s41592-021-01197-1)

Point-by-point response

Reviewer #1 (Remarks to the Author):

The authors have done a nice job of addressing reviewer concerns.

Thank you.

Reviewer #2 (Remarks to the Author):

The authors have addressed most, if not all, the comments and questions addressed by the reviewers. The additional experiments and detail provide strength to the data. In particular the additional data provides strength the mechanistic link to CHCHD2 which was lacking in the original submission.

Thank you.

Reviewer #3 (Remarks to the Author):

The authors have addressed the major points raised by the reviewer, and the manuscript, in its current form, is now suitable for publication.

Thank you.

Reviewer #5 (Remarks to the Author):

The authors have done excellent work to address all of the reviewer comments, the manuscript represents important and impactful work with a broad array of methods to explore the molecular pathophysiology of HD.

I have one final suggestion for the metabolomics methods write up to enhance reproducibility. Please include the flow rate and column temperature for the metabolomics method. These recommendations are based on the metabolomics Nature Methods (PMID: 34239102) which serves as a good checklist for metabolomics: <https://www.nature.com/articles/s41592-021-01197-1>.

Thank you. We included the needed information regarding flow rate and column temperature for metabolomics in the methods section (lines 1144-1146).